# Nonparametric Modern Hopfield Models

**Jerry Yao-Chieh Hu** [* 1]  **Bo-Yu Chen** [* 2]  **Dennis Wu** [1]  **Feng Ruan** [3]  **Han Liu** [1 3]

## Abstract

We present a nonparametric interpretation for deep learning compatible modern Hopfield models and utilize this new perspective to debut efficient variants. Our key contribution stems from interpreting the memory storage and retrieval processes in modern Hopfield models as a nonparametric regression problem subject to a set of query-memory pairs. Interestingly, our framework not only recovers the known results from the original dense modern Hopfield model but also fills the void in the literature regarding efficient modern Hopfield models, by introducing *sparse-structured* modern Hopfield models with sub-quadratic complexity. We establish that this sparse model inherits the appealing theoretical properties of its dense analogue — connection with transformer attention, fixed point convergence and exponential memory capacity. Additionally, we showcase the versatility of our framework by constructing a family of modern Hopfield models as extensions, including linear, random masked, top-$K$ and positive random feature modern Hopfield models. Empirically, we validate our framework in both synthetic and realistic settings for memory retrieval and learning tasks. Code is available at GitHub; future updates are on arXiv.

## 1 Introduction

We tackle the challenges in computational efficiency of modern Hopfield models (Wu et al., 2024b; Hu et al., 2023;

Ramsauer et al., 2020) — a class of transformer-compatible associative memories. In short, we present a nonparametric framework[1], and then debuting efficient modern Hopfield models with sub-quadratic complexity and appealing theoretical properties. Such a construction is of practical importance. As in many Hopfield-centric methods (Hu et al., 2024a; Wu et al., 2024a; Xu et al., 2024; Wu et al., 2024b; Schimunek et al., 2023; Fürst et al., 2022; Paischer et al., 2022; Seidl et al., 2022; Widrich et al., 2020), modern Hopfield models (and their derived deep learning layers) serve as powerful alternatives to the attention mechanism with additional functionalities, but lack efficient implementation for gigantic deep models (Hu et al., 2023, Section C.2).

This issue becomes more prominent in this era of Large Foundation Models (Bommasani et al., 2021). Foundation models are huge transformer-based models pretrained on massive datasets, and play a central role not only in machine learning but also in a wide range of scientific domains, such as ChatGPT (Brown et al., 2020; Floridi & Chiriatti, 2020) for natural language, BloombergGPT (Wu et al., 2023) for finance, DNABERT (Zhou et al., 2024a;b; Ji et al., 2021) for genomics, and many others. To push toward Hopfield-based large foundation models, this work provides a timely efficient solution, back-boned by a solid theoretical ground.

Modern Hopfield models (Ramsauer et al., 2020), motivated by the dense associative memory models (Demircigil et al., 2017; Krotov & Hopfield, 2016), are (auto-)associative memory models that (i) have exponential memory capacity, (ii) retrieve stored patterns based on input queries with only one retrieval step, and (iii) are compatible with deep learning architectures. They achieve (i) by adopting highly nonlinear energy functions, (ii) by adopting a memory-retrieval dynamics ensuring monotonic minimization of the energy function, and (iii) by the connection between their memory retrieval dynamics and attention mechanism. Deepening (ii) and (iii), Hu et al. (2023) and Wu et al. (2024b) propose a theoretical framework for deriving modern Hopfield models

---

[*]Equal contribution   [1]Department of Computer Science, Northwestern University, Evanston, IL, USA [2]Department of Physics and Computer Science, National Taiwan University, Taipei, Taiwan [3]Department of Statistics and Data Science, Northwestern University, Evanston, IL, USA. Correspondence to: Jerry Yao-Chieh Hu <jhu@u.northwestern.edu>, Bo-Yu Chen <b12202023@ntu.edu.tw>, Dennis Wu <hibb@u.northwestern.edu>, Feng Ruan <fengruan@northwestern.edu>, Han Liu <hanliu@northwestern.edu>.

*Proceedings of the 42nd International Conference on Machine Learning*, Vancouver, Canada. PMLR 267, 2025. Copyright 2025 by the author(s).

---

[1]After completing the draft, the authors became aware of the independent study by Nguyen et al. (2024) on a nonparametric (primal-dual) formulation for Transformer attention. To our knowledge, Nguyen et al. (2024) were the first to cast attention as an $\epsilon$-SVR solution (Awad et al., 2015; Vapnik, 2013; Kar & Karnick, 2012; Schölkopf & Smola, 2002). Our study presents a similar framework but focuses on Hopfield-style associative memory.

using various entropic regularizers. In addition, they introduce a sparse extension of the original modern Hopfield model to handle its computational burden and vulnerability to noise. As a result, their proposal not only connects to sparse attention mechanism (Correia et al., 2019; Martins & Astudillo, 2016) but also offers both provably computational advantages and robust empirical performance.

However, there are still some missing pieces:

- **(P1) Lack of Efficiency.** Computationally, while Hu et al. (2023) and Wu et al. (2024b) indeed introduce sparsity into their model, this sparsity does not implies computational efficiency. In fact, it only increases efficiency at the level of memory retrieval, (i.e. the sparsity in (Hu et al., 2023; Wu et al., 2024b) only leads to faster memory retrieval but not necessarily shorter running time, as discussed in (Hu et al., 2023, Section C.2)). Namely, the sparse modern Hopfield model still suffers by the $\mathcal{O}(n^2)$ complexity (with the input sequence length $n$), which hampers its scalability[2].

- **(P2) Lack of Rigorous Analysis on Sparsity.** Theoretically, because Hu et al. (2023) choose not to make strong assumptions (on the memory and query patterns) in order to maintain their model's generality, they only offer qualitative justifications (Hu et al., 2023, Section 3). They do not rigorously characterize how sparsity impacts different aspects of the sparse model, e.g., the retrieval error, the well-separation condition, and the memory capacity.

- **(P3) Incomplete Connection between Attention and Hopfield Models.** Methodologically, while numerous variants of the attention module exist (Choromanski et al., 2021; Katharopoulos et al., 2020; Beltagy et al., 2020; Child et al., 2019), Hu et al. (2023) only bridge a subset of them to modern Hopfield models. A natural question arises: How can we integrate the advancements of state-of-the-art attention into modern Hopfield models? As noted in (Hu et al., 2024b; 2023; Wu et al., 2024b), this question is far from trivial. Naively substituting the softmax activation function with other alternatives does not necessarily yield well-defined Hopfield models and might sabotage their desirable properties and functionalities.

**Overview of Our Theoretical Results.** To fill these gaps, this work presents a nonparametric framework for deep learning compatible modern Hopfield models.

To fill **(P1)**, this framework allows us to not only recover the standard dense modern Hopfield model (Ramsauer et al., 2020), but also introduce an efficient modern Hopfield model, termed *sparse-structured* model (Theorem 3.2).

To fill **(P2)**, our framework facilitates the derivation of a

retrieval error bound of the sparse modern Hopfield with explicit sparsity dependence (Theorem 4.1). This bound offers rigorous characterizations of the sparsity-induced advantages of the sparse model compared with its dense counterpart, including higher precision in memory retrieval (Corollary 4.1.1 and Corollary 4.1.2), enhanced robustness to noise (Remark 4.2) and exponential-in-$d$ capacity (Lemma 4.1 and Proposition 4.1, $d$ refers to pattern size). Interestingly, unlike existing Hopfield models (Hu et al., 2023; Wu et al., 2023; Ramsauer et al., 2020) requiring an explicit energy function to guarantee the stability of the model, we show that the sparse modern Hopfield model guarantees the fixed-point convergence even without details of the Hopfield energy function (Corollary 4.1.3).

To fill **(P3)**, beyond introducing the sparse modern Hopfield model, our framework supports a family of modern Hopfield models that connect with various attention variants. This complements the findings in (Hu et al., 2023; Wu et al., 2024b), pushing us toward a more unified understanding.

**Contributions.** Our contributions are as follows:

- We propose a nonparametric framework for deep learning compatible modern Hopfield models. Building upon this, we introduce the first efficient sparse modern Hopfield model with sub-quadratic complexity.

- We provide rigorous characterizations of the sparsity-induced advantages of the proposed efficient model: tighter retrieval error bound (Corollary 4.1.1 and Corollary 4.1.2), stronger noise robustness (Remark 4.2) and exponential-$d$-capacity (Lemma 4.1 and Proposition 4.1).

- Based on the proposed framework, we construct a family of modern Hopfield models connecting to many existing attention variants (Choromanski et al., 2021; Zaheer et al., 2020; Beltagy et al., 2020; Katharopoulos et al., 2020). We also verify their efficacy through thorough numerical experiments in both synthetic and realistic settings (*memory retrieval* and *learning* performance in Appendices G.1 to G.4 and efficiency in Appendix G.5.).

**Notations.** We denote vectors by lower case bold letters, and matrices by upper case bold letters. We write $\langle \mathbf{a}, \mathbf{b} \rangle := \mathbf{a}^\top \mathbf{b}$ as the inner product for vectors $\mathbf{a}, \mathbf{b}$. Let $\mathbf{a}[i]$ denotes the $i$-th element of vector $\mathbf{a}$. The index set $\{1, \cdots, I\}$ is denoted by $[I]$, where $I \in \mathbb{N}_+$. The spectral norm is denoted by $\|\cdot\|$, which is equivalent to the $l_2$-norm when applied to a vector. We denote the memory patterns by $\boldsymbol{\xi} \in \mathbb{R}^d$ and the query pattern by $\mathbf{x} \in \mathbb{R}^d$, and $\boldsymbol{\Xi} := [\boldsymbol{\xi}_1, \cdots, \boldsymbol{\xi}_M] \in \mathbb{R}^{d \times M}$ as shorthand for stored memory patterns $\{\boldsymbol{\xi}_\mu\}_{\mu \in [M]}$. Moreover, we set $m := \mathrm{Max}_{\mu \in [M]} \|\boldsymbol{\xi}_\mu\|$.

**Organization.** Section 2 reviews modern Hopfield models. Section 3 presents a nonparametric construction for

---

[2]See Remark 3.6 for the connection between time complexity of attention and of modern Hopfield models.

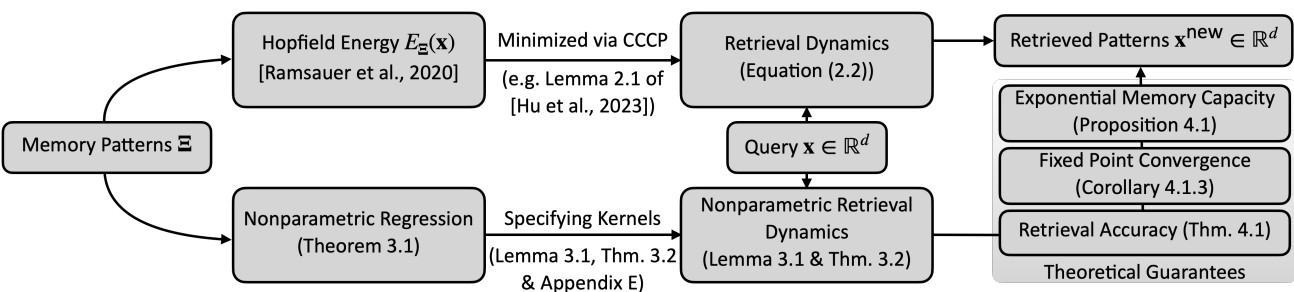

*Figure 1.* **A High-level Visualization.** The upper row formalizes the memorization and retrieval of standard modern Hopfield model (Wu et al., 2024b; Hu et al., 2023; Ramsauer et al., 2020). The lower row conceptualizes our nonparametric interpretation.

modern Hopfield models, and debut the sparse-structured (efficient) modern Hopfield models. Section 4 provides the theoretical analysis on the sparse-structured modern Hopfield models. Appendix E includes a family of modern Hopfield models as possible extensions. We conduct numerical experiments to support our framework in Appendix G.

## 1.1  Related Work

**Modern Hopfield Models for Deep Learning.** The classical Hopfield models (Amari, 1972; Hopfield, 1984; 1982; Krotov & Hopfield, 2016) are canonical models of the human brain's associative memory. Their primary function is the storage and retrieval of specific memory patterns. Recently, a resurgence of interest in Hopfield models within the machine learning field is attributed to developments in understanding memory storage capacities (Krotov & Hopfield, 2016; Demircigil et al., 2017; Wu et al., 2024a), innovative architecture (Hoover et al., 2023; Seidl et al., 2022; Fürst et al., 2022; Ramsauer et al., 2020), and their biological plausibility (Kozachkov et al., 2022; Krotov & Hopfield, 2021). Notably, the modern Hopfield models (Wu et al., 2024b; Hu et al., 2023; Ramsauer et al., 2020; Brandstetter, 2021), demonstrate not only a strong connection to the transformer attention mechanisms in deep learning, but also superior performance, and a theoretically guaranteed exponential memory capacity. In this regard, seeing the modern Hopfield models as an advanced extension of attention mechanisms opens up prospects for crafting Hopfield-centric architectural designs. Therefore, their applicability spans diverse areas like drug discovery (Schimunek et al., 2023), immunology (Widrich et al., 2020), tabular learning (Xu et al., 2024), time series forecasting (Wu et al., 2024b; Auer et al., 2024), reinforcement learning (Paischer et al., 2022), and large foundation models (Hu et al., 2024a; Fürst et al., 2022). This work emphasizes refining this line of research towards efficient models. We posit that this effort is crucial in guiding future research towards Hopfield-driven design paradigms, especially for larger models.

**Sparse Modern Hopfield Model.** (Ramsauer et al., 2020)

establish a connection between Hopfield models and the vanilla softmax attention. Motivated by this connection, (Hu et al., 2023; Wu et al., 2024b) (and later (Martins et al., 2023)) propose a theoretical framework for modern Hopfield models based on the relationship between entropic regularizers and finite-domain distributions with varying support sets. Importantly, they not only show that (Ramsauer et al., 2020) is just special case within their framework but also propose a sparse extension with superior properties (e.g., robust representation learning, fast fixed-point convergence, and exponential memory capacity) and connection to certain types of sparse attention. However, this is not end of the story. As highlighted in (Hu et al., 2023, Section E), their framework only bridges a subset of existing attention variants (with dense quadratic attention score matrix) and hence is not complete. This work fills this theoretical gap by providing a principle construction for the many modern Hopfield models with theoretical guarantees. Moreover, our framework supports a family of modern Hopfield models mirroring many popular structured efficient attention mechanisms, including Attention with Pre-defined Patterns (each sequence token attends to a predetermined subset of tokens instead of the entire sequence, e.g, Big Bird (Zaheer et al., 2020), Longformer (Beltagy et al., 2020), Blockwise (Qiu et al., 2019), Sparse (Child et al., 2019)), and Kernelized Attention (e.g., Performer (Choromanski et al., 2021), Linear (Clevert et al., 2015) and Multi-head (Vaswani et al., 2017)).

**Sparse-Structured Hopfield Models.** After completing this work, the authors attended ICML 2024 and learned of a study by Santos et al. (2024b) proposing a different model, also named the sparse-structured Hopfield network. Both models emphasize sparse retrieval patterns. However, Santos et al. (2024a;b) differ in (i) introducing sparsity via optimized Fenchel-Young energies and (ii) enhancing efficiency using the SparseMAP transformation and active set algorithm (Niculae et al., 2018) with predefined $k$-ary relations among stored patterns or top-$k$ operations.

## 2 Background: Modern Hopfield Models

This section presents the ideas we build on.

Let $\mathbf{x} \in \mathbb{R}^d$ be the input query pattern and $\boldsymbol{\Xi} = [\boldsymbol{\xi}_1, \cdots, \boldsymbol{\xi}_M] \in \mathbb{R}^{d \times M}$ the $M$ memory patterns.

**Hopfield Models.** The aim of Hopfield models (Amari, 1972; Hopfield, 1982; 1984) is to store these memory patterns $\boldsymbol{\Xi}$ and retrieve a specific memory $\boldsymbol{\xi}_\mu$ when given a query $\mathbf{x}$. They achieve these by embedding the memories in the energy landscape $E(\mathbf{x})$ of a physical system, where each memory $\boldsymbol{\xi}_\mu$ corresponds to a local minimum. When a query $\mathbf{x}$ is presented, the model initiates energy-minimizing retrieval dynamics $\mathcal{T}$ at the query, which then navigate the energy landscape to find the nearest local minimum, effectively retrieving the memory most similar to the query.

These models comprise two primary components: an *energy function* $E(\mathbf{x})$ that encodes memories into its local minima, and a *retrieval dynamics* $\mathcal{T}(\mathbf{x})$ that fetches a memory by iteratively minimizing $E(\mathbf{x})$ starting with a query. Constructing $E(\mathbf{x})$, is straightforward. As outlined in (Krotov & Hopfield, 2016), memories get encoded into $E(\mathbf{x})$ using the *overlap-construction*: $E(\mathbf{x}) = F(\boldsymbol{\Xi}^\mathsf{T}\mathbf{x})$, where $F : \mathbb{R}^M \to \mathbb{R}$ is a smooth function. This encourages the memories $\{\boldsymbol{\xi}_\mu\}_{\mu \in [M]}$ to reside at the stationary points of $E(\mathbf{x})$, i.e., $\nabla_\mathbf{x} F(\boldsymbol{\Xi}^\mathsf{T}\mathbf{x})|_{\boldsymbol{\xi}_\mu} = 0$ for all $\mu \in [M]$. The choice of $F$ results in different Hopfield model types, as demonstrated in (Krotov & Hopfield, 2021; Ramsauer et al., 2020; Demircigil et al., 2017; Krotov & Hopfield, 2016).

However, determining a suitable retrieval dynamics, $\mathcal{T}$, for a given energy $E(\mathbf{x})$ is more challenging. For effective memory retrieval, the iterative retrieval dynamics $\mathcal{T}$ must:

(T1) Monotonically reduce $E(\mathbf{x})$ when applied iteratively.

(T2) Ensure its fixed points coincide with the stationary points of $E(\mathbf{x})$ for precise retrieval.

**Modern Hopfield Models.** Ramsauer et al. (2020) propose the modern Hopfield model with a specific set of $E$ and $\mathcal{T}$ satisfying above requirements, and integrate it into deep learning architectures via its strong connection with attention mechanism, offering enhanced performance, and theoretically guaranteed exponential memory capacity. Specifically, they introduce the energy function:

$$E(\mathbf{x}) = -\operatorname{lse}(\beta, \boldsymbol{\Xi}^\mathsf{T}\mathbf{x}) + \frac{1}{2}\langle \mathbf{x}, \mathbf{x} \rangle, \qquad (2.1)$$

where the retrieval dynamics is given by

$$\mathbf{x}^{\text{new}} = \mathcal{T}_{\text{Dense}}(\mathbf{x}) = \boldsymbol{\Xi} \cdot \operatorname{Softmax}(\beta \boldsymbol{\Xi}^\mathsf{T}\mathbf{x}). \qquad (2.2)$$

The function $\operatorname{lse}(\beta, \mathbf{z}) \coloneqq \log\left(\sum_{\mu=1}^M \exp\{\beta z_\mu\}\right)/\beta$ is the log-sum-exponential for any given vector $\mathbf{z} \in \mathbb{R}^M$ and $\beta > 0$. Their analysis reveals that:

1. $\mathcal{T}_{\text{Dense}}$ dynamics converge well (T2) and can retrieve patterns accurately in just one step (T1).

2. Modern Hopfield model from (2.1) possesses an exponential memory capacity in pattern size $d$.

3. Notably, the one-step approximation of $\mathcal{T}_{\text{Dense}}$ mirrors the attention mechanism in transformers, leading to a novel deep learning architecture design: the Hopfield layers.

**Attention $\leftrightarrow$ Modern Hopfield Model.** To see above 3., suppose that $\mathbf{X}$ and $\boldsymbol{\Xi}$ are embedded from the *raw* query $\mathbf{R}$ and $\mathbf{Y}$ memory patterns, respectively, via $\mathbf{X}^\mathsf{T} = \mathbf{R}\mathbf{W}_Q \coloneqq \mathbf{Q}$, and $\boldsymbol{\Xi}^\mathsf{T} = \mathbf{Y}\mathbf{W}_K \coloneqq \mathbf{K}$, with some projection matrices $\mathbf{W}_Q$ and $\mathbf{W}_K$. Then, taking the transpose of $\mathcal{T}$ in (2.2) and multiplying with $\mathbf{W}_V$ such that $\mathbf{V} \coloneqq \mathbf{K}\mathbf{W}_V$, we obtain

$$\mathbf{Z} \coloneqq \mathbf{Q}^{\text{new}}\mathbf{W}_V = \operatorname{Softmax}\left(\beta \mathbf{Q}\mathbf{K}^\mathsf{T}\right)\mathbf{V}. \qquad (2.3)$$

This enables modern Hopfield models to serve as alternatives to attention mechanism with extra functionalities.

Given the equivalence (2.3), one might wonder if the quest for efficient modern Hopfield models is equivalent to seeking efficient attention mechanisms (Tay et al., 2022), specifically in terms of finding efficient implementations of the $\operatorname{Softmax}$ matrix computation. We contend that they are not the same. To build a modern Hopfield model, we expect not only its retrieval dynamics to connect to attention mechanism, but also it to serve as an associative memory model (Hu et al., 2024a; Wu et al., 2024b; Hu et al., 2023; Ramsauer et al., 2020) by design. Moreover, we observe that (T1) and (T2) are essentially about encoding memories onto the fixed points of $\mathcal{T}$.

These motivate us to view the construction of $\mathcal{T}$ as a learning problem: we aim to learn a function $\mathcal{T}$ satisfying (T2) from a dataset consisting of query-memory pairs. Thus, rather than using the traditional Hopfield model's learning rule — where the model memorizes memories by defining an energy function, like the overlap-construction (Hu et al., 2023) — we interpret the memorization process as learning a function that maps queries to memories. This new perspective allows us to construct novel modern Hopfield models that are equivalent to various attention variants.

## 3 Nonparametric Modern Hopfield Models

**High-Level Overview.**

- In Section 3.1, we formulate the memory storage and retrieval of modern Hopfield models as a nonparametric regression problem. We first align the definition of $\mathcal{T}$ (the retrieval dynamics (2.2)) with a nonparametric regression problem subject to a set of query-memory pairs. Then, by solving for optimality, we derive a nonparametric formulation of $\mathcal{T}$.

- In Section 3.2, we showcase our framework with two special cases: the standard dense modern Hopfield model (Ramsauer et al., 2020) (Lemma 3.1), and a new, efficient sparse-structured modern Hopfield model (Theorem 3.2).

## 3.1 Retrieval Dynamics

The retrieval dynamics (2.2) $\mathcal{T}_{\Xi}(\mathbf{x}) : \mathbb{R}^d \to \mathbb{R}^d$ maps an input query $\mathbf{x}$ to $\mathcal{T}_{\Xi}(\mathbf{x})$, with the aim of retrieving the memory pattern $\boldsymbol{\xi}_\mu$ closest to $\mathbf{x}$. To formalize this notion of retrieval, we need a few definitions and notation.

**Definition 3.1** (Generalized Fixed Point (Sriperumbudur & Lanckriet, 2009)). We say a set $\mathcal{S} \subseteq \mathbb{R}^d$ a *generalized fixed point* with respect to $\mathcal{T}_{\Xi}$ if $\mathcal{T}_{\Xi}(\mathbf{y}) \in \mathcal{S}$ for every $\mathbf{y} \in \mathcal{S}$.

**Remark 3.1** (Fixed Point). In contrast, a *fixed point* of $\mathcal{T}_{\Xi}$ is a point $\mathbf{y}$ for which $\mathcal{T}_{\Xi}(\mathbf{y}) = \mathbf{y}$.

In particular, if the retrieval dynamics is initiated at $\mathbf{x} \in \mathcal{S}$ where $\mathcal{S}$ is an invariant set[3], then subsequent iterates such as $\mathcal{T}_{\Xi}(\mathbf{x})$, $\mathcal{T}_{\Xi} \circ \mathcal{T}_{\Xi}(\mathbf{x})$, ... remain in the invariant set $\mathcal{S}$.

Now we introduce a neighborhood — $\mathcal{S}_\mu$, a ball of radius $R$ — at every memory pattern $\boldsymbol{\xi}_\mu$:

$$\mathcal{S}_\mu = \{ \boldsymbol{\xi} \mid \|\boldsymbol{\xi} - \boldsymbol{\xi}_\mu\| \le R \},$$

$$\text{where} \quad R := \frac{1}{2} \min_{\mu,\nu \in [M]; \mu \ne \nu} \|\boldsymbol{\xi}_\mu - \boldsymbol{\xi}_\nu\|.$$

By definition, neighborhoods associated with distinct memory patterns do not overlap: $\mathcal{S}_\mu \cap \mathcal{S}_\nu = \emptyset$ for $\mu \ne \nu$. To measure the progress of the dynamics in retrieving the memory pattern, we introduce the notion of memory storage and $\epsilon$-retrieval.

**Definition 3.2** (Storage and $\epsilon$-Retrieval). A memory pattern $\boldsymbol{\xi}_\mu$ is *stored* if $\mathcal{S}_\mu$ is a generalized fixed point of $\mathcal{T}$. A memory pattern $\boldsymbol{\xi}_\mu$ gets *$\epsilon$-retrieved* by $\mathcal{T}_{\Xi}$ with an input query $\mathbf{x}$ if $\|\mathcal{T}_{\Xi}(\mathbf{x}) - \boldsymbol{\xi}_\mu\| \le \epsilon$.

In below, when the context is clear, we suppress the notation dependence of $\mathcal{T}_{\Xi}$ on the memory patterns $\Xi$ for simplicity.

Definition 3.2 states that for an input $\mathbf{x}$ around a stored memory pattern $\boldsymbol{\xi}$, its corresponding mapping output $\mathcal{T}(\mathbf{x})$ should be located in the same sphere $\mathcal{S}$. This motivates us to view $\mathcal{T}$ as a function aiming to map the query $\mathbf{x}$ onto its nearest memory $\boldsymbol{\xi}$ within an error-tolerance margin $R$. More precisely, we construct such a function satisfying Definition 3.2 as a learning problem, using memory patterns as data. A natural choice for doing this function is through the soft-margin SVR (Awad et al., 2015; Vapnik, 2013; Kar & Karnick, 2012; Schölkopf & Smola, 2002) (also see Appendix C.1 for a concise overview): it fits the best hyperplane to the data points within a predefined error margin,

---

[3]A generalized fixed point $\mathcal{S}$ with respect to $\mathcal{T}_{\Xi}$ is also an *invariant set* with respect to $\mathcal{T}_{\Xi}$.

aiming to minimize the error rate while ensuring the model remains insensitive to errors within a certain threshold.

We first define the regression model. Given a weight matrix $\mathbf{W} \in \mathbb{R}^{d \times D_\Phi}$, and a feature map $\Phi : \mathbb{R}^d \to \mathbb{R}^{D_\Phi}$, denote $f_{W,\Phi} : \mathbb{R}^d \to \mathbb{R}^d$ to be the mapping

$$f_{W,\Phi}(\mathbf{x}) = \mathbf{W}\Phi(\mathbf{x}). \tag{3.1}$$

Denote $\mathcal{K}(x_1, x_2) := \langle \Phi(x_1), \Phi(x_2) \rangle$. This is a positive semidefinite kernel, and there is a unique RKHS $\mathcal{H}$ associated with this kernel $\mathcal{K}$ (Wainwright, 2019, Theorem 12.11).

To cast $\mathcal{T}$ as a SVR problem using (3.1), we now specify the data points that $f(\mathbf{x})$ should fit. Since the goal of $\mathcal{T}$ is to retrieve the memory pattern most similar to given query $\mathbf{x}$, we consider the training dataset $\mathcal{D} = \{ (\boldsymbol{\xi}_\mu + \delta\boldsymbol{\xi}_\mu, \boldsymbol{\xi}_\mu) \}_{\mu \in [M]}$. Namely, the input query $\mathbf{x} = \boldsymbol{\xi}_\mu + \delta\boldsymbol{\xi}_\mu$ is the contaminated target memory pattern with noise $\delta\boldsymbol{\xi}_\mu$, and the output $\mathbf{y} = \boldsymbol{\xi}_\mu$ is target memory pattern. For convenience, we shorthand $[\boldsymbol{\xi}_1 + \delta\boldsymbol{\xi}_1, \cdots, \boldsymbol{\xi}_M + \delta\boldsymbol{\xi}_M] = \Xi_\delta \in \mathbb{R}^{d \times M}$ as the contaminated memory patterns.

Next, we frame the memorization in modern Hopfield models as fitting $f$ to the dataset $\mathcal{D}$, and obtain the following nonparametric (support vector) regression problem. Given a dataset $\mathcal{D} = \{ (\boldsymbol{\xi}_\mu + \delta\boldsymbol{\xi}_\mu, \boldsymbol{\xi}_\mu) \}_{\mu \in [M]}$, consider the support vector regression using the feature map $\Phi$

$$\underset{\mathbf{W}, \boldsymbol{\eta}, \widetilde{\boldsymbol{\eta}}}{\text{Min}} \frac{1}{2} \|\mathbf{W}\|^2 + C \sum_{\mu=1}^{M} \langle \mathbb{1}, (\boldsymbol{\eta}_\mu + \widetilde{\boldsymbol{\eta}}_\mu) \rangle \quad \text{s.t.} \tag{3.2}$$

$$\begin{cases} -(\epsilon'\mathbb{1} + \widetilde{\boldsymbol{\eta}}_\mu) \le \boldsymbol{\xi}_\mu - \langle \mathbf{W}, \Phi(\boldsymbol{\xi}_\mu + \delta\boldsymbol{\xi}_\mu) \rangle \le \epsilon'\mathbb{1} + \boldsymbol{\eta}_\mu \\ \epsilon'\mathbb{1} + \boldsymbol{\eta}_\mu \le \epsilon\mathbb{1}/\sqrt{d} \\ \boldsymbol{\eta}_\mu \ge 0, \widetilde{\boldsymbol{\eta}}_\mu \ge 0, \quad \forall \mu \in [M], \end{cases}$$

where the constraints are component-wise, $\epsilon' > 0$ is a component-wise error margin, $C \ge 0$ is a penalty coefficient, and $\epsilon > 0$ is the memory retrieval error. We denote the unique (given the strong convexity of the optimization problem (3.2)) minimizer as $(\mathbf{W}_\Phi^*, \boldsymbol{\eta}_\Phi^*, \widetilde{\boldsymbol{\eta}}_\Phi^*)$, and the solution to (3.2) as $\mathcal{T}_{\text{SVR}}(\mathbf{x})$. By solving the optimality via the Lagrangian duality, we obtain the following.

**Theorem 3.1.** Let $\boldsymbol{\alpha}, \widetilde{\boldsymbol{\alpha}}$ denote the Lagrangian multipliers of the dual problem of (3.2). Let $\mathbf{W}^\star := (\mathbf{w}_1^\star, \dots \mathbf{w}_d^\star)^\mathsf{T} \in \mathbb{R}^{d \times D_\Phi}$ denote the minimizer of (3.2). Then,

$$\mathbf{w}_i^\star = \sum_{\mu=1}^{M} \underbrace{(\boldsymbol{\alpha}_\mu[i] - \widetilde{\boldsymbol{\alpha}}_\mu[i])}_{\in \mathbb{R}} \underbrace{\Phi(\boldsymbol{\xi}_\mu + \delta\boldsymbol{\xi}_\mu)}_{\in \mathbb{R}^{D_\Phi}},$$

where $\mathbf{a}[i]$ denotes the $i$-th element of a vector $\mathbf{a}$.

*Proof.* See Appendix D.1 for a detailed proof. $\square$

For any featurization map $\Phi$, Theorem 3.1 introduces a map $\mathcal{T}_{\text{SVR},\Phi} := f_{\mathbf{W}_\Phi^*, \Phi}$. By construction, for any $\Phi$, $\mathcal{T}_{\text{SVR},\Phi}$

obeys the $\epsilon$-retrieval property $\|\mathcal{T}_{\text{SVR},\Phi}(x) - \boldsymbol{\xi}_\mu\| \le \epsilon$, for any $\mu$ and $\mathbf{x} \in \mathcal{S}_\mu$. Hence, we arrive a nonparametric interpretation for constructing many modern Hopfield models. Given an input query $\mathbf{x}$, the $i$-th component of the retrieved pattern by applying $\mathcal{T}_{\text{SVR}}(\mathbf{x})$ once is

$$\mathbf{x}^{\text{new}}[i] := \mathcal{T}_{\text{SVR}}(\mathbf{x})[i] = \langle \mathbf{w}_i^\star, \Phi(\mathbf{x}) \rangle. \qquad (3.3)$$

**Remark 3.2.** Note that $\epsilon'$ is the component-wise SVR error, *not* the $\epsilon$ in Hopfield retrieval error defined in Definition 3.2.

**Remark 3.3.** Without any assumption on $\epsilon$, $\mathcal{T}_{\text{SVR}}$ converges to *generalized* fixed points, in contrast to the fixed point convergence in (Hu et al., 2023; Ramsauer et al., 2020). Thus, there is no multiple update convergence for $\mathcal{T}_{\text{SVR}}$ without specifying $\Phi$ (and thereby proving the fixed point convergence property.) We provide specific $\Phi$ with provably fixed point convergence in Section 3.2 and Remark 4.3.

**Remark 3.4.** This regression problem is nonparametric. That is, it does not assume a specific functional form for $\mathcal{T}_{\text{SVR}}$ and is flexible in the number of parameters, allowing the number of support vectors to adjust based on the data.

Intuitively, this optimization problem learns a $\mathcal{T}_{\text{SVR},\Phi}$ to replace $\mathcal{T}$ from the training dataset $\mathcal{D} = \{(\boldsymbol{\xi}_\mu + \delta\boldsymbol{\xi}_\mu, \boldsymbol{\xi}_\mu)\}_{\mu \in [M]}$. Thus, for any given query $\boldsymbol{\xi}_\mu + \delta\boldsymbol{\xi}_\mu$, $\mathcal{T}_{\text{SVR},\Phi}$ retrieves a target memory pattern $\boldsymbol{\xi}_\mu$ with $\epsilon$ precision, for all $\mu \in [M]$. Specifically, this $\epsilon$ precision comes from the upper bound of the maximum component-wise error $\epsilon' + \boldsymbol{\eta}_\mu[i]$ (and $\epsilon' + \widetilde{\boldsymbol{\eta}}_\mu[i]) \le \epsilon/\sqrt{d}$, defined in (3.2). This choice of SVR error margin mimics the $\epsilon$-retrieval of modern Hopfield models via the flexibility of soft-margin SVR. As a result, the objective of the SVR problem (3.2) coincides with the memorization and retrieval processes of modern Hopfield models. While $\mathcal{T}$ retrieves memory patterns $\{\boldsymbol{\xi}_\mu\}_{\mu \in [M]}$ based on $\mathbf{x}$ with an error tolerance $\epsilon$, the SVR problem (3.2)

- **(Memorization:)** Fits a function $\mathcal{T}_{\text{SVR}}$ satisfying Definition 3.2, which

- **(Retrieval:)** Maps queries onto memory patterns within a component-wise error-margin $\epsilon/\sqrt{d}$.

Importantly, Theorem 3.1 enables us to derive a family of nonparametric modern Hopfield models through constructing their retrieval dynamics with various kernel functions $\Phi(\cdot)$, including Dense (Ramsauer et al., 2020), Linear (Katharopoulos et al., 2020), Multi-Head (Vaswani et al., 2017), Sparse-Structured (Zaheer et al., 2020; Beltagy et al., 2020; Child et al., 2019) and Generalized Kernelizable or Positive Random Features (Choromanski et al., 2021) modern Hopfield models. Appendix E includes constructions of these models as extensions from our framework.

## 3.2 Nonparametric Dense and Sparse-Structured Modern Hopfield Models

In this section, we showcase the nonparametric framework Theorem 3.1 with two special cases. First, we recover the standard dense modern Hopfield model (Ramsauer et al., 2020). Then, we introduce the efficient *sparse-structured modern Hopfield models* with sub-quadratic complexity.

**Dense Modern Hopfield Model (Ramsauer et al., 2020).**

**Lemma 3.1** (Nonparametric Dense Modern Hopfield Model)**.** Let $\Phi(\cdot) = (\phi_0^{(0)}, \phi_1^{(1)}, \ldots, \phi_{D_1}^{(1)}, \ldots, \phi_1^{(n)}, \ldots, \phi_{D_n}^{(n)}, \ldots)$ with the formulation, for $1 \le D' \le D_n$ with $D_n := \binom{d+n-1}{n}$,

$$\phi_{D'}^{(n)} := \frac{(\sqrt{\beta}x_1)^{\ell_1} \cdots (\sqrt{\beta}x_d)^{\ell_d}}{\sum_{\mu=1}^M \langle \Phi(\boldsymbol{\xi}_\mu + \delta\boldsymbol{\xi}_\mu), \Phi(\mathbf{x}) \rangle \cdot \sqrt{\ell_1! \cdots \ell_d!}}, \quad (3.4)$$

where $\ell_1 + \cdots + \ell_d = n$. By Theorem 3.1, fitting $\mathcal{T}_{\text{SVR}}$ on $\mathcal{D}$ following (3.2) gives

$$\mathcal{T}_{\text{Dense}}(\mathbf{x}) = \boldsymbol{\Xi}\,\text{Softmax}\left(\beta\boldsymbol{\Xi}_\delta^{\mathsf{T}}\mathbf{x}\right) \in \mathbb{R}^d, \qquad (3.5)$$

where $\boldsymbol{\Xi}_\delta := [\boldsymbol{\xi}_1 + \delta\boldsymbol{\xi}_1, \cdots, \boldsymbol{\xi}_M + \delta\boldsymbol{\xi}_M] \in \mathbb{R}^{d \times M}$ denotes the contaminated memory patterns.

*Proof Sketch.* We first select $\Phi$ to be the Taylor expansion of the $\exp$ function via the $\exp$ kernel's feature expansion (Nguyen et al., 2024; Hamid et al., 2014; Kar & Karnick, 2012; Schölkopf & Smola, 2002). By solving the optimization problem (3.2), we arrive a retrieval dynamics resembling (2.2). See Appendix D.2 for a detailed proof. $\square$

**Remark 3.5** (*Hetero-* v.s. *Auto-*Associative Memory.)**.** So far, we derive a nonparametric framework for hetero-associative modern Hopfield models, differentiating $\mathbf{x}$ and $\mathbf{y}$ by incorporating inherent noise $\delta\boldsymbol{\xi}$ into $\mathcal{D}$. If we eliminate noises $\{\delta\boldsymbol{\xi}_\mu\}_{\mu \in [M]}$ from the training memory patterns, (3.5) reduces to that of the standard *auto-associative* dense modern Hopfield model, as shown in (2.2).

With Remark 3.5, Lemma 3.1 facilitates the replication of known results from the standard dense modern Hopfield model (Ramsauer et al., 2020). The recovery of dense modern Hopfield model provides a sanity check for our nonparametric framework.

**Sparse-Structured Modern Hopfield Models.** Next, we present a set of efficient modern Hopfield models with sparse-structured patterns via the following mask.

**Definition 3.3** (Sparse-Structured Mask)**.** Let $\mathcal{M} := \{\mathcal{M}(1), \ldots, \mathcal{M}(k)\} \subseteq \{1, \ldots, M\}$ be the reduced support set for $\mathcal{T}_{\text{SVR}}$ of size $k \le M$. Then, for $\mu \in [M]$, the

optimization problem in (3.2) reduces to

$$\underset{\mathbf{W},\boldsymbol{\eta},\widetilde{\boldsymbol{\eta}}}{\text{Min}} \frac{1}{2}\|\mathbf{W}\|^2 + C\sum_{\mu\in\mathcal{M}}\langle\mathbb{1},(\boldsymbol{\eta}_\mu+\widetilde{\boldsymbol{\eta}}_\mu)\rangle \quad \text{s.t.} \quad (3.6)$$

$$\begin{cases} -(\epsilon'\mathbb{1}+\widetilde{\boldsymbol{\eta}}_\mu) \le \boldsymbol{\xi}_\mu - \langle\mathbf{W},\Phi(\boldsymbol{\xi}_\mu+\delta\boldsymbol{\xi}_\mu)\rangle \le \epsilon'\mathbb{1}+\boldsymbol{\eta}_\mu \\ \epsilon'\mathbb{1}+\boldsymbol{\eta}_\mu \le \epsilon\mathbb{1}/\sqrt{d} \\ \boldsymbol{\eta}_\mu \ge 0, \widetilde{\boldsymbol{\eta}}_\mu \ge 0, \forall\mu\in\mathcal{M}. \end{cases}$$

With Definition 3.3, we obtain the following sparse-structured retrieval dynamics (and thereby its corresponding Hopfield model(s)) by fitting $\mathcal{T}_{\text{SVR}}$ on $\mathcal{D}$ masked by $\mathcal{M}$.

**Theorem 3.2** (Sparse-Structured Modern Hopfield Models).
Let $\Phi(\cdot) = (\phi_0^{(0)}, \phi_1^{(1)}, \dots, \phi_{D_1}^{(1)}, \dots, \phi_1^{(n)}, \dots, \phi_{D_n}^{(n)}, \dots)$ with, for $1 \le D' \le D_n$ with $D_n := \binom{d+n-1}{n}$,

$$\phi_{D'}^{(n)} := \frac{(\sqrt{\beta}x_1)^{\ell_1}\cdots(\sqrt{\beta}x_d)^{\ell_d}}{\sum_{\mu\in\mathcal{M}}\langle\Phi(\boldsymbol{\xi}_\mu+\delta\boldsymbol{\xi}_\mu),\Phi(\mathbf{x})\rangle\cdot\sqrt{\ell_1!\cdots\ell_d!}},$$

where $\ell_1 + \cdots + \ell_d = n$. By Theorem 3.1, fitting $\mathcal{T}_{\text{SVR}}$ on $\mathcal{D}$ masked by $\mathcal{M}$ following (3.6) gives

$$\mathcal{T}_{\text{Sparse}}(\mathbf{x}) = \sum_{\mu\in\mathcal{M}}\underbrace{\left[\text{Softmax}(\beta\boldsymbol{\Xi}_{\mathcal{M}}^\top\mathbf{x})\right]_\mu}_{\in\mathbb{R}}\boldsymbol{\xi}_\mu, \quad (3.7)$$

where $\boldsymbol{\Xi}_{\mathcal{M}} := [\cdots, \boldsymbol{\xi}_j+\delta\boldsymbol{\xi}_j\cdots] \in \mathbb{R}^{d\times|\mathcal{M}|}$ with $j \in [|\mathcal{M}|]$.

*Proof.* See Appendix D.3 for a detailed proof. □

We emphasize that (3.7) is in fact generic and is able to describe many sparse-structured modern Hopfield models with various support sets. Importantly, it allows us to construct efficient variants with sub-quadratic complexity, and hence fills the void in the literature regarding efficient modern Hopfield models, as discussed in (Hu et al., 2023).

We present three efficient variants based on (3.7) below. To analyze efficiency for long query sequences[4], we first generalize (3.7) from a single query $\mathbf{x}$ to a sequence of $L$ query denoted by $\mathbf{X} = [\mathbf{x}_1, \dots, \mathbf{x}_L]$. Let the binary matrix $\mathbb{I}_{\mathcal{M}}$ be the corresponding sparse-sturctured mask.

**Example 1** (Random Masked Modern Hopfield Model with $\mathcal{O}(kL)$ Complexity). By setting $\mathcal{M}$ to randomly mask $(M-k)$ entries, we obtain an efficient modern Hopfield model with a sub-quadratic $\mathcal{O}(kL)$ complexity. This model connects to the random attention of BigBird (Zaheer et al., 2020).

**Example 2** (Efficient Modern Hopfield Model with $\mathcal{O}(L\sqrt{L})$ Complexity). By setting $\mathcal{M}$ for each query in a way that $\mathbb{I}_{\mathcal{M}}$ reproduces the sliding window pattern of

window size $\sqrt{L}$, we obtain an efficient modern Hopfield model with a sub-quadratic $\mathcal{O}(L\sqrt{L})$ complexity. This model connects to the Longformer attention (Beltagy et al., 2020) by design.

**Example 3** (Top-$K$ Modern Hopfield Model). Let the sequence $\{p_\mu\}_{\mu\in[M]}$ be the inner products of memories $\{\boldsymbol{\xi}_\mu\}_{\mu\in[M]}$ and query $\mathbf{x}$, i.e., $p_\mu := \langle\mathbf{x},\boldsymbol{\xi}_\mu\rangle$, and let $p^\star$ be the $K$-th largest element in $\{p_\mu\}_{\mu\in[M]}$. Then we obtain a sparse-structured mask $\mathcal{M}$ such that

$$\begin{cases} \mu\in\mathcal{M}, \text{ if } p_\mu \ge p^\star \\ \mu\notin\mathcal{M}, \text{ if } p_\mu < p^\star. \end{cases} \quad (3.8)$$

With (3.8), we arrive a top-$K$ modern Hopfield model with quadratic complexity, i.e., inefficient. This model connects to the top-$K$ attention (Gupta et al., 2021) by design.

**Remark 3.6** (Time Complexity of Modern Hopfield Models and Attention Mechanism). The time complexity of modern Hopfield models and Hopfield layers is given by:

- Time complexity of modern Hopfield model: $\mathcal{O}(Md^2)$.

- When used as cross-attention (Hopfield layer) with length-$L$ (query) and length-$M$ (memory) input sequences: $\mathcal{O}(LMd^2)$.

- When used as self-attention with length-$L$ input sequence (set $M = L$): $\mathcal{O}(n^2d^2)$.

Our efficient modern Hopfield models achieve high efficiency through two means: a sparse-structured mask and various choices of the kernel $\Phi$. The sparse-structured mask, with a support set size of $k \le M$, reduces the complexity from $\mathcal{O}(Md^2)$ to $\mathcal{O}(kd^2)$. Additionally, different choices of kernel, such as the linear kernel and positive random kernel in Appendix E, lead to efficient implementations.

Numerically, we verify their performance in Appendices G.1 to G.4 and efficiency in Appendix G.5 (e.g., duration time in Figure 6 and Floating point operations in Figure 7.)

## 4 Theoretical Analysis

In this section, we characterize how sparsity affects the sparse-structured models defined in (3.7). Our theoretical analysis on these new sparse models[5] consists of the following two aspects:

1. Derive the sparsity-dependent retrieval error bound and prove their fixed point convergence.

2. Characterize the fundamental limit of memory capacity.

---

[4]Considering long query sequences is crucial, as they contribute to inefficiency (see (Hu et al., 2023, Section C.2)).

[5]We use plural "models" as $\mathcal{M}$ in (3.7) is a generic expression for many models with different sparse patterns.

As a reminder, we adopt Definition 3.2 for memory storage and retrieval. Additionally, we recall the following definition regarding the separation between memory patterns.

**Definition 4.1** (Separation of Patterns). The separation of a memory pattern $\boldsymbol{\xi}_\mu$ from all other memory patterns $\boldsymbol{\Xi}$ is defined as its minimal inner product difference to any other patterns: $\Delta_\mu := \text{Min}_{\nu,\nu\neq\mu} \left[ \langle \boldsymbol{\xi}_\mu, \boldsymbol{\xi}_\mu \rangle - \langle \boldsymbol{\xi}_\mu, \boldsymbol{\xi}_\nu \rangle \right]$.

### 4.1 Memory Retrieval: Error Bounds and Fixed Point Convergence

**Memory Retrieval Error Bounds.** To analyze the accuracy of memory retrieval, we derive the upper bound on retrieval error of the sparse-structured models.

**Theorem 4.1** (Sparsity-Dependent Retrieval Error). Let $\mathcal{T}_{\text{Sparse}}$ be the sparse-structured retrieval dynamics (3.7). For query $\mathbf{x} \in \mathcal{S}_\mu$, it holds

$$\|\mathcal{T}_{\text{Sparse}}(\mathbf{x}) - \boldsymbol{\xi}_\mu\| \tag{4.1}$$

$$\leq m(M+k-2)\exp\left\{-\beta\big(\langle\boldsymbol{\xi}_\mu,\mathbf{x}\rangle - \underset{\nu\in[M],\nu\neq\mu}{\text{Max}}\langle\boldsymbol{\xi}_\mu,\boldsymbol{\xi}_\nu\rangle\big)\right\},$$

for all $\mu \in \mathcal{M}$, where $k := |\mathcal{M}| \in [M]$ denotes the size of the support set $\mathcal{M}$, and $m = \text{Max}_\mu \|\boldsymbol{\xi}_\mu\|$.

*Proof.* See Appendix D.4 for a detailed proof. $\square$

Interestingly, the retrieval error bound in Theorem 4.1 is sparsity-dependent, which is governed by the size of the support set $\mathcal{M}$, i.e. sparsity dimension $k := |\mathcal{M}|$.

**Remark 4.1** (Comparing with the Sparse Modern Hopfield Model (Hu et al., 2023)). Compared to the retrieval error bound in (Hu et al., 2023), which lacks explicit dependence on its input (data)-dependent sparsity, the sparsity (size of $\mathcal{M}$) here is pre-specified. When there are fewer elements in the sparse-structured mask, i.e., when $k$ is small, the retrieval error bound is tighter, and vice versa.

**Remark 4.2** (Noise Robustness). By Theorem 4.1, in cases involving contaminated query or memory, i.e. $\widetilde{\mathbf{x}} = \mathbf{x} + \boldsymbol{\delta x}$ (noise in query) or $\widetilde{\boldsymbol{\xi}} = \boldsymbol{\xi} + \boldsymbol{\delta \xi}$ (noise in memory), the impact of noise on the sparse retrieval error (4.1) is less than that its impact on the dense counterpart due to the smaller coefficient $(M+k-2)$.

**Corollary 4.1.1.** Let $\mathcal{T}_{\text{Dense}}$ and $\mathcal{T}_{\text{Sparse}}$ be the dense (3.7) and sparse-structured (3.7) retrieval dynamics, respectively. For any query pattern $\mathbf{x} \in \mathcal{S}_\mu$ and $\mu \in \mathcal{M}$, it holds

$$\|\mathcal{T}_{\text{Sparse}}(\mathbf{x}) - \boldsymbol{\xi}_\mu\| \leq \|\mathcal{T}_{\text{Dense}}(\mathbf{x}) - \boldsymbol{\xi}_\mu\|.$$

*Proof.* See Appendix D.5 for a detailed proof. $\square$

Computationally, Corollary 4.1.1 suggests that $\mathcal{T}_{\text{Sparse}}$ necessitates fewer iterations to reach fixed points compared to $\mathcal{T}_{\text{Dense}}$, given the same error tolerance level. In other words, $\mathcal{T}_{\text{Sparse}}$ retrieves stored memory patterns faster than $\mathcal{T}_{\text{Dense}}$.

**Remark 4.3** (Multiple-Update). Another important implication of Corollary 4.1.1 is that $\mathcal{T}_{\text{Sparse}}$ exhibits similar multiple-update functionality to existing models (Hu et al., 2023; Wu et al., 2024b; Ramsauer et al., 2020).

To bridge to deep learning methodologies, we show that $\mathcal{T}_{\text{Sparse}}$ retrieves memory patterns with high accuracy after a single activation in the following corollary, akin to (Hu et al., 2023; Wu et al., 2024b).

**Corollary 4.1.2** (One-Step Retrieval with High Accuracy). For any query $\mathbf{x} \in S_\mu$ and $\mu \in \mathcal{M}$, $\mathcal{T}_{\text{Sparse}}$ retrieve the memory pattern $\boldsymbol{\xi}_\mu$ with retrieval error $\epsilon$ exponentially suppressed by $\Delta_\mu$.

*Proof.* See Appendix D.5 for a detailed proof. $\square$

Corollary 4.1.2 indicates that, with sufficiently large $\Delta_\mu$, $\mathcal{T}_{\text{Sparse}}$ retrieves memory patterns in a single *iteration*, allowing the integration of sparse-structured modern Hopfield models into deep learning architectures similarly to (Xu et al., 2024; Hu et al., 2024a; Wu et al., 2024b; Schimunek et al., 2023; Hoover et al., 2023; Seidl et al., 2022).

**Fixed Point Convergence.** By design, the retrieval dynamics constructed via Lemma 3.1 satisfy (T2). We now verify this adherence as a sanity check. Interestingly, while previous studies (Hu et al., 2023; Wu et al., 2024b; Ramsauer et al., 2020) rely on the detailed energy functions to show the convergence properties of modern Hopfield models, we prove them for sparse-structured modern Hopfield models even without knowing $E$ in the next corollary. It affirms that $\mathcal{T}_{\text{Sparse}}$ satisfies (T2).

**Corollary 4.1.3** (Fixed Point Convergence). Let $\mathcal{T}_{\text{Sparse}}$ be the sparse-structured retrieval dynamics (3.7). For all $\mu \in \mathcal{M}$, the query $\mathbf{x} \in S_\mu$ converges to a fixed point if it is iteratively applied by $\mathcal{T}_{\text{Sparse}}$.

*Proof.* See Appendix D.6 for a detailed proof. $\square$

### 4.2 Memory Capacity

To characterize the fundamental limit of memory capacity, we ask the following two questions for sparse-structured modern Hopfield models following (Hu et al., 2023):

(A) What is the necessary condition for a pattern $\boldsymbol{\xi}_\mu$ being considered well-stored, and correctly retrieved?

(B) What is the expected number of memory patterns such that the above condition is satisfied?

**Well-Separation Condition.** To address (A), we identify the necessary condition for a pattern being well-stored and retrieved by the sparse-structured modern Hopfield models.

---

**Lemma 4.1** (Well-Separation Condition). Following Definition 3.2, for $\mu \in \mathcal{M}$, suppose every memory pattern $\{\boldsymbol{\xi}_\mu\}_{\mu \in \mathcal{M}}$ is enclosed by a sphere $\mathcal{S}_\mu := \{\mathbf{x} \mid \|\mathbf{x} - \boldsymbol{\xi}_\mu\| \leq R\}$, with finite radius $R := \frac{1}{2}\operatorname{Min}_{\mu,\nu \in \mathcal{M}; \mu \neq \nu}\|\boldsymbol{\xi}_\mu - \boldsymbol{\xi}_\nu\|$. Then, the retrieved dynamics $\mathcal{T}_{\text{Sparse}}$ maps $\mathcal{S}_\mu$ to itself if

1. The starting point $\mathbf{x}$ is inside $\mathcal{S}_\mu$: $\mathbf{x} \in \mathcal{S}_\mu$.

2. The *well-separation* condition:

$$\Delta_\mu \geq \frac{1}{\beta}\ln\left(\frac{(M+k-2)m}{R}\right) + 2mR.$$

---

*Proof.* See Appendix D.7 for a detailed proof. $\square$

Intuitively, the well-separation condition establishes a threshold that ensures any pattern $\{\boldsymbol{\xi}_\mu\}_{\mu \in \mathcal{M}}$ is distinguishable from all others, enabling patterns to be well-stored at a fixed point of $\mathcal{T}_{\text{Sparse}}$ and retrieved with $R$ precision by $\mathcal{T}_{\text{Sparse}}$. Notably, Lemma 4.1 reveals that the lower bound on $\Delta_\mu$ diminishes as $k$ decreases. Consequently, as $\mathcal{M}$ becomes sparser, satisfying the well-separation condition becomes easier, facilitating the storage of patterns and leading to a larger memory capacity lower bound for sparse-structured modern Hopfield models.

**Memory Capacity.** To address (B), we derive the lower bound for the maximum number of memory patterns that are well-stored and retrievable according to Lemma 4.1:

---

**Proposition 4.1** (Modified from (Hu et al., 2023)). Define the probability of storing and retrieving a memory pattern as $1 - p$. Memory capacity, the maximum number of patterns randomly sampled from a sphere with radius $m$ that the sparse modern Hopfield models can store and retrieve, has a lower bound: $M_{\text{Sparse}} \geq \sqrt{p}C^{\frac{d-1}{4}}$, where $C$ is the solution for the identity $C = {}^{b}/{W_0(\exp\{a + \ln b\})}$ with the principal branch of Lambert $W$ function, $a := \left({}^{4}/{d-1}\right)\left(\ln\left[{}^{m(\sqrt{p}+k-1)}/{R}\right] + 1\right)$ and $b := {}^{4m^2\beta}/{5(d-1)}$.

---

*Proof.* See Appendix D.8 for a detailed proof. $\square$

**Remark 4.4.** Proposition 4.1 gives a memory capacity exponential in the pattern size $d$ (maximum allowed value $k$). Since $k \leq M$, the scaling behavior of sparse-structured modern Hopfield models is similar to that of (Ramsauer et al., 2020; Hu et al., 2023). This result mirrors findings in (Wu et al., 2024b; Hu et al., 2023; Ramsauer et al., 2020).

## 5 Conclusion and Discussion

We introduce a nonparametric framework for modern Hopfield models. We use two examples to validate our framework: the original dense & the sparse-structured modern Hopfield models. With Lemma 3.1, we replicate the known results of the original modern Hopfield model (Ramsauer et al., 2020). With Theorem 3.2, we introduce the efficient sparse-structured Hopfield models with robust theoretical properties: tighter retrieval error bound (Corollary 4.1.1 & Corollary 4.1.2), stronger noise robustness (Remark 4.2) and exponential-in-$d$ capacity (Lemma 4.1 & Proposition 4.1).

**Comparing with Existing Works.** Our framework complements existing works (Hu et al., 2023; Wu et al., 2024b; Martins et al., 2023) by filling the efficiency gaps and connecting to various attentions in the following. Notably, when the size of the support set $k = M$, the results of Theorem 4.1, Lemma 4.1 and Proposition 4.1 reduce to those of the dense modern Hopfield model (Ramsauer et al., 2020).

**Extensions.** In Appendix E, we present a family of modern Hopfield models connecting to many other existing attention mechanisms, including Linear (Katharopoulos et al., 2020), Multi-Head (Vaswani et al., 2017), and Generalized Kernelizable or PRFs (Positive Random Features) (Choromanski et al., 2021) modern Hopfield models.

**Hopfield Layers and Numerical Experiments.** In line with (Hu et al., 2023; Wu et al., 2024b; Ramsauer et al., 2020), we introduce deep learning layers as competitive attention alternatives with memory-enhanced functionalities (Remark F.1), corresponding to our nonparametric modern Hopfield models (sparse-structured and above extensions) in Appendix F. Numerically, we verify their *memory retrieval* (as associative memory models) and *supervised learning* (as transformer alternatives) performance in Appendices G.1 to G.4 and efficiency in Appendix G.5.

**Accuracy-Efficiency Tradeoff.** For learning tasks conducted in Appendix G, we do not expect generally superior performance from efficient models. Ultimately, there is the provably accuracy-efficiency tradeoff (Keles et al., 2023; Deng et al., 2023) based on complexity analysis of matrix multiplication (hence, this result is transferable to modern Hopfield models (Hu et al., 2024b)). This work only provides a theoretical framework supporting the derivation of efficient variants of modern Hopfield model, with no strictly superior performance guarantee. However, we do observe that, in many cases, linear and random features modern Hopfield models deliver acceptable results.

**Limitations and Future Work.** We defer the discussion of limitations and future work to Appendix B.

## Impact Statement

By the theoretical nature of this work, we expect no negative social impacts.

## Acknowledgments

JH thanks Thomas F. Burns, Dmitry Krotov, Saul Santos, Andre Martins, Mimi Gallagher, Sara Sanchez, Dino Feng, and Andrew Chen for enlightening discussions; Weimin Wu and Jennifer Zhang for collaborations on related topics; Jaiyi Wang for assistance with numerical experiments; and the Red Maple Family for their support. The authors also thank the anonymous reviewers and program chairs for their constructive comments, especially the Area Chair of NeurIPS 2024 for pointing out key typos in appendix.

JH is partially supported by the Walter P. Murphy Fellowship. HL is partially supported by NIH R01LM1372201, AbbVie and Dolby. This research was supported in part through the computational resources and staff contributions provided for the Quest high performance computing facility at Northwestern University which is jointly supported by the Office of the Provost, the Office for Research, and Northwestern University Information Technology. The content is solely the responsibility of the authors and does not necessarily represent the official views of the funding agencies.

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

# Supplementary Material

# A    Table of Notations

Table 1. Mathematical Notations and Symbols

| Symbol | Description |
|---|---|
| $\mathbf{a}[i]$ | The $i$-th component of vector $\mathbf{a}$ |
| $\langle \mathbf{a}, \mathbf{b} \rangle$ | Inner product for vectors $\mathbf{a}, \mathbf{b} \in \mathbb{R}^d$ |
| $[I]$ | Index set $\{1, \cdots, I\}$, where $I \in \mathbb{N}^+$ |
| $\|\cdot\|$ | Spectral norm, equivalent to the $l_2$-norm when applied to a vector |
| $d$ | Dimension of patterns |
| $M$ | Number of stored memory patterns |
| $\beta$ | Scaling factor of the energy function controlling the learning dynamics. We set $\beta = 1/\sqrt{d}$ in practice |
| $\mathbf{x}$ | State/configuration/query pattern in $\mathbb{R}^d$ |
| $\mathbf{x}^\star$ | Stationary points of the Hopfield energy function |
| $\boldsymbol{\xi}$ | Memory patterns (keys) in $\mathbb{R}^d$ |
| $\delta\boldsymbol{\xi}$ | Noises in memory patterns in $\mathbb{R}^d$ |
| $\mathcal{D}$ | Training data set $\{(\boldsymbol{\xi}_\mu + \delta\boldsymbol{\xi}_\mu, \boldsymbol{\xi}_\mu)\}_{\mu \in [M]}$ |
| $\Xi$ | Shorthand for $M$ stored memory (key) patterns $\{\boldsymbol{\xi}_\mu\}_{\mu \in [M]}$ in $\mathbb{R}^{d \times M}$ |
| $\Xi_\delta$ | Shorthand for $M$ contaminated memory (key) patterns $\{\delta\boldsymbol{\xi}_\mu\}_{\mu \in [M]}$ in $\mathbb{R}^{d \times M}$ |
| $\Xi^\mathsf{T}\mathbf{x}$ | $M$-dimensional overlap vector $(\langle \boldsymbol{\xi}_1, \mathbf{x} \rangle, \cdots, \langle \boldsymbol{\xi}_\mu, \mathbf{x} \rangle, \cdots, \langle \boldsymbol{\xi}_M, \mathbf{x} \rangle)$ in $\mathbb{R}^M$ |
| $\Phi(\cdot)$ | Kernelized feature mapping $\Phi(\cdot) : \mathbb{R}^d \to D_\phi$ |
| $\phi$ | Element in the $\Phi(\cdot) = (\phi_0^{(0)}, \phi_1^{(1)}, \ldots, \phi_{D_1}^{(1)}, \ldots, \phi_1^{(n)}, \ldots, \phi_{D_n}^{(n)}, \ldots)$ |
| $D_\Phi$ | Dimension of the kernel space, i.e., dimension of output of $\Phi(\cdot)$ |
| $h(\cdot)$ | Normalization mapping in the regression model defined by (3.1) |
| $\mathbf{W}$ | Weighted matrix in the regression model defined by (3.1) in $\mathbb{R}^{d \times D_\Phi}$ |
| $\mathbf{w}_i$ | $i$-th row of the weighted matrix $\mathbf{W}$ in $\mathbb{R}^{D_\Phi}$ |
| $\mathcal{K}(\cdot, \cdot)$ | Kernel function takes the inner product form $\mathcal{K}(\cdot, \cdot) = \langle \Phi(\cdot), \Phi(\cdot) \rangle$ in $\mathcal{K} : \mathbb{R}^{D_\Phi} \times \mathbb{R}^{D_\Phi} \to \mathbb{R}_+$ |
| $\epsilon'$ | Component-wise term error margin in the support vector regression problem |
| $\boldsymbol{\eta}, \widetilde{\boldsymbol{\eta}}$ | Slack variables in the support vector regression |
| $C$ | Penalized coefficient of the support vector regression |
| $\mathcal{L}$ | Lagrangian corresponding to (3.2) |
| $\boldsymbol{\alpha}, \widetilde{\boldsymbol{\alpha}}, \boldsymbol{\lambda}, \widetilde{\boldsymbol{\lambda}}$ | Dual variables in the Lagrangian $\mathcal{L}$ |
| $\mathcal{M}$ | Reduced support set for $\mathcal{T}_{\text{SVR}}$ $\mathcal{M} := \{\mathcal{M}(1), \ldots, \mathcal{M}(k)\} \subseteq \{1, \ldots, M\}$ |
| $\mathbb{1}_{\mathcal{M}(\mu)}$ | Indicator function corresponding to $\mathcal{M}$, where $\mathbb{1}_{\mathcal{M}(\mu)} = 1$ for $\mu \in \mathcal{M}$ and $\mathbb{1}_{\mathcal{M}(\mu)} = 0$ for $\mu \notin \mathcal{M}$ |
| $k$ | Size of the support set $\mathcal{M}$, defined as $k := |\mathcal{M}|$ |
| $m$ | Largest norm of memory patterns, denoted as $m := \text{Max}_{\mu \in [M]} \|\boldsymbol{\xi}_\mu\|$ |
| $R$ | Minimal Euclidean distance across all possible pairs of memory patterns, denoted as $R := \frac{1}{2} \text{Min}_{\mu,\nu \in [M]} \|\boldsymbol{\xi}_\mu - \boldsymbol{\xi}_\nu\|$ |
| $S_\mu$ | Sphere centered at memory pattern $\boldsymbol{\xi}_\mu$ with finite radius $R$ |
| $\mathbf{x}_\mu^\star$ | Fixed point of $\mathcal{T}$ covered by $S_\mu$, i.e., $\mathbf{x}_\mu^\star \in S_\mu$ |
| $\Delta_\mu$ | Separation of a memory pattern $\boldsymbol{\xi}_\mu$ from all other memory patterns $\Xi$, defined in (4.1) |
| $\widetilde{\Delta}_\mu$ | Separation of $\boldsymbol{\xi}_\mu$ at a given $\mathbf{x}$ from all memory patterns $\Xi$, defined in (4.1) |

# B    Limitations and Future Work

By the theoretical nature of this work, we rely on certain simplifying assumptions. These assumptions limit the generality of our results.

We require a specific feature mapping $\Phi$ to guarantee fixed-point convergence (Section 3.2). This requirement imposes structure on the retrieval dynamics. Without it, an unconstrained nonparametric Hopfield update can converge to a "generalized" fixed point, not the intended memory (Remark 3.3). Then, multiple iterations may fail to recover the true stored pattern. We emphasize that meeting this requirement is easy (Section 3.2); see also the nonparametric modern Hopfield model family in Appendix E.

Our relative error analysis (Theorem 4.1) assumes that the correct memory item is in the chosen support set. If a random-masking step removes that item, retrieval fails (Figure 2). We remark that — while this assumption is restrictive, it is necessary for tractable analysis.

Our exponential capacity result in Proposition 4.1 needs a strong "well-separation" condition. Each pattern must be distinct enough from the others. Many real datasets have correlated or structured patterns, so this assumption may be hard to satisfy. Still, it is standard in modern Hopfield model literature (Krotov & Hopfield, 2016; Demircigil et al., 2017; Ramsauer et al., 2020; Iatropoulos et al., 2022; Hu et al., 2023; Wu et al., 2024b; Santos et al., 2024a;b). To mitigate this in practice, (Wu et al., 2024a; Hu et al., 2024c) relax this data dependency by optimizing the Hopfield-energy landscape for larger memory capacity.

Lastly, we do not analyze the extensions introduced in Appendix E. We do not prove their convergence or capacity.

Looking ahead, we plan to address these theoretical gaps and to relax strict assumptions like well-separated patterns.

# C   Supplementary Theoretical Backgrounds

## C.1   Soft-Margin Support Vector Regression

Soft-margin Support Vector Regression (SVR) (Awad et al., 2015; Jaggi, 2014; Vapnik, 2013; Kar & Karnick, 2012; Schölkopf & Smola, 2002) generalizes Support Vector Machines (SVM) to regression tasks. It finds a function

$$f(\mathbf{x}) = \mathbf{W}\,\Phi(\mathbf{x}) + \mathbf{b}$$

that remains within an $\epsilon'$-tube of the target outputs, while allowing limited violations (soft margin) when data points lie outside this tube.

**Notation and Setup.**

- $\{(\mathbf{x}_\mu, \mathbf{y}_\mu)\}_{\mu=1}^M$ is the training set, where $\mathbf{x}_\mu \in \mathbb{R}^d$ and $\mathbf{y}_\mu \in \mathbb{R}^d$.

- $\Phi : \mathbb{R}^d \to \mathbb{R}^{D_\Phi}$ is a feature map into a (possibly high-dimensional) space.

- $\mathbf{W} \in \mathbb{R}^{d \times D_\Phi}$ and $\mathbf{b} \in \mathbb{R}^d$ are the regression parameters.

**Primal Formulation.**   To tolerate errors beyond $\epsilon'$ while penalizing them, SVR introduces nonnegative slack variables $\boldsymbol{\eta}_\mu$ and $\widetilde{\boldsymbol{\eta}}_\mu$ for each data point. The soft-margin SVR with $\ell_2$-loss is formulated as

$$\min_{\mathbf{W}, \boldsymbol{\eta}, \widetilde{\boldsymbol{\eta}}} \frac{1}{2}\|\mathbf{W}\|^2 + C \sum_{\mu=1}^M \langle \mathbb{1}, (\boldsymbol{\eta}_\mu + \widetilde{\boldsymbol{\eta}}_\mu) \rangle \quad \text{subject to} \quad \begin{cases} \mathbf{y}_\mu - \langle \mathbf{W}, \Phi(\mathbf{x}_\mu) \rangle - \mathbf{b} \leq \epsilon'\mathbb{1} + \boldsymbol{\eta}, \\ \langle \mathbf{W}, \Phi(\mathbf{x}_\mu) \rangle + \mathbf{b} - \mathbf{y}_\mu \leq \epsilon'\mathbb{1} + \widetilde{\boldsymbol{\eta}}, \\ \boldsymbol{\eta}_\mu, \widetilde{\boldsymbol{\eta}}_\mu \geq 0, \quad \mu \in [M], \end{cases} \quad \text{(C.1)}$$

where $C > 0$ controls the trade-off between model fidelity and penalty on large deviations $(\boldsymbol{\eta}_\mu, \widetilde{\boldsymbol{\eta}}_\mu)$. Since (C.1) is strongly convex, it admits a unique global minimizer.

This formulation follows the standard SVR derivation; see (Schölkopf & Smola, 2002) for a comprehensive treatment.

**Lagrangian and Dual Problem.**   To solve (C.1), we form the Lagrangian:

$$\mathcal{L} = \frac{1}{2} \sum_{i=1}^d \|\mathbf{w}_i\|^2 + C \sum_{\mu=1}^M \sum_{i=1}^d \left[ \boldsymbol{\lambda}_\mu[i]\,\boldsymbol{\eta}_\mu[i] + \widetilde{\boldsymbol{\lambda}}_\mu[i]\,\widetilde{\boldsymbol{\eta}}_\mu[i] \right]$$
$$- \sum_{\mu=1}^M \sum_{i=1}^d \left[ \boldsymbol{\alpha}_\mu[i]\big(\epsilon' + \boldsymbol{\eta}_\mu[i] - \mathbf{y}_\mu[i] + \langle \mathbf{w}_i, \Phi(\mathbf{x}_\mu) \rangle + \mathbf{b}[i]\big) + \widetilde{\boldsymbol{\alpha}}_\mu[i]\big(\epsilon' + \widetilde{\boldsymbol{\eta}}_\mu[i] - \langle \mathbf{w}_i, \Phi(\mathbf{x}_\mu) \rangle - \mathbf{b}[i] + \mathbf{y}_\mu[i]\big) \right],$$

where $\boldsymbol{\lambda}_\mu, \widetilde{\boldsymbol{\lambda}}_\mu$ are multipliers for the slack constraints $\boldsymbol{\eta}_\mu, \widetilde{\boldsymbol{\eta}}_\mu \geq 0$, and $\boldsymbol{\alpha}_\mu, \widetilde{\boldsymbol{\alpha}}_\mu$ are multipliers for the $\epsilon'$-tube constraints. By applying the Karush-Kuhn-Tucker (KKT) conditions to $\mathcal{L}$, one obtains the dual problem. In practical kernelized SVR, one uses a kernel $K(\mathbf{x}_\mu, \mathbf{x}_\nu) = \langle \Phi(\mathbf{x}_\mu), \Phi(\mathbf{x}_\nu) \rangle$, which may bypass explicit construction of $\Phi$.

**Summary.**   Soft-margin SVR balances a tight $\epsilon'$-tube fit with penalty-based tolerance for outliers. Strong convexity guarantees a unique solution in the primal, while the dual formulation reveals a sparse, data-driven representation in terms of support vectors. This dependence on the training data for model complexity classifies SVR as a nonparametric method.

**Remark C.1** (Nonparametric Nature of SVR).   Although the primal objective in (C.1) involves a fixed matrix $\mathbf{W}$, the dual solution is data-dependent. In particular, the regressor can be written as

$$f(\mathbf{x}) = \sum_{\mu=1}^M (\alpha_\mu - \widetilde{\alpha}_\mu)\,\langle \Phi(\mathbf{x}_\mu),\,\Phi(\mathbf{x}) \rangle + b,$$

where only the points with nonzero $(\boldsymbol{\alpha}_\mu - \widetilde{\boldsymbol{\alpha}}_\mu)$ (the *support vectors*) affect $f$. The number of such points can grow with $M$, making SVR a nonparametric method whose capacity adapts to the size of the training data.

# D    Proofs of Main Text

## D.1    Theorem 3.1

*Proof of Theorem 3.1.* The Lagrangian of convex optimization problem defined in (3.2) is

$$
\begin{aligned}
\mathcal{L} :=& \frac{1}{2} \sum_{i=1}^{d} \|\mathbf{w}_i\|^2 + C \sum_{\mu=1}^{M} \sum_{i=1}^{d} (\boldsymbol{\lambda}_\mu[i]\boldsymbol{\eta}_\mu[i] + \widetilde{\boldsymbol{\lambda}}_\mu[i]\widetilde{\boldsymbol{\eta}}_\mu[i]) \\
& - \sum_{\mu=1}^{M} \sum_{i=1}^{d} \boldsymbol{\alpha}_\mu[i] \left( \epsilon' + \boldsymbol{\eta}_\mu[i] - \boldsymbol{\xi}_\mu[i] + \langle \mathbf{w}_i, \Phi(\boldsymbol{\xi}_\mu + \delta\boldsymbol{\xi}_\mu) \rangle \right) \\
& - \sum_{\mu=1}^{M} \sum_{i=1}^{d} \widetilde{\boldsymbol{\alpha}}_\mu[i] \left( \epsilon' + \widetilde{\boldsymbol{\eta}}_\mu[i] - \langle \mathbf{w}_i, \Phi(\boldsymbol{\xi}_\mu + \delta\boldsymbol{\xi}_\mu) \rangle + \boldsymbol{\xi}_\mu[i] \right),
\end{aligned} \tag{D.1}
$$

where $\boldsymbol{\lambda}_\mu[i], \widetilde{\boldsymbol{\lambda}}_\mu[i], \boldsymbol{\alpha}_\mu[i]$ and $\widetilde{\boldsymbol{\alpha}}_\mu[i]$ are Lagrange multipliers. Next, we solve stationary condition with respect to $\mathbf{w}_i, \boldsymbol{\eta}_\mu[i]$ and $\widetilde{\boldsymbol{\eta}}_\mu[i]$ from above Lagrangian and derive corresponding optimal solution. The Lagrangian in (D.1) admits a stationary solution, which is given by:

$$
\begin{cases}
\mathbf{w}_i - \sum_{\mu=1}^{M} \left( \boldsymbol{\alpha}_\mu[i] - \widetilde{\boldsymbol{\alpha}}_\mu[i] \right) \Phi(\boldsymbol{\xi}_\mu + \delta\boldsymbol{\xi}_\mu) = 0, \\
C - \boldsymbol{\lambda}_\mu[i] - \boldsymbol{\alpha}_\mu[i] = 0, \\
C - \widetilde{\boldsymbol{\lambda}}_\mu[i] - \widetilde{\boldsymbol{\alpha}}_\mu[i] = 0.
\end{cases} \tag{D.2}
$$

Substitute (D.2) into (3.1) to write

$$
\mathbf{x}^{\text{new}}[i] = \mathcal{T}_{\text{SVR}}(\mathbf{x})[i] := \langle \mathbf{w}_i^\star, \Phi(\mathbf{x}) \rangle,
$$

with the learned weight matrix

$$
\mathbf{w}_i^\star := \sum_{\mu=1}^{M} \underbrace{\left( \boldsymbol{\alpha}_\mu[i] - \widetilde{\boldsymbol{\alpha}}_\mu[i] \right)}_{\in \mathbb{R}} \underbrace{\Phi(\boldsymbol{\xi}_\mu + \delta\boldsymbol{\xi}_\mu)}_{\in \mathbb{R}^{D_\Phi}} \in \mathbb{R}^{D_\Phi}.
$$

The complementary slackness condition and dual feasibility of (D.1) are given by

$$
\begin{cases}
\boldsymbol{\alpha}_\mu[i] \left( \epsilon' + \boldsymbol{\eta}_\mu[i] - \boldsymbol{\xi}_\mu[i] + \langle \mathbf{w}_i, \Phi(\boldsymbol{\xi}_\mu + \delta\boldsymbol{\xi}_\mu) \rangle \right) = 0 \\
\widetilde{\boldsymbol{\alpha}}_\mu[i] \left( \epsilon' + \widetilde{\boldsymbol{\eta}}_\mu[i] - \langle \mathbf{w}_i, \Phi(\boldsymbol{\xi}_\mu + \delta\boldsymbol{\xi}_\mu) \rangle + \boldsymbol{\xi}_\mu[i] \right) = 0 \\
\boldsymbol{\alpha}_\mu[i], \widetilde{\boldsymbol{\alpha}}_\mu[i], \boldsymbol{\lambda}_\mu[i], \widetilde{\boldsymbol{\lambda}}_\mu[i] \geq 0,
\end{cases} \tag{D.3}
$$

for all $\mu \in [M]$ and $i \in [d]$.

This completes the proof. □

## D.2    Lemma 3.1

To simplify our proofs, we define

$$
\Phi(\mathbf{x}) := \frac{\overline{\Phi}(\mathbf{x})}{h(\mathbf{x})}, \tag{D.4}
$$

where $h(\cdot) : \mathbb{R}^d \to \mathbb{R}$ is some normalization function for later convenience.

To prove Lemma 3.1, we introduce the following three auxiliary lemmas.

**Lemma D.1.** Let $\boldsymbol{\alpha}_\mu[i] \geq 0$, $\widetilde{\boldsymbol{\alpha}}_\mu[i] \geq 0$ be a solution to (D.2) with KKT conditions (D.3). Then $\boldsymbol{\alpha}_\mu[i] - \widetilde{\boldsymbol{\alpha}}_\mu[i]$ has the following bounds

$$-C \leq \boldsymbol{\alpha}_\mu[i] - \widetilde{\boldsymbol{\alpha}}_\mu[i] \leq C, \quad \forall \mu \in [M], i \in [d].$$

*Proof of Lemma D.1.* We prove this lemma by contradiction. Recall that for each fixed values of $\mu$ and $i$

$$\boldsymbol{\alpha}_\mu[i] \geq 0, \widetilde{\boldsymbol{\alpha}}_\mu[i] \geq 0.$$

Firstly, we assume $\boldsymbol{\alpha}_\mu[i], \widetilde{\boldsymbol{\alpha}}_\mu[i] \in \mathbb{R}_+$ (*non-zero*), for all $\mu \in [M]$ and $i \in [d]$. Recall complementary slackness conditions from (D.3)

$$\begin{cases} \epsilon' + \boldsymbol{\eta}_\mu[i] - \boldsymbol{\xi}_\mu[i] + \langle \mathbf{w}_i, \Phi(\boldsymbol{\xi}_\mu + \delta\boldsymbol{\xi}_\mu) \rangle = 0 \\ \epsilon' + \widetilde{\boldsymbol{\eta}}_\mu[i] - \langle \mathbf{w}_i, \Phi(\boldsymbol{\xi}_\mu + \delta\boldsymbol{\xi}_\mu) \rangle + \boldsymbol{\xi}_\mu[i] = 0. \end{cases}$$

Combine above two equations to write

$$\boldsymbol{\eta}_\mu[i] + \widetilde{\boldsymbol{\eta}}_\mu[i] = -2\epsilon' \leq 0.$$

Since the component-wise error $\epsilon' \geq 0$, we have $\boldsymbol{\eta}_\mu[i] + \widetilde{\boldsymbol{\eta}}_\mu[i] \leq 0$. This conclusion contradicts the assumption of the non-negative condition on slack variables $\boldsymbol{\eta}_\mu[i], \widetilde{\boldsymbol{\eta}}_\mu[i] \geq 0$. Therefore, together with (D.2), at least one of $\boldsymbol{\alpha}_\mu[i], \widetilde{\boldsymbol{\alpha}}_\mu[i]$ must be 0, for all $\mu$ and all $i$. Subsequently, we have

$$0 \leq \boldsymbol{\alpha}_\mu[i] \leq C \quad \text{and} \quad 0 \leq \widetilde{\boldsymbol{\alpha}}_\mu[i] \leq C,$$

which leads to

$$-C \leq \boldsymbol{\alpha}_\mu[i] - \widetilde{\boldsymbol{\alpha}}_\mu[i] \leq C.$$

This completes the proof. $\qquad\square$

**Lemma D.2** (Multinomial Expansion). Given $\mathbf{x}, \mathbf{y} \in \mathbb{R}^d$. The identity

$$\frac{(\mathbf{x}^\mathsf{T}\mathbf{y})^n}{n!} = \sum_{\ell_1 + \cdots + \ell_d = n} \left( \frac{x_1^{\ell_1} \cdots x_d^{\ell_d}}{\sqrt{\ell_1! \cdots \ell_d!}} \right) \left( \frac{y_1^{\ell_1} \cdots y_d^{\ell_d}}{\sqrt{\ell_1! \cdots \ell_d!}} \right),$$

holds for all $n \in \mathbb{N}$.

*Proof.*

$$\begin{aligned}
\frac{(\mathbf{x}^\mathsf{T}\mathbf{y})^n}{n!} &= \frac{1}{n!}(x_1 y_1 + \cdots + x_d y_d)^n \\
&= \frac{1}{n!} \left[ (x_1 y_1)^n + \cdots + \frac{n!}{\ell_1! \cdots \ell_d!} \prod_{i=1}^d (x_i y_i)^{\ell_i} + \cdots + (x_d y_d)^n \right] && \left( \textstyle\sum_{i=1}^d \ell_i = n \right) \\
&= \sum_{\ell_1 + \cdots + \ell_d = n} \frac{1}{\ell_1! \cdots \ell_d!} \prod_{i=1}^d (x_i)^{\ell_i} \prod_{i=1}^d (y_i)^{\ell_i} \\
&= \sum_{\ell_1 + \cdots + \ell_d = n} \frac{\left( x_1^{\ell_1} \cdots x_d^{\ell_d} \right) \left( y_1^{\ell_1} \cdots y_d^{\ell_d} \right)}{\ell_1! \cdots \ell_d!} \\
&= \sum_{\ell_1 + \cdots + \ell_d = n} \left( \frac{x_1^{\ell_1} \cdots x_d^{\ell_d}}{\sqrt{\ell_1! \cdots \ell_d!}} \right) \left( \frac{y_1^{\ell_1} \cdots y_d^{\ell_d}}{\sqrt{\ell_1! \cdots \ell_d!}} \right).
\end{aligned}$$

This completes the proof. $\qquad\square$

Our next lemma restates a known result that the exponential dot-product kernel admits an infinite-dimensional feature expansion via its power series (Nguyen et al., 2024; Hamid et al., 2014; Kar & Karnick, 2012; Schölkopf & Smola, 2002). We include the derivation here for completeness.

**Lemma D.3** (Closed-Form Exponential Dot-Product Kernel). Let $\mathcal{K}(\cdot, \cdot)$ be the exponential dot-product kernel:

$$\mathcal{K}(\mathbf{x}, \mathbf{y}) := \exp\{\langle \mathbf{x}, \mathbf{y} \rangle\} = \langle \overline{\Phi}(\mathbf{x}), \overline{\Phi}(\mathbf{y}) \rangle = \sum_{n=0}^{\infty} \frac{(\mathbf{x}^\mathsf{T} \mathbf{y})^n}{n!},$$

where $\mathbf{x}, \mathbf{y} \in \mathbb{R}^d$ and $\Phi$ maps the feature vectors $\mathbf{x}$ and $\mathbf{y}$ into infinite dimensional space. Then,

$$\overline{\Phi}(\cdot) = (\bar{\phi}_0^{(0)}, \bar{\phi}_1^{(1)}, \dots, \bar{\phi}_{D_1}^{(1)}, \dots, \bar{\phi}_1^{(n)}, \dots, \bar{\phi}_{D_n}^{(n)}, \dots),$$

has a closed form solution

$$\bar{\phi}_{D'}^{(n)} = \frac{x_1^{\ell_1} \cdots x_d^{\ell_d}}{\sqrt{\ell_1! \cdots \ell_d!}},$$

where $\ell_1 + \cdots + \ell_d = n$, $1 \leq D' \leq D_n$ and $D_n := \binom{d+n-1}{n}$.

*Proof of Lemma D.3.* Applying Lemma D.2 on the exp kernel, we have

$$\langle \overline{\Phi}(\mathbf{x}), \overline{\Phi}(\mathbf{y}) \rangle = \sum_{n=0}^{\infty} \frac{(\mathbf{x}^\mathsf{T} \mathbf{y})^n}{n!}$$

$$= \sum_{n=0}^{\infty} \sum_{\ell_1 + \cdots + \ell_d = n} \frac{1}{\ell_1! \cdots \ell_d!} \prod_{i=1}^{d} (x_i)^{\ell_i} \prod_{i=1}^{d} (y_i)^{\ell_i}$$

$$= \sum_{n=0}^{\infty} \sum_{\ell_1 + \cdots + \ell_d = n} \frac{\left( x_1^{\ell_1} \cdots x_d^{\ell_d} \right) \left( y_1^{\ell_1} \cdots y_d^{\ell_d} \right)}{\ell_1! \cdots \ell_d!}$$

$$= \sum_{n=0}^{\infty} \sum_{\ell_1 + \cdots + \ell_d = n} \left( \frac{x_1^{\ell_1} \cdots x_d^{\ell_d}}{\sqrt{\ell_1! \cdots \ell_d!}} \right) \left( \frac{y_1^{\ell_1} \cdots y_d^{\ell_d}}{\sqrt{\ell_1! \cdots \ell_d!}} \right).$$

From above, we observe that, for each fixed $n$, there are $\binom{d+n-1}{n}$ terms in the summation. Consequently, $\overline{\Phi}(\mathbf{x})$ has a solution

$$\overline{\Phi}(\mathbf{x}) = (\bar{\phi}_0^{(0)}, \underbrace{\bar{\phi}_1^{(1)}, \dots, \bar{\phi}_{D_1}^{(1)}}_{\binom{d+1-1}{1} \text{ elements}}, \dots, \underbrace{\bar{\phi}_1^{(n)}, \dots, \bar{\phi}_{D_n}^{(n)}}_{\binom{d+n-1}{n} \text{ elements}}, \dots),$$

where $D_n = \binom{d+n-1}{n}$ and

$$\bar{\phi}_{D'}^{(n)} = \frac{x_1^{\ell_1} \cdots x_d^{\ell_d}}{\sqrt{\ell_1! \cdots \ell_d!}},$$

for $1 \leq D' \leq D_n$ and $\ell_1 + \cdots + \ell_d = n$. This completes the proof. $\square$

*Proof of Lemma 3.1.* Recall that the learned weight matrix $\mathbf{W}$ is composed of

$$\mathbf{w}_i^\star = \sum_{\mu=1}^{M} (\boldsymbol{\alpha}_\mu[i] - \widetilde{\boldsymbol{\alpha}}_\mu[i]) \frac{\overline{\Phi}(\boldsymbol{\xi}_\mu + \delta\boldsymbol{\xi}_\mu)}{h(\mathbf{x})}.$$

Substitute $\mathbf{w}^\star$ into (3.1) to write

$$\mathcal{T}_{\text{Dense}}(\mathbf{x}) = \left( \sum_{\mu=1}^{M} \frac{\alpha_\mu[1] - \widetilde{\alpha}_\mu[1]}{h(\boldsymbol{\xi}_\mu + \delta\boldsymbol{\xi}_\mu)} \frac{\langle \overline{\Phi}(\boldsymbol{\xi}_\mu + \delta\boldsymbol{\xi}_\mu), \overline{\Phi}(\mathbf{x}) \rangle}{h(\mathbf{x})}, \dots, \sum_{\mu=1}^{M} \frac{\alpha_\mu[d] - \widetilde{\alpha}_\mu[d]}{h(\boldsymbol{\xi}_\mu + \delta\boldsymbol{\xi}_\mu)} \frac{\langle \overline{\Phi}(\boldsymbol{\xi}_\mu + \delta\boldsymbol{\xi}_\mu), \overline{\Phi}(\mathbf{x}) \rangle}{h(\mathbf{x})} \right).$$

Let $\boldsymbol{\xi}_\mu := \left( \frac{\alpha_\mu[1] - \widetilde{\alpha}_\mu[1]}{h(\boldsymbol{\xi}_\mu + \delta\boldsymbol{\xi}_\mu)}, \dots, \frac{\alpha_\mu[d] - \widetilde{\alpha}_\mu[d]}{h(\boldsymbol{\xi}_\mu + \delta\boldsymbol{\xi}_\mu)} \right)$ and $h(\mathbf{x}) := \sum_{\mu=1}^M \langle \overline{\Phi}(\boldsymbol{\xi}_\nu + \delta\boldsymbol{\xi}_\nu), \overline{\Phi}(\mathbf{x}) \rangle$. Then $\mathcal{T}_{\text{Dense}}$ reduces to

$$\mathcal{T}_{\text{Dense}}(\mathbf{x}) = \sum_{\mu=1}^M \frac{\langle \overline{\Phi}(\boldsymbol{\xi}_\mu + \delta\boldsymbol{\xi}_\mu), \overline{\Phi}(\mathbf{x}) \rangle}{\sum_{\nu=1}^M \langle \overline{\Phi}(\boldsymbol{\xi}_\nu + \delta\boldsymbol{\xi}_\nu), \overline{\Phi}(\mathbf{x}) \rangle} \boldsymbol{\xi}_\mu.$$

Following Lemma D.3, here we define the inner product of $\overline{\Phi}$ as a kernel $\mathcal{K} : \mathbb{R}^{D_\phi} \times \mathbb{R}^{D_\phi} \to \mathbb{R}_+$

$$\langle \overline{\Phi}(\mathbf{x}), \overline{\Phi}(\boldsymbol{\xi}_\mu + \delta\boldsymbol{\xi}_\mu) \rangle := \mathcal{K}(\mathbf{x}, \boldsymbol{\xi}_\mu + \delta\boldsymbol{\xi}_\mu).$$

$\mathcal{T}_{\text{Dense}}$ is now given by

$$\mathcal{T}_{\text{Dense}}(\mathbf{x}) = \sum_{\mu=1}^M \frac{\mathcal{K}(\mathbf{x}, \boldsymbol{\xi}_\mu + \delta\boldsymbol{\xi}_\mu)}{\sum_{\nu=1}^M \mathcal{K}(\mathbf{x}, \boldsymbol{\xi}_\nu + \delta\boldsymbol{\xi}_\nu)} \boldsymbol{\xi}_\mu. \tag{D.5}$$

Observe that (2.2) $\mathcal{T}_{\text{Dense}}$ takes a Boltzmann form: $\exp\{\cdot\} / \sum_{\nu=1}^M \exp\{\cdot\}$. By Lemma D.3, we take

$$\overline{\phi}_{D'}^{(n)} = \frac{(\sqrt{\beta}x_1)^{\ell_1} \cdots (\sqrt{\beta}x_d)^{\ell_d}}{\sqrt{\ell_1! \cdots \ell_d!}},$$

with the kernel

$$\mathcal{K}(\mathbf{x}, \boldsymbol{\xi}_\mu + \delta\boldsymbol{\xi}_\mu) = \sum_{n=0}^\infty \frac{(\langle \sqrt{\beta}\mathbf{x}, \sqrt{\beta}\boldsymbol{\xi}_\mu + \sqrt{\beta}\delta\boldsymbol{\xi}_\mu \rangle)^n}{n!}. \tag{D.6}$$

Substitute (D.6) into (D.5) and write

$$\mathcal{T}_{\text{Dense}}(\mathbf{x}) = \sum_{\mu=1}^M \frac{\sum_{n=0}^\infty \left( \langle \sqrt{\beta}\mathbf{x}, \sqrt{\beta}\boldsymbol{\xi}_\mu + \sqrt{\beta}\delta\boldsymbol{\xi}_\mu \rangle \right)^n / n!}{\sum_{\nu=1}^M \sum_{t=0}^\infty \left( \langle \sqrt{\beta}\mathbf{x}, \sqrt{\beta}\boldsymbol{\xi}_\nu + \sqrt{\beta}\delta\boldsymbol{\xi}_\nu \rangle \right)^t / t!} \boldsymbol{\xi}_\mu.$$

By Taylor's theorem, $\mathcal{T}_{\text{Dense}}$ takes the form

$$\mathcal{T}_{\text{Dense}}(\mathbf{x}) = \sum_{\mu=1}^M \frac{\exp\{\beta \langle \mathbf{x}, \boldsymbol{\xi}_\mu + \delta\boldsymbol{\xi}_\mu \rangle\}}{\sum_{\nu=1}^M \exp\{\beta \langle \mathbf{x}, \boldsymbol{\xi}_\nu + \delta\boldsymbol{\xi}_\nu \rangle\}} \boldsymbol{\xi}_\mu = \Xi \, \text{Softmax} \left( \beta \Xi_\delta^\top \mathbf{x} \right), \tag{D.7}$$

where $\Xi = (\boldsymbol{\xi}_1, \cdots, \boldsymbol{\xi}_M) \in \mathbb{R}^{d \times M}$ and $\Xi_\delta = (\boldsymbol{\xi}_1 + \delta\boldsymbol{\xi}_1, \cdots, \boldsymbol{\xi}_M + \delta\boldsymbol{\xi}_M) \in \mathbb{R}^{d \times M}$ denote memories and noises in memories, respectively. This completes the proof. $\qquad\square$

### D.3    Theorem 3.2

*Proof of Theorem 3.2.* To take $\mathbf{w}^\star$ for the sparse-structured model, the partial derivatives of $\mathcal{L}$ with respect to $\mathbf{w}_i, \boldsymbol{\eta}_\mu[i]$ and $\widetilde{\boldsymbol{\eta}}_\mu[i]$ must satisfy the stationarity condition

$$\begin{cases} \mathbf{w}_i - \sum_{\mu \in \mathcal{M}} (\boldsymbol{\alpha}_\mu[i] - \widetilde{\boldsymbol{\alpha}}_\mu[i]) \Phi(\boldsymbol{\xi}_\mu + \delta\boldsymbol{\xi}_\mu) = 0, \\ C - \boldsymbol{\lambda}_\mu[i] - \boldsymbol{\alpha}_\mu[i] = 0, \\ C - \widetilde{\boldsymbol{\lambda}}_\mu[i] - \widetilde{\boldsymbol{\alpha}}_\mu[i] = 0. \end{cases}$$

Then, we arrive

$$\mathbf{w}_i^\star = \sum_{\mu \in \mathcal{M}} (\boldsymbol{\alpha}_\mu[i] - \widetilde{\boldsymbol{\alpha}}_\mu[i]) \frac{\overline{\Phi}(\boldsymbol{\xi}_\mu + \delta\boldsymbol{\xi}_\mu)}{h(\mathbf{x})}.$$

Following a similar approach as in Appendix D.2, we derive the retrieval dynamics for the sparse-structured modern Hopfield model:

$$\mathcal{T}_{\text{Sparse}}(\mathbf{x}) = \sum_{\mu \in \mathcal{M}} \left[ \text{Softmax}(\beta \Xi_\mathcal{M}^\top \mathbf{x}) \right]_\mu \boldsymbol{\xi}_\mu,$$

where the softmax is computed over $\beta \Xi_\mathcal{M}^\top \mathbf{x}$ with $\Xi_\mathcal{M} := \underbrace{[\cdots, \boldsymbol{\xi}_j + \delta\boldsymbol{\xi}_j \cdots]}_{j \in [|\mathcal{M}|]}$. This completes the proof. $\qquad\square$

### D.4  Theorem 4.1

*Proof of Theorem 4.1.* To connect $\mathcal{T}_{\text{Sparse}}$ with $\Delta_\mu$, first we derive the bound on $\|\mathcal{T}_{\text{Sparse}}(\mathbf{x}) - \boldsymbol{\xi}_\mu\|$ via (Ramsauer et al., 2020) for $\mu \in \mathcal{M}$

$$
\begin{aligned}
\|\mathcal{T}_{\text{Sparse}}(\mathbf{x}) - \boldsymbol{\xi}_\mu\| &\leq \left\| \boldsymbol{\xi}_\mu - \sum_{\nu \in \mathcal{M}} \left[ \text{Softmax}\left(\beta \Xi_\delta^\mathsf{T} \mathbf{x}\right) \right]_\nu \boldsymbol{\xi}_\nu \right\| \\
&\leq \left\| (1 - [\text{Softmax}\left(\beta \Xi_\delta^\mathsf{T} \mathbf{x}\right)]_\mu)\boldsymbol{\xi}_\mu + \sum_{\nu \in \mathcal{M}, \nu \neq \mathcal{M}} [\text{Softmax}\left(\beta \Xi_\delta^\mathsf{T} \mathbf{x}\right)]_\nu \boldsymbol{\xi}_\nu \right\| \\
&\leq \widetilde{\epsilon}\|\boldsymbol{\xi}_\mu\| + \frac{\widetilde{\epsilon}}{M-1} \sum_{\nu \in \mathcal{M}, \nu \neq \mu} \|\boldsymbol{\xi}_\nu\| \\
&\leq \widetilde{\epsilon}m + \frac{\widetilde{\epsilon}}{M-1}(k-1)m \\
&\leq m\frac{M+k-2}{M-1}\widetilde{\epsilon} \\
&= m(M+k-2)\exp\left\{ -\beta\left( \langle \mathbf{x}, \boldsymbol{\xi}_\mu \rangle - \underset{\nu \in [M]}{\text{Max}} \langle \mathbf{x}, \boldsymbol{\xi}_\nu \rangle \right) \right\},
\end{aligned} \tag{D.8}
$$

where $k := |\mathcal{M}|$, $m := \text{Max}_\mu \|\boldsymbol{\xi}_\mu\|$, $\widetilde{\epsilon} := (M-1)\exp\{-\beta\left(\langle \mathbf{x}, \boldsymbol{\xi}_\mu \rangle - \text{Max}_{\nu \in [M]} \langle \mathbf{x}, \boldsymbol{\xi}_\nu \rangle\right)\}$ and the inequality

$$
\left[ \text{Softmax}(\beta \Xi^\mathsf{T} \mathbf{x}) \right]_\nu = \frac{\exp\{\beta\left( \langle \mathbf{x}, \boldsymbol{\xi}_\nu \rangle - \langle \mathbf{x}, \boldsymbol{\xi}_\mu \rangle \right)\}}{1 + \sum_{\nu' \neq \mu} \exp\{\beta\left( \langle \mathbf{x}, \boldsymbol{\xi}_{\nu'} \rangle - \langle \mathbf{x}, \boldsymbol{\xi}_\mu \rangle \right)\}} \leq \exp\left\{ -\beta\left( \langle \mathbf{x}, \boldsymbol{\xi}_\mu \rangle - \underset{\nu \in [M]}{\text{Max}} \langle \mathbf{x}, \boldsymbol{\xi}_\nu \rangle \right) \right\},
$$

is used in (D.8). This completes the proof. $\qquad\square$

### D.5  Corollary 4.1.1 and Corollary 4.1.2

*Proof of Corollary 4.1.1 and Corollary 4.1.2.* Since $\langle \boldsymbol{\xi}_\mu, \mathbf{x} \rangle \geq \langle \boldsymbol{\xi}_\nu, \mathbf{x} \rangle$ for all $\nu \neq \mu$, we have

$$
[\text{Softmax}\left(\beta \Xi_\delta^\mathsf{T} \mathbf{x}\right)]_\mu \geq [\text{Softmax}\left(\beta \Xi_\delta^\mathsf{T} \mathbf{x}\right)]_\nu, \quad \text{for} \quad \nu \neq \mu.
$$

For the support set $\mathcal{M}$, we have

$$
\begin{aligned}
[\text{Softmax}\left(\beta \Xi_\mathcal{M}^\mathsf{T} \mathbf{x}\right)]_\mu &= \frac{\langle \boldsymbol{\xi}_\mu, \mathbf{x} \rangle}{\sum_{j \in \mathcal{M}} \langle \boldsymbol{\xi}_j, \mathbf{x} \rangle} \tag{D.9} \\
&\geq [\text{Softmax}\left(\beta \Xi_\delta^\mathsf{T} \mathbf{x}\right)]_\mu. \qquad \text{(By the smaller denominator in (D.9))}
\end{aligned}
$$

This implies $\sum_{\nu \in \mathcal{M}} \left[ \text{Softmax}\left(\beta \Xi_\mathcal{M}^\mathsf{T} \mathbf{x}\right) \right]_\nu \boldsymbol{\xi}_\nu$ is "pulled more strongly" toward $\boldsymbol{\xi}_\mu$ than $\sum_{\nu=1}^M \left[ \text{Softmax}\left(\beta \Xi_\delta^\mathsf{T} \mathbf{x}\right) \right]_\nu \boldsymbol{\xi}_\nu$. To see this, we write

$$
\begin{aligned}
&\|\mathcal{T}_{\text{Sparse}}(\mathbf{x}) - \boldsymbol{\xi}_\mu\| \\
&= \| \underbrace{\left( \left[ \text{Softmax}\left(\beta \Xi_\mathcal{M}^\mathsf{T} \mathbf{x}\right) \right]_\mu - 1 \right) \boldsymbol{\xi}_\mu}_{:=(I)} + \sum_{\nu \in \mathcal{M}, \nu \neq \mu} \left[ \text{Softmax}\left(\beta \Xi_\mathcal{M}^\mathsf{T} \mathbf{x}\right) \right]_\nu \boldsymbol{\xi}_\nu \|,
\end{aligned}
$$

and

$$
\begin{aligned}
&\|\mathcal{T}_{\text{Dense}}(\mathbf{x}) - \boldsymbol{\xi}_\mu\| \\
&= \| \underbrace{\left( \left[ \text{Softmax}\left(\beta \Xi_\delta^\mathsf{T} \mathbf{x}\right) \right]_\mu - 1 \right) \boldsymbol{\xi}_\mu}_{(II)} + \sum_{\nu \in [M], \nu \neq \mu} \left[ \text{Softmax}\left(\beta \Xi_\delta^\mathsf{T} \mathbf{x}\right) \right]_\nu \boldsymbol{\xi}_\nu \|.
\end{aligned}
$$

By (D.9), $(I) \geq (II)$. This means the pull toward $\boldsymbol{\xi}_\mu$ is strictly stronger, and hence the softmax-weighted average $\sum_{\nu \in \mathcal{M}} \left[\mathrm{Softmax}\left(\beta \boldsymbol{\Xi}_{\mathcal{M}}^{\mathsf{T}} \mathbf{x}\right)\right]_\nu \boldsymbol{\xi}_\nu$ lies closer to $\boldsymbol{\xi}_\mu$:

$$\|\mathcal{T}_{\mathrm{Sparse}}(\mathbf{x}) - \boldsymbol{\xi}_\mu\| - \|\mathcal{T}_{\mathrm{Dense}}(\mathbf{x}) - \boldsymbol{\xi}_\mu\| \leq 0 \iff \|\mathcal{T}_{\mathrm{Sparse}}(\mathbf{x}) - \boldsymbol{\xi}_\mu\| \leq \|\mathcal{T}_{\mathrm{Dense}}(\mathbf{x}) - \boldsymbol{\xi}_\mu\|. \tag{D.10}$$

This completes the proof of Corollary 4.1.1.

From (Ramsauer et al., 2020, Theorem 4), for any query $\mathbf{x}$, $\mathcal{T}_{\mathrm{Dense}}$ approximately retrieves a memory pattern $\boldsymbol{\xi}_\mu$ with retrieval error $\epsilon$ exponentially suppressed by $\Delta_\mu$:

$$\|\mathcal{T}(\mathbf{x}) - \boldsymbol{\xi}_\mu\| \leq 2m(M-1) \exp\left\{-\beta\left(\Delta_\mu - 2m \, \mathrm{Max}\left[\|\mathbf{x} - \boldsymbol{\xi}_\mu\|, \|\mathbf{x} - \mathbf{x}_\mu^\star\|\right]\right)\right\}.$$

By (D.10), $\mathcal{T}_{\mathrm{Sparse}}$ also enjoys above retrieval error bound. Therefore, $\mathcal{T}_{\mathrm{Sparse}}(\mathbf{x})$ retrieves a memory pattern $\boldsymbol{\xi}_\mu$ with high accuracy after a single activation with a sufficiently large $\Delta_\mu$. This completes the proof. $\qquad\square$

## D.6 Corollary 4.1.3

*Proof of Corollary 4.1.3.* Recall (Hu et al., 2023, Lemma 2.2) that for initial query $\mathbf{x}_0 \in S_\mu$

$$\lim_{t \to \infty} \|\mathbf{x}_t - \boldsymbol{\xi}_\mu\| = 0, \tag{D.11}$$

where $\{\mathbf{x}_t\}_{t=0}^\infty$ is a sequence generated by $\mathcal{T}_{\mathrm{Dense}}$ from $\mathbf{x}_0$, i.e. $\mathcal{T}_{\mathrm{Dense}}(\mathbf{x}_t) = \mathbf{x}_{t+1}$.

Moreover, recall that for any query pattern $\mathbf{x} \in S_\mu$

$$0 \leq \|\mathcal{T}_{\mathrm{Sparse}}(\mathbf{x}) - \boldsymbol{\xi}_\mu\| \leq \|\mathcal{T}_{\mathrm{Dense}}(\mathbf{x}) - \boldsymbol{\xi}_\mu\|. \tag{D.12}$$

By applying squeeze theorem on (D.12) and (D.11), we have

$$\lim_{t \to \infty} \|\widetilde{\mathbf{x}}_t - \boldsymbol{\xi}_\mu\| = 0,$$

where $\{\widetilde{\mathbf{x}}_t\}_{t=0}^\infty$ is a sequence generated by $\mathcal{T}_{\mathrm{Sparse}}$, i.e. $\mathcal{T}_{\mathrm{Sparse}}(\widetilde{\mathbf{x}}_t) = \widetilde{\mathbf{x}}_{t+1}$.

This completes the proof.

$\qquad\square$

## D.7 Lemma 4.1

*Proof of 4.1.* Following (Wu et al., 2024b; Hu et al., 2023), we define the separation of $\boldsymbol{\xi}_\mu$ at a given $\mathbf{x}$ from all memory patterns $\boldsymbol{\Xi}$ as

$$\widetilde{\Delta}_\mu := \min_{\nu, \nu \neq \mu} \left[\langle \mathbf{x}, \boldsymbol{\xi}_\mu \rangle - \langle \mathbf{x}, \boldsymbol{\xi}_\nu \rangle\right].$$

Plug above into (D.8), and get

$$\|\mathcal{T}_{\mathrm{Sparse}}(\mathbf{x}) - \boldsymbol{\xi}_\mu\| \leq m(M + k - 2) \exp\left\{-\beta \widetilde{\Delta}_\mu\right\}.$$

By Cauchy-Schwartz inequality, for all $\mu \in \mathcal{M}$,

$$|\langle \boldsymbol{\xi}_\mu, \boldsymbol{\xi}_\mu \rangle - \langle \mathbf{x}, \boldsymbol{\xi}_\mu \rangle| \leq \|\boldsymbol{\xi}_\mu - \mathbf{x}\| \cdot \|\boldsymbol{\xi}_\mu\| \leq \|\boldsymbol{\xi}_\mu - \mathbf{x}\| m,$$

we write $\widetilde{\Delta}_\mu$ in terms of $\Delta_\mu$:

$$\widetilde{\Delta}_\mu = \Delta_\mu - 2\|\boldsymbol{\xi}_\mu - \mathbf{x}\| m = \Delta_\mu - 2mR, \qquad\qquad (\text{By } \mathbf{x} \in S_\mu)$$

where $R$ is radius of the sphere $S_\mu$. Since $\mathcal{T}$ is a mapping $\mathcal{T} : S_\mu \to S_\mu$, output of the mapping $\mathcal{T}$ falls in $\mathcal{S}_\mu$ with radius $R$. Therefore, $R$ is lower-bounded by

$$R \geq (M + k - 2) \exp\{-\beta(\Delta_\mu - 2mR)\} m \geq \|\mathcal{T}(\mathbf{x}) - \boldsymbol{\xi}_\mu\|,$$

and thus

$$\Delta_\mu \geq \frac{1}{\beta} \ln\left(\frac{(M + k - 2)m}{R}\right) + 2mR.$$

This completes the proof. $\qquad\square$

## D.8  Proposition 4.1

We built our proof on top of (Hu et al., 2023, Lemma 2.1), which consists 3 steps:

- **(Step 1.)** We establish a more refined well-separation condition, ensuring that patterns $\{\boldsymbol{\xi}_\mu\}_{\mu\in[M]}$ are well-stored in $\mathcal{H}$ and can be retrieved by $\mathcal{T}$ with an error $\epsilon$ at most $R$.

- **(Step 2.)** This condition is then related to the cosine similarity of memory patterns, from which we deduce an inequality governing the probability of successful pattern storage and retrieval.

- **(Step 3.)** We pinpoint the conditions for exponential memory capacity and confirm their satisfaction.

*Proof of Proposition 4.1.*  Our proof is built on top of (Hu et al., 2023, Corollary 3.1.1) with a different well-separation condition.

Let $\Delta_{\min} := \mathrm{Min}_{\mu\in[M]} \Delta_\mu$ and $\theta_{\mu\nu}$ Here we define $\Delta_{\min}$ and $\theta_{\mu\nu}$ be the angle between two patterns $\boldsymbol{\xi}^\mu$ and $\boldsymbol{\xi}^\nu$.

In order for a pattern $\boldsymbol{\xi}_\mu$ to be well-stored, by Lemma 4.1, we need

$$\Delta_{\min} \geq \frac{1}{\beta} \ln\left(\frac{(M+k-2)m}{R}\right) + 2mR.$$

On the other hand, we observe

$$\Delta_{\min} = \mathrm{Min}_{1\leq\mu\leq\nu\leq M} \left[m^2\left(1-\cos(\theta_{\mu\nu})\right)\right] = m^2\left[1-\cos(\theta_{\min})\right],$$

where $\theta_{\min} := \mathrm{Min}_{1\leq\mu\leq\nu\leq M} \theta_{\mu\nu} \in [0,\pi]$. Then, we have

$$m^2\left[1-\cos(\theta_{\min})\right] \geq \frac{1}{\beta}\ln\left(\frac{(M+k-2)m}{R}\right) + 2mR. \tag{D.13}$$

As a result, the probability of successful storage and retrieval, i.e., the minimal separation $\Delta_{\min}$ that satisfies Lemma 4.1, is given by

$$P\left(\Delta_\mu \geq \frac{1}{\beta}\ln\left(\frac{(M+k-2)m}{R}\right) + 2mR\right) = 1-p.$$

Inserting (D.13) into above, we obtain

$$P\left(m^2\left[1-\cos(\theta_{\min})\right] \geq \frac{1}{\beta}\ln\left(\frac{(M+k-2)m}{R}\right) + 2mR\right) = 1-p.$$

From (Olver et al., 2010, Equation (4.22.2)), for $0 \leq \cos(\theta_{\min}) \leq 1$, $\cos(\theta_{\min})$ has an upper bound

$$\cos(\theta_{\min}) \leq 1 - \frac{\theta_{\min}^2}{5}.$$

It holds

$$P\left(\frac{m^2\theta_{\min}^2}{5} \geq \frac{1}{\beta}\ln\left(\frac{(M+k-2)m}{R}\right) + 2mR\right) = 1-p,$$

which leads to

$$P\left(M^{\frac{2}{d-1}}\theta_{\min} \geq \frac{\sqrt{5}M^{\frac{2}{d-1}}}{m}\left[\frac{1}{\beta}\ln\left(\frac{(M+k-2)m}{R}\right) + 2mR\right]^{\frac{1}{2}}\right) = 1-p.$$

For later convenience, here we introduce an extra $M^{2/d-1}$ on both sides.

Let $\omega_d := \frac{2\pi^{d+1/2}}{\Gamma(\frac{d+1}{2})}$ be the area of a $d$-dimensional unit sphere manifold, with $\Gamma(\cdot)$ denoting the gamma function.

Following (Brauchart et al., 2018, Lemma 3.5), we have

$$P\left(M^{\frac{2}{d-1}}\theta_{\min} \geq \frac{\sqrt{5}M^{\frac{2}{d-1}}}{m}\left[\frac{1}{\beta}\ln\left(\frac{(M+k-2)m}{R}\right) + 2mR\right]^{\frac{1}{2}}\right) = 1 - p$$

$$\geq 1 - \frac{1}{2}\gamma_{d-1}5^{\frac{d-1}{2}}M^2 m^{-(d-1)}\left[\frac{1}{\beta}\ln\left(\frac{(M+k-2)m}{R}\right) + 2mR\right]^{\frac{d-1}{2}}, \tag{D.14}$$

where $\gamma_d$ is the ratio between the surface areas of the unit spheres in $(d-1)$ and $d$ dimensions:

$$\gamma_d := \frac{1}{d}\frac{\omega_{d-1}}{\omega_d} = \frac{1}{d\sqrt{\pi}}\frac{\Gamma\left(\frac{d+1}{2}\right)}{\Gamma\left(\frac{d}{2}\right)}.$$

Recall $d, M \in \mathbb{N}_+$, $p \in [0, 1]$. Hence, it holds $M = \sqrt{p}C^{\frac{d-1}{4}}$ for some real values $C \in \mathbb{R}$.

Then, by (D.14), we have

$$5^{\frac{d-1}{2}}\left(\sqrt{p}C^{\frac{d-1}{4}}\right)^2 m^{-(d-1)}\left\{\frac{1}{\beta}\ln\left[\frac{\left(\sqrt{p}C^{\frac{d-1}{4}} + k - 1\right)m}{R}\right] + \frac{1}{\beta}\right\}^{\frac{d-1}{2}} - p \leq 0,$$

and thus

$$5^{\frac{d-1}{2}}C^{\frac{d-1}{2}}m^{-(d-1)}\left\{\frac{1}{\beta}\ln\left[\frac{\left(\sqrt{p}C^{\frac{d-1}{4}} + k - 1\right)m}{R}\right] + \frac{1}{\beta}\right\}^{\frac{d-1}{2}} \leq 1. \tag{D.15}$$

Further, we rewrite (D.15) as

$$\frac{5C}{m^2\beta}\left\{\ln\left[\frac{\left(\sqrt{p}C^{\frac{d-1}{4}} + k - 1\right)m}{R}\right] + 1\right\} - 1 \leq 0,$$

and identify

$$a := \frac{4}{d-1}\left\{\ln\left[\frac{m(\sqrt{p} + k - 1)}{R}\right] + 1\right\}, \quad b := \frac{4m^2\beta}{5(d-1)}.$$

By (Hu et al., 2023, Lemam 3.1), $C$ takes the form

$$C = \frac{b}{W_0(\exp\{a + \ln b\})}, \tag{D.16}$$

where $W_0(\cdot)$ is the upper branch of the Lambert $W$ function. Since the domain of the Lambert $W$ function is $x > (-1/e, \infty)$ and the fact $\exp\{a + \ln b\} > 0$, the solution for (D.16) exists.

When the inequality (D.15) holds, the lower bound on the exponential storage capacity $M$ can be written as:

$$M \geq \sqrt{p}C^{\frac{d-1}{4}}.$$

In particular, the above lower bound takes a form similar to (Ramsauer et al., 2020, Theorem 3).

This completes the proof. $\qquad\qquad\qquad\qquad\qquad\qquad\qquad\qquad\qquad\qquad\qquad\qquad\quad\square$

# E Nonparametric Modern Hopfield Family

In this section, we derive a family of modern Hopfield models as possible extensions based on the proposed framework (Theorem 3.1).[6]

## E.1 Linear Modern Hopfield Model

**Proposition E.1** (Linear Modern Hopfield Model). Let $\Phi(\mathbf{x}) = (\phi_1(\mathbf{x}), \ldots, \phi_d(\mathbf{x}))$ with the component $\phi$:

$$\phi_i(\mathbf{x}) := \frac{\text{elu}(\mathbf{x}[i]) + 1}{\sum_{\mu=1}^{M} \langle \Phi(\mathbf{x}), \Phi(\boldsymbol{\xi}_\mu + \delta\boldsymbol{\xi}_\mu) \rangle}, \quad \forall i \in [d], \tag{E.1}$$

where $\text{elu}(\cdot)$ denotes the exponential linear unit activation function proposed by (Clevert et al., 2015). By Theorem 3.1, fitting $\mathcal{T}_{\text{SVR}}$ on $\mathcal{D}$ following (3.2) gives

$$\mathcal{T}_{\text{Linear}}(\mathbf{x}) = \frac{\sum_{\mu=1}^{M} \langle \Phi(\mathbf{x}), \Phi(\boldsymbol{\xi}_\mu + \delta\boldsymbol{\xi}_\mu) \rangle \boldsymbol{\xi}_\mu}{\sum_{\nu=1}^{M} \langle \Phi(\mathbf{x}), \Phi(\boldsymbol{\xi}_\nu + \delta\boldsymbol{\xi}_\nu) \rangle}.$$

By setting the kernel mapping $\Phi$ to linear feature map (E.1), we obtain a **linear modern Hopfield model** with linear complexity $\mathcal{O}(n)$. Compared with dense modern Hopfield model, our proposed linear modern Hopfield model has time and memory complexity $\mathcal{O}(n)$ instead of $\mathcal{O}(n^2)$ since we only need to compute $\sum_{\mu=1}^{M} \Phi(\boldsymbol{\xi}_\mu + \delta\boldsymbol{\xi}_\mu)\boldsymbol{\xi}_\mu$ and $\sum_{\mu=1}^{M} \Phi(\boldsymbol{\xi}_\mu + \delta\boldsymbol{\xi}_\mu)$ once and reuse them for the computation of every query pattern. This model is by design connected to the random attention of linear attention (Katharopoulos et al., 2020).

## E.2 Multi-Head Modern Hopfield Models

To derive the multi-head Hopfield model, we cast $\mathcal{T}_{\text{Multi}}$ as multiple SVR problems such that the memorization of memory patterns $\boldsymbol{\Xi}$ corresponds to training a regression model $\mathcal{T}_{\text{Multi}}$ on datasets $\{\boldsymbol{\Xi}_s\}_{s\in[H]}$ with noises $\{\boldsymbol{\Xi}\}$. These $S$ training data sets are given as $\{(\boldsymbol{\xi}_\mu^1 + \delta\boldsymbol{\xi}_\mu^1, \boldsymbol{\xi}_\mu^1)\}_{\mu\in[M]}, \cdots, \{(\boldsymbol{\xi}_\mu^H + \delta\boldsymbol{\xi}_\mu^H, \boldsymbol{\xi}_\mu^H)\}_{\mu\in[M]}$. To handle multiple regression problems, we extend the regression model (3.1) into the following.

**Definition E.1** (Multi-Head Regression Model). Given an input vector $\mathbf{x} \in \mathbb{R}^d$. The output $\widehat{\mathbf{y}} \in \mathbb{R}^d$ of the regression model $\mathcal{T}_{\text{multi}}$ is defined as:

$$\widehat{\mathbf{y}} = \mathcal{T}_{\text{Multi}}(\mathbf{x}) := \sum_{s=1}^{H} \mathbf{W}_O^s \left( \mathbf{W}^s \Phi^s(\mathbf{x}) \right) \in \mathbb{R}^d,$$

where $\mathbf{W}_O^s \in \mathbb{R}^{d\times d}$, $\mathbf{W}^s = [\mathbf{w}_1^s, \cdots, \mathbf{w}_d^s]^\mathsf{T} \in \mathbb{R}^{d\times D_\Phi}$ for all $s \in [H]$, and $\Phi^s(\mathbf{x}) = (\phi_1^s(\mathbf{x}), \cdots, \phi_{D_\Phi}^s(\mathbf{x})) : \mathbb{R}^d \to \mathbb{R}^{D_\Phi}$ denote a series of output projection matrices, weighted matrix and kernel mapping, respectively.

Adopting this multi-head regression model, we introduce the following multi-head modern Hopfield model.

**Proposition E.2** (Multi-Head Modern Hopfield Models). Let $\Phi(\cdot) = (\phi_0^{(0)}, \phi_1^{(1)}, \ldots, \phi_{D_1}^{(1)}, \ldots, \phi_1^{(n)}, \ldots, \phi_{D_n}^{(n)}, \ldots)$ with, for $1 \le D' \le D_n$,

$$\phi_{D'}^{(n)} := \frac{(\sqrt{\beta}x_1)^{\ell_1} \cdots (\sqrt{\beta}x_d)^{\ell_d}}{\sum_{\mu=1}^{M} \langle \Phi(\boldsymbol{\xi}_\mu + \delta\boldsymbol{\xi}_\mu), \Phi(\mathbf{x}) \rangle \cdot \sqrt{\ell_1! \cdots \ell_d!}},$$

where $\ell_1 + \cdots + \ell_d = n$, and $D_n := \binom{d+n-1}{n}$. By Theorem 3.1, fitting $\mathcal{T}_{\text{SVR}}$ on $H$ training data sets $\{(\boldsymbol{\xi}_\mu^1 + \delta\boldsymbol{\xi}_\mu^1, \boldsymbol{\xi}_\mu^1)\}_{\mu\in[M]}, \cdots, \{(\boldsymbol{\xi}_\mu^H + \delta\boldsymbol{\xi}_\mu^H, \boldsymbol{\xi}_\mu^H)\}_{\mu\in[M]}$ following (3.2) gives

$$\mathcal{T}_{\text{Multi}}(\mathbf{x}) = \sum_{s=1}^{H} \mathbf{W}_O^s \left( \boldsymbol{\Xi}_s \, \text{Softmax}(\beta\boldsymbol{\Xi}_\delta^\mathsf{T}\mathbf{x}) \right).$$

---

[6]Hu et al. (2024b) provide a theoretical characterization of these possible extensions from the perspective of fine-grained complexity theory.

This model is by design connected to the standard multi-head attention.

### E.3 PRFs (Positive Random Features) Kernel Modern Hopfield Model

**Proposition E.3** (Positive Random Features Modern Hopfield Model). Let $\Phi(\cdot) = (\phi_1, \ldots, \phi_{D_\Phi})$ with

$$\Phi(\mathbf{x}) \coloneqq \frac{\Psi(\mathbf{x})}{\sqrt{D_\Phi}}(\psi_1(\langle \mathbf{p}_1, \mathbf{x} \rangle), \ldots, \psi_1(\langle \mathbf{p}_m, \mathbf{x} \rangle), \ldots, \psi_l(\langle \mathbf{p}_1, \mathbf{x} \rangle), \ldots, \psi_l(\langle \mathbf{p}_m, \mathbf{x} \rangle)),$$

where $D_\Phi = l \cdot m$, $\Psi : \mathbb{R}^d \to \mathbb{R}$, $\psi_1, \ldots, \psi_m$ are functions that map from $\mathbb{R} \to \mathbb{R}$, and $\mathbf{p}_1, \ldots, \mathbf{p}_m \stackrel{iid}{\sim} \mathcal{P}$ are vectors from some distribution $\mathcal{P} \in \Delta^d$ ($\Delta^d \coloneqq \{\mathbf{p} \in \mathbb{R}^d_+ \mid \sum_{i=1}^d p_i = 1\}$ is the $(d-1)$-dimensional unit simplex.). By Theorem 3.1, fitting $\mathcal{T}_{\text{SVR}}$ on $\mathcal{D}$ following (3.2) gives

$$\mathcal{T}_{\text{PRF}}(\mathbf{x}) = \sum_{\mu=1}^M \mathbb{E}_{\mathcal{D}}[\widehat{D}^{-1} \langle \Phi(\mathbf{x}), \Phi(\boldsymbol{\xi}_\mu + \delta\boldsymbol{\xi}_\mu) \rangle]\boldsymbol{\xi}_\mu,$$

where we adopt the normalization map $\widehat{D}^{-1} \coloneqq \langle \boldsymbol{\xi}_1, \mathbf{x} \rangle$ given by (Choromanski et al., 2021).

Comparing with regular modern Hopfield model, PRF Hopfield model[7] only has the linear space and time complexity, without any additional treatment such as introducing sparsity or low-rankness. The significance of this representational capability lies in its ability to facilitate a precise comparison between softmax and alternative kernels in the context of extensive tasks, surpassing the capabilities of regular modern Hopfield models and enabling a comprehensive exploration of optimal kernels. This model is by design connected to the Performer-type attention (Choromanski et al., 2021). In practice, the default option for $\mathcal{P}$ is standard Gaussian (Choromanski et al., 2021).

---

[7]Along the same line of research, Hoover et al. (2024) also utilizes random feature approximation in a recurrent setting to facilitate compressed memory storage for associative memory models.

# F  Nonparametric Modern Hopfield Layers for Deep Learning

Building on the link between the nonparametric modern Hopfield models and the attention mechanisms, we introduce the Nonparametric Hopfield (NPH) layers for deep learning.

Following (Hu et al., 2023; Ramsauer et al., 2020), we say $\mathbf{X}$ and $\boldsymbol{\Xi}$ are in the associative space (embedded space), as they are embedded from the *raw* query $\mathbf{R}$ and $\mathbf{Y}$ memory patterns, respectively, via $\mathbf{X}^{\mathsf{T}} = \mathbf{R}\mathbf{W}_Q \coloneqq \mathbf{Q}$, and $\boldsymbol{\Xi}^{\mathsf{T}} = \mathbf{Y}\mathbf{W}_K \coloneqq \mathbf{K}$, with some $\mathbf{W}_Q$ and $\mathbf{W}_K$. Taking the transpose of $\mathcal{T}$ in (3.3) (with a given feature map $\Phi$) and multiplying with $\mathbf{W}_V$ such that $\mathbf{V} \coloneqq \mathbf{K}\mathbf{W}_V$, we have

$$\mathbf{Z} \coloneqq \mathbf{Q}^{\text{new}}\mathbf{W}_V = \mathcal{T}_{\text{SVR}}\left(\beta\mathbf{Q}\mathbf{K}^{\mathsf{T}}\right)\mathbf{V}, \tag{F.1}$$

which leads to an attention mechanisms with various $\mathcal{T}_{\text{SVR}}$ as activation functions. Plugging back the raw patterns $\mathbf{R}$ and $\mathbf{Y}$, we arrive the Nonparametric Modern Hopfield (NPH) layer(s),

$$\text{NPH}\left(\mathbf{R}, \mathbf{Y}\right) = \mathcal{T}_{\text{SVR}}\left(\beta\mathbf{R}\mathbf{W}_Q\mathbf{W}_K^{\mathsf{T}}\mathbf{Y}^{\mathsf{T}}\right)\mathbf{Y}\mathbf{W}_K\mathbf{W}_V, \tag{F.2}$$

which can be seamlessly integrated into deep learning architectures. Concretely, the NPH layers take matrices $\mathbf{R}$, $\mathbf{Y}$ as inputs, with the weight matrices $\mathbf{W}_Q$, $\mathbf{W}_K$, $\mathbf{W}_V$. Depending on its configuration, it offers several functionalities:

1. **Memory Retrieval:** In this learning-free setting, weight matrices $\mathbf{W}_K$, $\mathbf{W}_Q$, and $\mathbf{W}_V$ are set as identity matrices. Here, $\mathbf{R}$ represents the query input, and $\mathbf{Y}$ denotes the stored memory patterns for retrieval.

2. NPH**:** This configuration takes $\mathbf{R}$ and $\mathbf{Y}$ as inputs. Intending to substitute the attention mechanism, the weight matrices $\mathbf{W}_K$, $\mathbf{W}_Q$, and $\mathbf{W}_V$ are rendered learnable. Furthermore, $\mathbf{R}$, $\mathbf{Y}$, and $\mathbf{Y}$ serve as the sources for query, key, and value respectively. Achieving a self-attention-like mechanism requires setting $\mathbf{R}$ equal to $\mathbf{Y}$.

3. NPHPooling**:** With inputs $\mathbf{Q}$ and $\mathbf{Y}$, this layer uses $\mathbf{Q}$ as a static **prototype pattern**, while $\mathbf{Y}$ contains patterns over which pooling is desired. Given that the query pattern is replaced by the static prototype pattern $\mathbf{Q}$, the only learnable weight matrices are $\mathbf{W}_K$ and $\mathbf{W}_V$.

4. NPHLayer**:** The NPHLayer layer takes the query $\mathbf{R}$ as its single input. The layer equips with learnable weight matrices $\mathbf{W}_K$ and $\mathbf{W}_V$, which function as our stored patterns and their corresponding projections. This design ensures that our key and value are decoupled from the input. In practice, we set $\mathbf{W}_Q$ and $\mathbf{Y}$ as identity matrices.

**Remark F.1.** We emphasize that Hopfield memory models and Hopfield networks/layers are conceptually distinct:

- **Hopfield memory model:** A fixed content-addressable memory model with no training; retrieval is based on similarity to stored patterns. This is our focus.

- **Hopfield networks (e.g., Hopfield Layers (Brandstetter, 2021)):** Neural layers integrated into deep learning, trained with backprop. They builds on the "Dense Associative Memory $\leftrightarrow$ Transformer Attention" correspondence (Ramsauer et al., 2020), later generalized by (Hu et al., 2023; Wu et al., 2024b). These includes architectural innovations such as additional memory-enhanced functionalities studied in prior works (e.g., (Ramsauer et al., 2020) for prototype learning, (Schimunek et al., 2023) for template enriching and (Wu et al., 2024b) for fast test-time adaptation.)

Our work extends the prior modern Hopfield memory model by framing it as a nonparametric regression problem. This allows efficient variants (Sections 3 and 4) and connects it to attention mechanisms (e.g., Performer (Choromanski et al., 2021)) under a rigorous, unified theory.

# G   Experimental Studies

**Tasks.**   We verify the method proposed in the main content with the following experimental sections. These tasks mainly focus on (1) validating the theoretical results, (2) the real-world application and (3) computational efficiency.

- Appendix G.1: Memory Retrieval Task (Figure 2).

- Appendix G.2: Multiple Instance Learning on MNIST (Figure 4).

- Appendix G.3: Multiple Instance Learning on Real World Datasets.

- Appendix G.4: Time Series Prediction.

- Appendix G.5: Computational Efficiency.

**Baselines and Considered Models.**   We consider the following variations of Modern Hopfield Models in this paper:

- Dense Modern Hopfield (Ramsauer et al., 2020)

- Sparse Modern Hopfield (Hu et al., 2023)

- Sparse-Structured Modern Hopfield:
    - Random Masked Modern Hopfield
    - Window Modern Hopfield
    - Top-K Modern Hopfield

- Linear Modern Hopfield

- Random Feature Modern Hopfield

**Experiment Environment.**   All experiments are conducted on the platform with NVIDIA GEFORCE RTX 2080 Ti and INTEL XEON SILVER 4214 @ 2.20GHz. We use PyTorch 1.8.0 for all experiments, and use RayTune for hyperparameter search.

**Remark G.1** (One-Step Retrieval).   For simplicity and as a proof of concept, all experiments in this work use single-step (feed-forward) retrieval without iterative updates. This avoids confusion with multi-step or recurrent retrieval.

## G.1   Memory Retrieval Task (Figure 2)

In the memory retrieval task, we examine two datasets: MNIST (sparse) and CIFAR10 (dense). We employ the sum-of-squares distance between the retrieved image and the ground truth image to measure retrieval error. This experiment encompasses two settings:

1. Half-masked image recovery, and

2. Noisy image recovery.

In the half-masked image recovery scenario, we obscure half of the pixels in the image. The memory set size ($M$) is varied from 10 to 200, and we report the average retrieval error (sum-of-square difference) over 50 runs. In the noisy image recovery scenario, we fix the memory set size at 100, and introduce varying scales of Gaussian noise to the image, with variance ranging from 0.1 to 1.4.

**Implementation Details.**   The memory set itself is chosen randomly from the dataset in each iteration. We adhere to the implementation outlined in (Hu et al., 2023).

**Results.** We first see clear differences in retrieval success among different Hopfield models (Figure 2).

- For top-$k$ Hopfield models, retrieval success depends directly on sparsity set by $k$. Lower $k$ means higher sparsity and lower retrieval errors. This aligns our theory (Theorem 4.1).

- The top-$k$ model performs similarly to the Sparse Hopfield model (Hu et al., 2023) on sparse data (MNIST) with smaller $k$ (higer sparsity). This aligns our theoretical result on capacity (Proposition 4.1). Our sparse-structured model maintains exponential memory capacity when the target memory is within support, ensured by the top-$k$ operation.

- In contrast, random masked Hopfield models perform poorly, especially on MNIST (higher sparsity). This is because random masking may remove the target memory. This violates a key assumption ($\mu \in \mathcal{M}$) in Theorem 4.1. Thus, their poor performance is expected.

These numerical results (corresponding to Theorem 4.1 and Proposition 4.1) confirm that careful sparse strategies is capable of matching or exceeding dense Hopfield retrieval in sparse settings.

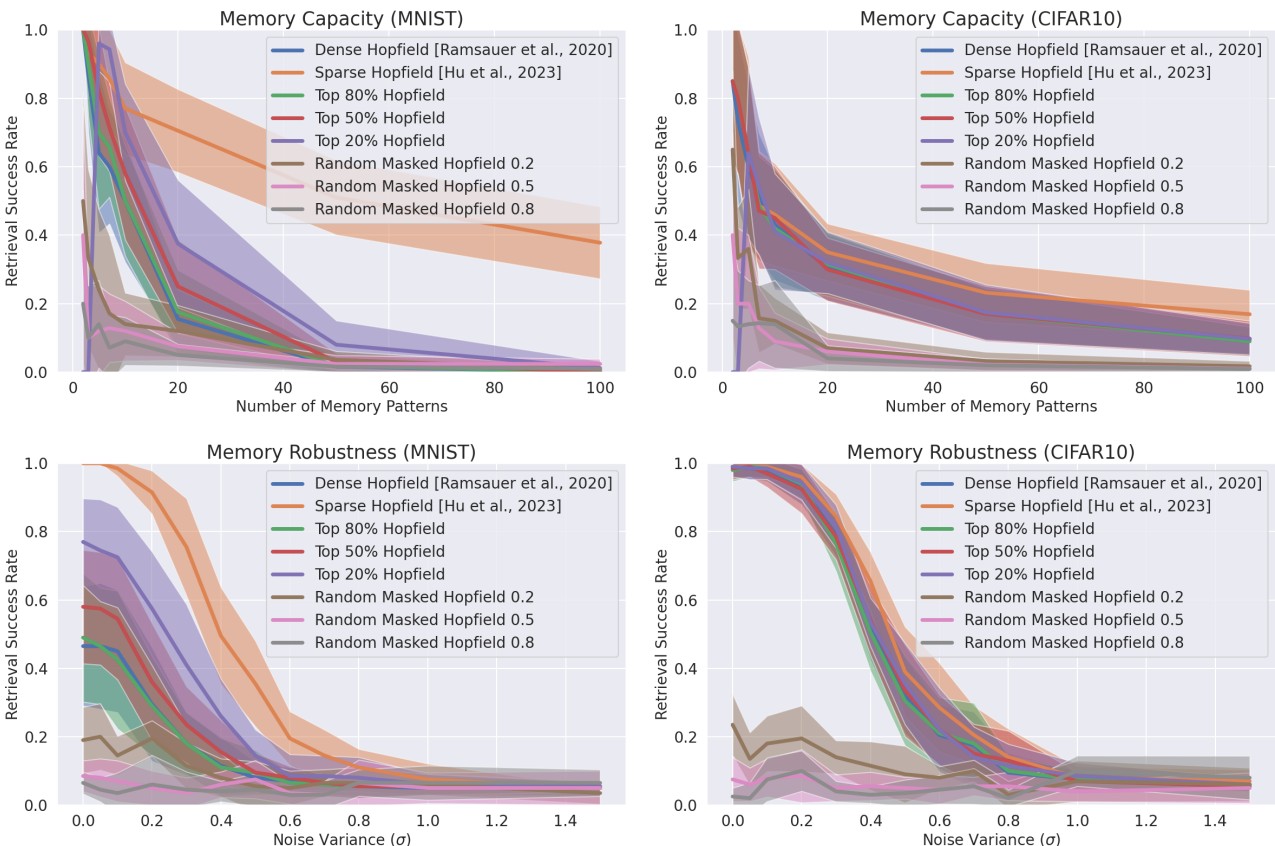

*Figure 2.* **Numerical Justifications for Theoretical Results: Memory Capacity and Noise Robustness. (Upper):** Retrieval success from half-masked queries (Theorem 4.1 and Proposition 4.1). **(Lower):** Retrieval success with different levels of Gaussian noise (Remark 4.2). We set $\beta = 0.01$ (MNIST) and 0.1 (CIFAR10). Lines show averages over 10 runs. Shaded areas show standard deviations. Top-$k$ Hopfield models have retrieval rates tied to their sparsity: smaller $k$ gives lower success, aligning our theory (Theorem 4.1). Top-$k$ models perform similarly to Sparse Hopfield (Hu et al., 2023) on sparse data (MNIST). This aligns our theory on memory capacity (Proposition 4.1). Random masked models perform poorly, especially on MNIST. This is expected since random masking can remove the target memory, violating our the $\mu \in \mathcal{M}$ assumption in Theorem 3.2. These results confirm our theory: careful sparsity can match or surpass dense models in sparse settings.

## G.2 Multiple Instance Learning on MNIST (Figure 3 & Figure 4)

> **Quoted from (Hu et al., 2023, Section 4.2):**
>
> **Multiple Instance Learning (MIL)** (Ilse et al., 2018; Carbonneau et al., 2018) is a variation of supervised learning where the training set consists of labeled bags, each containing multiple instances. The goal of MIL is to predict the bag labels based on the instances they contain, which makes it particularly useful in scenarios where labeling individual instances is difficult or impractical, but bag-level labels are available. Examples of such scenarios include medical imaging (where a bag could be an image, instances could be patches of the image, and the label could indicate the presence or absence of disease) and document classification (where a bag could be a document, instances could be the words or sentences in the document, and the label could indicate the topic or sentiment of the document).

In this experiment, we evaluate Dense Hopfield, Sparse Hopfield, and Top-$K$ Hopfield models on a Multiple Instance Learning (MIL) task using MNIST bags. This task is standard in modern Hopfield model literture (Ramsauer et al., 2020; Hu et al., 2023; Santos et al., 2024b).

**Setup.** We designate one digit from MNIST as a negative signal, and the remaining digits as positive signals. The objective is to predict whether a given bag of instances (digits) contains the negative signal. We vary the memory size (number of instances per bag) to study how task difficulty affects each model's performance and convergence. We vary the memory set size ($M$) from 5 to 100 and report the mean accuracy over 10 runs. We compare the performance of Dense Hopfield, Sparse Hopfield, Top-K Hopfield (with 20%, 50%, and 80%), Random Feature Hopfield, Random Masked Hopfield and Linear Hopfield models. We omit the Window Hopfield model for reasons mentioned earlier.

**Implementation Details.** We employ an embedding layer to project the flattened MNIST images into the hidden space, followed by a layer of layer normalization. Subsequently, we utilize the Hopfield Pooling layer to pool over all the instances in the bag, followed by a second layer normalization layer. Finally, a fully connected layer is used to project the hidden representation of the bag into the label space. All models are trained using the AdamW optimizer for 150 epochs, with a cosine annealing learning rate decay applied to all models. Note that we exclude Window Hopfield in this and the subsequent MIL experiment since Window Hopfield requires both the query and memory pattern numbers to be large to perform the sliding window operation. However, in our model structure, the number of query patterns in the pooling layer is set to 2. The details of the hyperparameters can be found in Table 2.

**Results.** We report the results in Figure 3. Additionally, we also conduct a convergence analysis in Figure 4 with bag size = 50. We plot the loss and accuracy curve on MNIST MIL training and test set. Below, we summarize the key findings, connecting them to our theoretical guarantees (e.g. $\epsilon$-sparse convergence from Theorem 3 and the sparse masking assumption):

- **Robust Performance with Increasing Memory Size (Figure 3):** We measure accuracy as the bag size grows (x-axis of Table 2). Top-$K$ Hopfield and Sparse Hopfield keep high accuracy even with large bags. Dense Hopfield struggles: it attends to all instances, so distractors dilute the target signal. As a result, its performance drops when the bag is big. In contrast, Sparse and Top-$K$ Hopfield focus on the most relevant entries. They ignore low-similarity memories and avoid noise. Thus, they hold strong performance as bag size increases. This aligns with our theory (Theorem 4.1 and Corollary 4.1.1), which shows that masking out small similarities enforces a clear margin for correct retrieval. Random masked models do poorly if the mask removes the target memory, violating the $\mu \in \mathcal{M}$ assumption. Hence, selective sparsity helps retrieve the correct instance reliably, even with many distractors.

- **Convergence and Training Dynamics (Figure 4):** We compare the training loss and accuracy curves of various Hopfield models on the MNIST MIL task. Sparse and Top-$K$ Hopfield models converge quickly and stably: their loss drops sharply and accuracy climbs rapidly, reaching near-perfect performance within a few epochs. Notable, Random Feature Hopfield model also exhibits relatively fast convergence and competitive performance. This behavior aligns with our $\epsilon$-sparse convergence theorem (Theorem 4.1), as restricting updates to the largest memory entries drives the network to retrieve the correct instance in each bag without interference. In contrast, Dense Hopfield model (Ramsauer et al., 2020) improves more gradually and can plateau or oscillate before converging, reflecting weaker theoretical guarantees when many small memory entries compete. Also, Random masked models perform poorly with large masking (e.g. 80%). This is expected since random masking can remove the target memory, violating our the $\mu \in \mathcal{M}$

assumption in Theorem 3.2. Ultimately, all models converge, but the sparse variants reach high accuracy faster and more reliably.

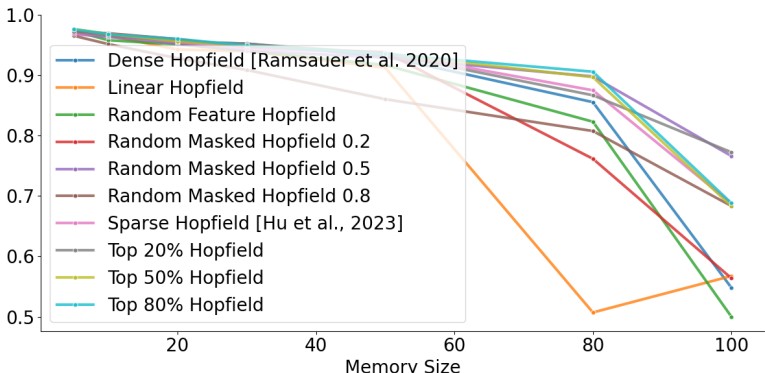

*Figure 3.* **MIL Accuracy vs. Memory Size on MNIST.** The y-axis represents the accuracy on test set. We compare test accuracy for Dense Hopfield (no sparsity), Sparse Hopfield (Hu et al., 2023), and Top-$k$ Hopfield (exactly $k$ active memory slots) models as the number of instances per bag increases. Larger memory size (more instances in a bag) makes the task more difficult due to more distractors. Dense (blue), Linear (orange), and Random Masked Hopfield models lose accuracy significantly with larger bags. In contrast, Sparse Hopfield (pink) and Top-$k$ models maintain high accuracy. Namely, sparse models filter irrelevant instances effectively. This aligns with our theoretical prediction that sparse retrieval improves robustness to large bag sizes (Theorem 4.1).

*Table 2.* Hyperparameter used in the MIL MNIST experiment.

| parameter | values |
| --- | --- |
| batch size | 256 |
| learning rate | 1e-3 |
| embedding dimension | 256 |
| number of heads | 4 |
| head dimension | 64 |
| test set size | 500 |
| train set size | 2000 |
| scaling | 0.1 |
| num of pattern | 2 |
| epochs | 150 |

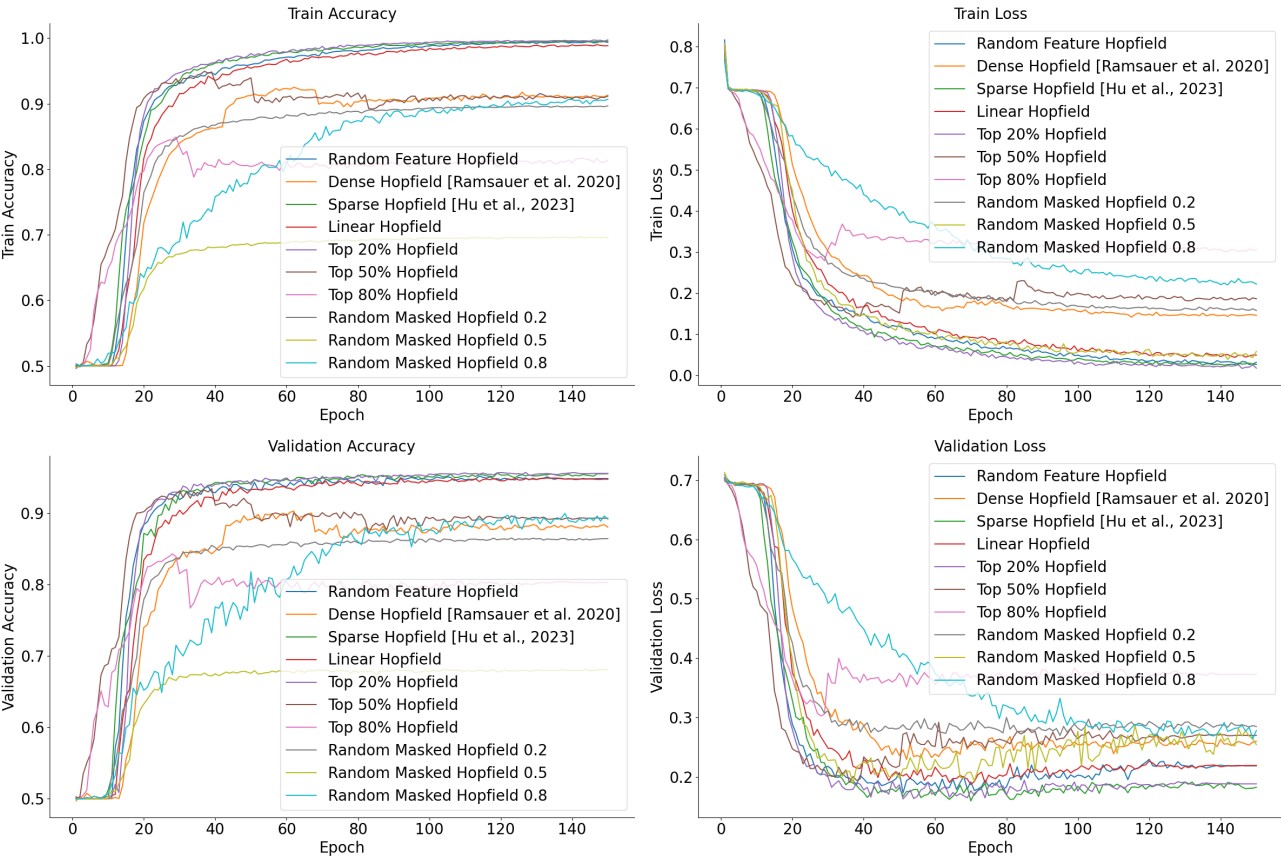

*Figure 4.* **Convergence Analysis of Hopfield Models on MNIST MIL. (Upper)**: Training loss and accuracy curves for various Hopfield models on the MNIST multiple instance learning task (Theorem 4.1). **(Bottom)**: Validation loss and accuracy curves on the same task (Theorem 4.1). All models are trained for 150 epochs with cosine annealing learning rate decay. Each line is the mean over 10 runs. Sparse Hopfield model by (Hu et al., 2023) attains the nearly the highest validation accuracy. Random Feature Hopfield model also converges quickly with competitive performance. Top-20% Hopfield model converges fast and shows minimal performance drop. Dense Hopfield model converges more slowly, exhibiting occasional plateaus. Again, Random masked models perform poorly with large masking (e.g. 80%). This is expected since random masking can remove the target memory, violating our the $\mu \in \mathcal{M}$ assumption in Theorem 3.2. In contrast, sparse updates (e.g. Sparse and Top-$K$) retrieve relevant entries more reliably, avoid spurious attractors, and ensure stable convergence. Consequently, these models learn faster and reach higher accuracy on sparse data (Theorem 4.1 and Corollary 4.1.1), while Dense Hopfield model faces interference from many memory entries. See Appendix G for more details.

### G.3 Multiple Instance Learning on Real World Datasets

For this experiment, we follow (Ramsauer et al., 2020; Hu et al., 2023) to conduct MIL experiments on real world datasets. However, we employ a simpler model structure and a smaller hyperparameter search space, rendering our results incomparable. We utilize four datasets: Elephant, Fox, and Tiger for image annotation (Ilse et al., 2018), and UCSB breast cancer classification (Kandemir et al., 2014). We compare Dense Hopfield, Sparse Hopfield, TopK Hopfield at 20%, 50%, and 80%, Random Feature Hopfield, and Linear Hopfield. Random Masked Hopfield is excluded due to its non-deterministic inference, and Window Hopfield is omitted as previously mentioned. The results are presented in Table 5.

**Dataset Details.**  The experiment is conducted on four MIL datasets. **Elephant**, **Fox**, and **Tiger** are designed for image annotation and consist of preprocessed and segmented colored images. Each image is characterized by descriptors for color, texture, and shape. These datasets each contain 100 positive images featuring the specified animal and 100 negative images drawn from a set of images depicting other animals. Additionally, we evaluate our model on the **UCSB** breast cancer classification task. In the UCSB dataset, each instance comprises a patch of a histopathological image depicting either cancerous or normal tissue. The detailed statistics of the datasets are reported in Table 3.

*Table 3.* Statistics of MIL benchmark datasets

| Name | Instances | Features | Bags | +bags | −bags |
|------|-----------|----------|------|-------|-------|
| **Elephant** | 1391 | 230 | 200 | 100 | 100 |
| **Fox** | 1302 | 230 | 200 | 100 | 100 |
| **Tiger** | 1220 | 230 | 200 | 100 | 100 |
| **UCSB** | 2002 | 708 | 58 | 26 | 32 |

**Implementation Details.**  We follow the experimental setting in (Ramsauer et al., 2020) and employ stratified 10-fold cross-validation to evaluate the performance of each baseline Hopfield model. In each fold, we utilize a stratified sampling process to partition the data into a training set and a validation set, with a split rate of 0.1. Hyperparameters are optimized via random search by maximizing the ROC-AUC score on the validation set. All reported ROC-AUC scores represent the average results over 5 runs with different random seeds. The random search space is delineated in Table 4, with the number of trials set to 50 for each fold. The embedding layer, a pre-HopfieldPooling linear network, has its layer width determined by the number of hidden units. A dropout operation, also referred to as bag dropout, is applied post the embedding layer and the Hopfield Pooling layer. Notably, to better showcase the performance of Top-k Hopfield, dropout is not applied to the attention weight. All models are trained using the Adam optimizer over 50 epochs. To mitigate overfitting, an early-stopping mechanism is employed, selecting the best checkpoint based on the validation set.

**Results.**  For real-world MIL datasets, Sparse Hopfield dominates most tasks (except for UCSB). However, other sparse-structured Hopfield models, especially Top-20% Hopfield, show comparable performance with Sparse Hopfield, indicating a potential trade-off between computational efficiency and model performance. For random feature and linear Hopfield, they did not outperform other baselines. However, their retrieval dynamics behave differently than other sparse-structured Hopfield models. Understanding how to fully utilize their potential and identifying the scenarios where they are most suitable is worth studying in the future

*Table 4.* Hyperparameter random search space on the respective validation sets of the Elephant, Fox, Tiger and UCSB breast cancer datasets.

| parameter | values |
|---|---|
| batch size | $\{4, 8, 16\}$ |
| learning rates | $\{10^{-3}, 10^{-4}, 10^{-5}\}$ |
| weight decay | $\{0, 10^{-3}, 10^{-4},\}$ |
| layer width | $\{128, 256, 512\}$ |
| number of heads | $\{4, 8\}$ |
| scaling factors | $\{0.1, 1\}$ |
| dropout | $\{0.0, 0.3\ 0.5\}$ |

*Table 5.* Results for MIL benchmark datasets in terms of AUC score. The results suggest that the proposed model achieves performance comparable to the existing Dense and Sparse Modern Hopfield models (Hu et al., 2023; Ramsauer et al., 2020). Note that, since our aim here is to conduct an *atomic* setting for fair comparison, we employ a simpler network structure (with smaller hyperparameter search space) compared to the ones used in (Hu et al., 2023; Ramsauer et al., 2020). Consequently, our results do not align with those in (Hu et al., 2023) for Dense and Sparse Modern Hopfield Models.

| Method | Tiger | Fox | Elephant | UCSB |
|---|---|---|---|---|
| Dense Hopfield (Ramsauer et al., 2020) | 0.813 | 0.563 | 0.877 | 0.524 |
| Sparse Hopfield (Hu et al., 2023) | 0.830 | 0.573 | 0.893 | 0.585 |
| Top-20% Hopfield | 0.824 | 0.562 | 0.848 | 0.586 |
| Top-50% Hopfield | 0.812 | 0.566 | 0.852 | 0.572 |
| Top-80% Hopfield | 0.812 | 0.560 | 0.872 | 0.551 |
| Random Feature Hopfield | 0.802 | 0.508 | 0.875 | 0.566 |
| Linear Hopfield | 0.797 | 0.571 | 0.869 | 0.561 |

## G.4 Time Series Prediction

We further showcase the performance (in Table 6) and efficiency (in Figure 5) of the proposed nonparametric modern Hopfield models with multivariate time series prediction tasks.

*Table 6.* **Time series prediction using different Hopfield layers (Appendix F) across five datasets.** We evaluate each dataset with different prediction horizons (showed in the second column). We report the average Mean Square Error (MSE) and Mean Absolute Error (MAE) metrics of 5 runs. **RF** denotes the **R**andom **F**eature Hopfield layer. One notable observation is that the noise level of the dataset significantly influences time series prediction. Therefore, employing Hopfield layers with strong noise-robustness offers performance improvements. Moreover, based on our results, the proposed efficient Hopfield models not only offer significant computational efficiency but also maintain comparable performance. Especially, the Random Feature Hopfield and Linear Hopfield layers (models) not only match but even outperform Dense Hopfield model in several settings. As a side note, Window Hopfield exhibits significant performance degradation in most settings. This degradation arises because it solely focuses on local information. Being the only Hopfield model that does not span the entire associative range (i.e., sequence length), it overlooks a substantial portion of the autoregressive correlation present in time series data. We also record the time used for one epoch on ETTh1 dataset with different prediction horizon (input length as well). The duration time per epoch was showed in Figure 5.

| Models | | Dense | | Sparse | | Top20% | | Top50% | | Top80% | | Window | | RF | | Linear | |
|---|---|---|---|---|---|---|---|---|---|---|---|---|---|---|---|---|---|---|
| Metric | | MSE | MAE | MSE | MAE | MSE | MAE | MSE | MAE | MSE | MAE | MSE | MAE | MSE | MAE | MSE | MAE |
| ETTh1 | 96 | 0.137 | 0.307 | 0.144 | 0.314 | 0.148 | 0.319 | 0.153 | 0.321 | 0.147 | 0.318 | 1.043 | 0.881 | 0.147 | 0.312 | 0.149 | 0.320 |
| | 192 | 0.153 | 0.326 | 0.152 | 0.325 | 0.146 | 0.318 | 0.161 | 0.333 | 0.150 | 0.320 | 1.003 | 0.870 | 0.158 | 0.332 | 0.141 | 0.313 |
| | 336 | 0.148 | 0.319 | 0.146 | 0.319 | 0.156 | 0.327 | 0.122 | 0.286 | 0.160 | 0.333 | 0.889 | 0.767 | 0.151 | 0.322 | 0.138 | 0.307 |
| | 720 | 0.169 | 0.331 | 0.148 | 0.314 | 0.184 | 0.345 | 0.161 | 0.327 | 0.123 | 0.287 | 0.756 | 0.761 | 0.141 | 0.271 | 0.171 | 0.333 |
| ETTm1 | 96 | 0.148 | 0.301 | 0.147 | 0.301 | 0.144 | 0.311 | 0.151 | 0.310 | 0.142 | 0.31o | 0.943 | 0.854 | 0.151 | 0.314 | 0.155 | 0.319 |
| | 192 | 0.189 | 0.350 | 0.187 | 0.340 | 0.191 | 0.347 | 0.185 | 0.338 | 0.188 | 0.341 | 1.054 | 0.893 | 0.190 | 0.347 | 0.192 | 0.348 |
| | 336 | 0.163 | 0.320 | 0.165 | 0.322 | 0.168 | 0.331 | 0.161 | 0.312 | 0.169 | 0.330 | 0.873 | 0.334 | 0.175 | 0.333 | 0.176 | 0.337 |
| | 720 | 0.159 | 0.300 | 0.161 | 0.303 | 0.165 | 0.313 | 0.167 | 0.313 | 0.169 | 0.320 | 0.764 | 0.731 | 0.162 | 0.309 | 0.165 | 0.310 |
| ECL | 96 | 0.378 | 0.371 | 0.373 | 0.370 | 0.384 | 0.382 | 0.386 | 0.386 | 0.383 | 0.376 | 0.989 | 0.854 | 0.390 | 0.403 | 0.365 | 0.378 |
| | 192 | 0.486 | 0.426 | 0.535 | 0.507 | 0.502 | 0.427 | 0.501 | 0.464 | 0.519 | 0.481 | 1.000 | 0.843 | 0.543 | 0.438 | 0.549 | 0.464 |
| | 336 | 0.748 | 0.693 | 0.760 | 0.688 | 0.650 | 0.549 | 0.674 | 0.571 | 0.638 | 0.545 | 1.012 | 0.849 | 0.767 | 0.588 | 0.672 | 0.578 |
| | 720 | 0.961 | 0.711 | 0.993 | 0.758 | 1.145 | 0.843 | 1.166 | 0.847 | 1.211 | 0.872 | 1.061 | 0.865 | 1.362 | 0.896 | 1.052 | 0.770 |
| WTH | 96 | 0.347 | 0.474 | 0.347 | 0.477 | 0.348 | 0.474 | 0.348 | 0.474 | 0.356 | 0.479 | 0.952 | 0.819 | 0.345 | 0.470 | 0.355 | 0.476 |
| | 192 | 0.399 | 0.505 | 0.386 | 0.497 | 0.360 | 0.482 | 0.370 | 0.490 | 0.361 | 0.482 | 0.977 | 0.828 | 0.368 | 0.487 | 0.354 | 0.478 |
| | 336 | 0.407 | 0.512 | 0.387 | 0.501 | 0.376 | 0.489 | 0.397 | 0.503 | 0.403 | 0.505 | 0.931 | 0.808 | 0.392 | 0.504 | 0.407 | 0.514 |
| | 720 | 0.669 | 0.631 | 0.632 | 0.623 | 0.590 | 0.604 | 0.569 | 0.593 | 0.618 | 0.618 | 0.564 | 0.595 | 0.564 | 0.595 | 0.747 | 0.676 |
| Traffic | 96 | 1.466 | 0.654 | 1.489 | 0.638 | 1.483 | 0.645 | 1.517 | 0.630 | 1.477 | 0.638 | 1.520 | 0.625 | 1.515 | 0.635 | 1.489 | 0.644 |
| | 192 | 1.551 | 0.654 | 1.550 | 0.657 | 1.557 | 0.649 | 1.548 | 0.657 | 1.551 | 0.652 | 1.570 | 0.637 | 1.551 | 0.654 | 1.551 | 0.653 |
| | 336 | 1.595 | 0.663 | 1.595 | 0.662 | 1.599 | 0.663 | 1.592 | 0.665 | 1.604 | 0.657 | 1.612 | 0.646 | 1.613 | 0.646 | 1.614 | 0.646 |
| | 720 | 1.660 | 0.681 | 1.671 | 0.671 | 1.664 | 0.674 | 1.676 | 0.663 | 1.682 | 0.661 | 1.683 | 0.661 | 1.682 | 0.661 | 1.681 | 0.660 |

### G.4.1 Implementation Details

For ease of comparison, we employ the simplest possible architecture: an embedding layer to project each signal into a hidden space, followed by a single Hopfield layer. By doing so, we treat every signal as a query pattern. Next, we employ a Hopfield Pooling layer to pool over all the signals into a single hidden vector. Finally, we utilize a fully connected layer to generate the prediction. For all experiments, we maintain the same input and prediction horizon for simplicity. The results can be found in Table 6 and Figure 5.

**Datasets.** We conduct the experiments on four multivariate time series real-world datasets: ETTh1 (Electricity Transformer Temperature-hourly), ETTm1 (Electricity Transformer Temperature-minutely), WTH (Weather), ECL (Electricity Consuming Load), Traffic.

**Setup.** For each dataset, we use their univariate setting for our time series prediction experiment. We choose Dense, Sparse, Random Feature, Linear, TopK and Window Hopfield as baselines. We select 4 different prediction horizons for demonstration, which are $96, 196, 336, 720$. We report the average error of 5 runs, evaluated using Mean Square Error (MSE) and Mean Absolute Error (MAE) metrics. For window Hopfield, we set the window size as $8, 12, 14, 16$, w.r.t. $96, 196, 336, 720$.

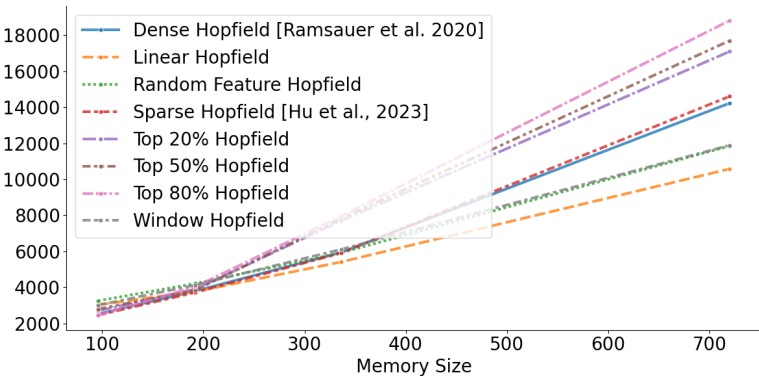

*Figure 5.* The processing time comparison among different Hopfield models utilized in the time series prediction task described in Table 6. We evaluate the efficiency of multivariate time series prediction on ETTh1 dataset. The findings are consistent with the efficiency discussion in Section 3.2, where the Sparse/Dense/Top-K models (all with $\mathcal{O}(d^2)$ complexity) necessitate more time to complete an epoch. In conjunction with the results in Figure 4, it is evident that the efficient modern Hopfield models (Window, Linear, Random Feature) not only converge in fewer or comparable epochs but also require less time per epoch compared to the less efficient (Sparse/Dense/Top-K) Hopfield models.

### G.5 Computational Efficiency

Here we demonstrate the computational overhead for different efficient modern Hopfield variants. We focus on the computational time duration and Flops (the number of Floating point operations). The results demonstrate

- For random masked Hopfield, the computational time scales up with respect to probability.

- Random feature Hopfield, Linear Hopfield and Window Hopfield demonstrates fast computational overhead in practice. In addition, these efficient Hopfield models also enjoy significantly lower floating point operations with only a marginal sacrifice in performance.

- Under PyTorch (version 1.11.0) framework, random masked Hopfield is not able to obtain computational efficiency improvement despite from its sparse-structured nature.

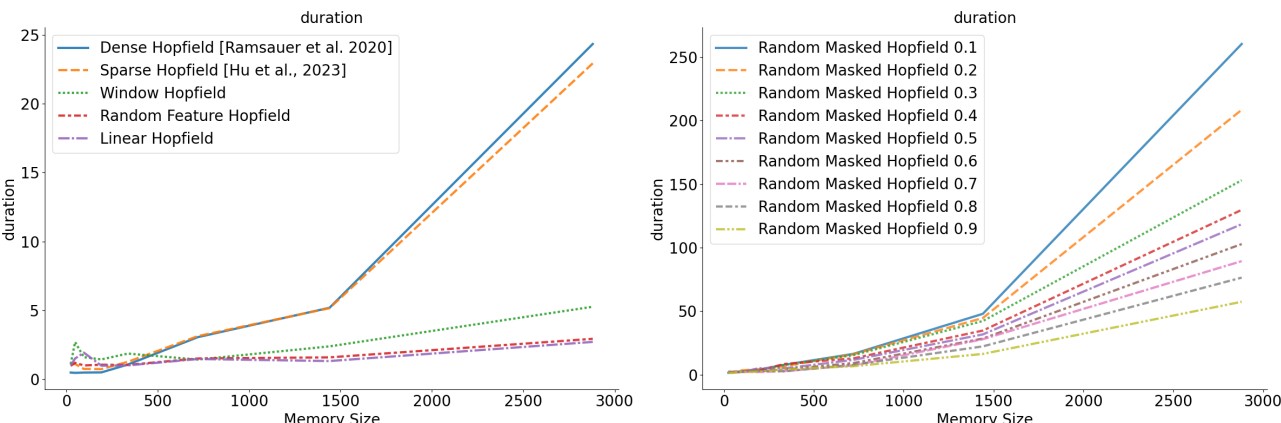

*Figure 6.* **(LHS:)** Comparison of duration (ms) per batch for different Hopfield Models. **(RHS:)** The scaling behavior of Random Masked Hopfield with different masking ratios. The probability denotes the ratio being masked out. We employ various variants of the `Hopfield` layers to process a batch of tensors, with a batch size of 4 and a hidden dimension of 16. We vary the input memory size (input length). Note that we separate the Random Masked Hopfield from other baselines since the sparse matrix operation in PyTorch, still in the beta stage, may not be as fully optimized as dense tensor operations.

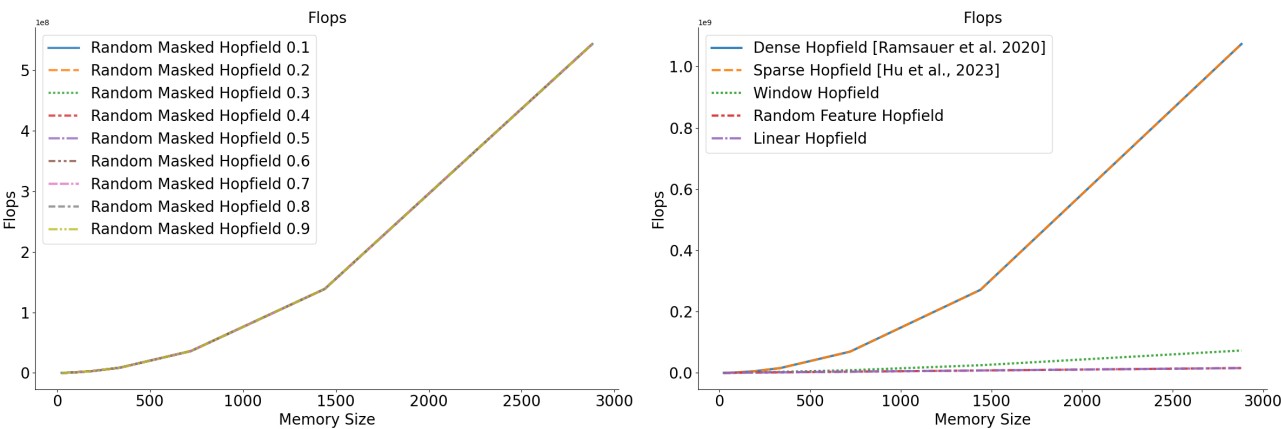

*Figure 7.* **(LHS:)** The FLOPs comparison for Random Masked Hopfield with different probabilities is depicted. The lines for Dense and Sparse Hopfield are overlapped, as are the lines for Random Feature Hopfield and Linear Hopfield. **(RHS:)** The FLOPs comparison across different Hopfield Models is shown. We employ the same settings as in the duration figure. Note that the **fvcore** package may count sparse matrix operations as normal floating point operations, which is why we might not see a difference.

**Implementation Details.** In this section, we exclusively evaluate the computational efficiency of different Hopfield models with respect to varying input lengths using the *Hopfield* layer. We report the average duration time per batch, as shown in Figure 6, and the FLOPs concerning different input lengths (memory sizes), as depicted in Figure 7. It's notable that different code implementation methods could potentially affect computational efficiency. We use a randomized batched tensor as input $x$, where $x \in \mathbb{R}^{\text{memory size} \times 16}$, and the batch size is 4 [8]. For Random Feature Hopfield and Linear Hopfield, we adhere to the Performer implementation[9], while for Window Hopfield, we follow the Longformer implementation[10]. For Random Masked Hopfield, we utilize the `torch.sparse.sampled_addmm`[11] feature, and for other baselines, we employ standard PyTorch built-in functions for implementation. We report the average forward pass time over 10 runs, alongside the FLOPs, with both metrics evaluated on different input lengths. FLOPs are calculated using the **fvcore** package[12]. Note that most publicly available packages for FLOPs profiling are either under development or in beta, hence calculation errors are anticipated. Additionally, the `torch.sparse` package is also in beta, implying its performance may not be fully optimized, especially regarding FLOPs calculation and operation overhead.

**Discussion.** Note that, by nature, both Dense and Sparse Hopfield exhibit the same FLOPs. Moreover, it is observed that Random Feature Hopfield and Linear Hopfield also share the same FLOPs, as the only distinction between them lies in the kernel function. Regarding Window Hopfield, its FLOPs fall in between, demonstrating notable efficiency compared to both Dense and Sparse Hopfield. In terms of duration time per batch, Sparse Hopfield appears slightly faster than its dense counterpart, likely due to the additional zeros generated by sparsemax. Window Hopfield, on the other hand, showcases a significant reduction in duration compared to Sparse Hopfield. Lastly, it is noted that the processing time for both Random Feature Hopfield and Linear Hopfield converges as the memory size increases.

---

[8]approximately $(4 \times 4 \times 16 \times \text{memory size})$ bytes

[9]https://github.com/lucidrains/performer-pytorch

[10]https://github.com/allenai/longformer

[11]https://pytorch.org/docs/stable/generated/torch.sparse.sampled_addmm.html#torch.sparse.sampled_addmm

[12]https://github.com/facebookresearch/fvcore

