# OpenReview forum: "Nonparametric Modern Hopfield Models"
_ICML.cc/2025/Conference — ICML 2025 poster_

### Official Review · Reviewer_g2qn · 2025-03-03

**Overall Recommendation:** 3

**Summary:**

This work proposes a non-parametric procedure to construct retrieval dynamics maps for modern Hopfield networks, based on a supervised support vector regression-like problem where contaminated patterns serve as a training data. The proposed procedure is shown to recover the standard dense retrieval dynamics from Ramsauer et al. (2020) as a particular choice of kernel. Building upon this procedure, the authors show that more efficient dynamics, based on applying a mask to the pattern samples, lead to sparse map with sub-quadratic complexity, while preserving desirable properties such as one-step retrieval of the patterns. Retrieval error upper-bounds for these Sparse-Structured models is also are also discussed.

**Claims And Evidence:**

The core of the manuscript is to propose a new method for designing retrieval dynamics maps for modern Hopfield networks. A few theoretical properties of the method are discussed, and these claims are supported by rigorous mathematical proofs. Numerical evidence for the advantage of the sparse-structured maps is also discussed in the appendix.

Overall, it is not very clear what are the drawbacks of the proposed method. For someone not familiar with the modern Hopfield literature, it would have been nice to have a broader limitation discussion in the paper.

**Essential References Not Discussed:**

Unfortunately I am not familiar with this literature.

**Experimental Designs Or Analyses:**

No.

**Methods And Evaluation Criteria:**

The bechmarking presented in the appendix uses standard data sets for retrieval tasks. It would have been nice to have experiments with synthetic data satisfying all the assumptions in the theory, to illustrate tasks for which the sparse method can significantly outperform the dense method.

**Other Comments Or Suggestions:**

I strongly encourage the authors to discuss the limitations of their work in perspective with the related literature.

**Other Strengths And Weaknesses:**

Overall, the paper is well-written and easy to follow. I appreciate the authors introduce modern Hopfield networks, as it helps the unfamiliar reader putting the results in perspective.

The main weakness is the lack of a critical discussion of the proposed method. All the results are seemingly positive, and a lack of a honest "limitation" section describing the main drawbacks raises a red flag for the unfamiliar reader.

**Questions For Authors:**

- In condition (T1), do you really need to monotonically decrease $E(x)$? This is strong, in principle rules out stochastic dynamics rules, e.g. SGD.
- Below eq. (3.2), are the inequalities applied to vectors component-wise? It would be nice to precise this in the text.

**Relation To Broader Scientific Literature:**

From my reading, the related literature on dense and sparse modern Hopfield networks is well acknowledged. However, I am unfortunately not familiar with this literature.

**Theoretical Claims:**

I skimmed through the proofs, and it looks sound. It mostly leverages standard results in SVR adapted to the setting of the paper.

---

> ### Author Rebuttal · Authors · 2025-03-26
>
> ### We thank the reviewer for the valuable comments. We have revised our paper to address each point.
>
> The revised draft is in this [anonymous Dropbox folder](https://www.dropbox.com/scl/fo/81sygn18f4ma3ridm1xlf/AHFYlvzMMlYZnNRhBN9U8mw?rlkey=e1tvpqs6v83kx2rspfvmfgswh&st=z9iuk4iu&dl=0).  All modifications are marked in BLUE color. Thanks!
>
> ---
>
> ### **1. Critical Discussion on Limitations (Newly Added Appendix B.2 https://imgur.com/a/R4PLZMs)**
>
>   - **Fixed Feature Mapping (Reviewer Concern: Realistic Embedding Scenarios)**
>     - We base our theoretical analysis on a fixed feature mapping.
>     - This choice streamlines convergence proofs but restricts the model if features evolve.
>     - Yet, **this condition is easy to meet. Moreover, in practice, static memory model do not have evolving feature map.**
>
>   - **Robustness to Random Masking (Reviewer Concern: Potential Retrieval Failures)**
>     - Our retrieval guarantee assumes the correct memory item remains in the support set.
>     - If the random mask excludes it, retrieval may fail.
>     - **While this assumption is restrictive, it is necessary for tractable analysis.**
>
>   - **Well-Separated Patterns for Exponential Capacity (Reviewer Concern: Data Correlations)**
>     - Our exponential capacity analysis requires patterns to be sufficiently distinct.
>     - This simplifies theoretical arguments, yet real datasets often have correlated patterns.
>     - Yet, **this setup is standard for characterizing capacity in literature** (Krotov & Hopfield, 2016; Demircigil et al., 2017; Ramsauer et al., 2020; Iatropoulos et al., 2022; Hu et al., 2023; Wu et al., 2024b; Santos et al., 2024a;b). Moreover, it can be relaxed by optimizing  the Hopfield-energy landscape for larger memory capacity.
>
>   - **Unproven Extensions in Section 5 (Reviewer Concern: Additional Convergence and Capacity Proofs)**
>     - We outline several model extensions but do not provide full proofs (no convergence and capacity guarantees).
>
> We see these points as opportunities for growth. Addressing them will extend our framework to broader settings and make retrieval more robust. We also invite the reviewer to check the newly added Appendix B.2 for detail in our latest revision.
>
> ---
>
> ### **2. Experiments and Sparsity (Methods And Evaluation Criteria)**
>    - **Our current experiments already simulate the conditions the theory addresses**, especially regarding sparsity of patterns.
>    - Namely, our experiments already simulate both sparse (MNIST) and denser (CIFAR-10) data. This aligns with our theoretical assumptions.
>    - **Our current numerical results already justify our theory.** Figure 2 in Appendix G.1 shows sparse models perform better on MNIST. This matches our claim that higher sparsity improves retrieval capacity and reduces error (Proposition 4.1, Theorem 4.1).
>    - In MIL supervised experiments, sparse models converge faster (Corollary 4.1.1). These results match the modern Hopfield literature (e.g., Hu et al. 2023, Wu et al. 2024, Ramsauer et al. 2020).
>
> We have revise the draft to emphasize these:
> * Revised G.1 "Results": https://imgur.com/a/OS2q77m and
> * Revised Fig 2 "Caption": https://imgur.com/a/JjxZwxv
> * Revised G.2 "Results", "Captions" of Fig 3, Fig 4: https://imgur.com/a/50mFsX5
>
> ---
>
> ### **3. Technical Clarifications**
>    - **Monotonic Decrease of $E(x)$ (Condition T1)**
>   Our convergence analysis assumes that each update decreases (or does not increase) the Hopfield energy $E(x)$. **This is a standard property of energy-based Hopfield models** and guarantees deterministic convergence to a fixed point. It does not apply to stochastic or non-monotonic updates. We emphasize that **the Hopfield memory model serves a different purpose than deep learning models**: here, minimizing energy corresponds to pattern retrieval, not parameter learning. Comparing this condition to stochastic optimization in deep learning (e.g., SGD) is not appropriate—the objectives and dynamics differ fundamentally.
>
>    - **Inequalities Below Eq. (3.2)**: We confirm these are component-wise inequalities. We clarify this in the revised text (https://imgur.com/a/wu2eW9A).
>
> ---
>
> We appreciate the reviewer’s time and attention to details. The revisions should address the concern about missing limitations and clarify our theoretical and experimental contributions. We hope the updated manuscript meets your expectations.

---

> > ### Comment · Reviewer_g2qn · 2025-04-01
> >
> > I thank the authors for their rebutal and for accounting for my suggestion. I am keeping my score.

---

> > > ### Author Response · Authors · 2025-04-03
> > >
> > > Thank you again for your thorough review. We hope our clarifications and revisions have fully addressed your concerns. Please let us know if further clarification or revision is needed. We greatly appreciate your time and effort.

---

### Official Review · Reviewer_v1Bp · 2025-03-10

**Overall Recommendation:** 4

**Summary:**

This is a theoretical paper that introduces a non-parametric interpretation of Hopfield Nets. The proposed method uses SVR to learn a parameter matrix $W$ mapping from feature space to data space. Additionally, rather than focusing on "memorizing data" as the original Hopfield Nets do, the authors propose to frame the AM task as a map from queries to memories, where the queries can be considered corrupted versions of the memories (auto-association), or units that are distinct from the keys (hetero-association).

**Claims And Evidence:**

The authors propose a novel "Nonparametric Hopfield" (NPH) Models, and claim this technique

1. Is compatible with modern Deep Learning architectures (this is not novel as the [Hopfield Layers](https://github.com/ml-jku/hopfield-layers) integrates similar ideas into modern architectures).
2. Comes with with strong (sub-quadratic) efficiency advantages (this is not novel since sub-quadratic complexity is also achieved by e.g., Performer) due to sparsity and use of basis functions.
3. The non-parametric Modern Hopfield framework can be used to formulate many existing methods.

Given that none of the above claimed contributions are novel on their own, it seems that the fundamental contribution and characterization of their NPH framework itself. Evidence for these contributions are proofs in the appendix.

**Essential References Not Discussed:**

It is worth mentioning [Hoover et. al 2024](https://arxiv.org/abs/2410.24153) for its use of random features in a *recurrent* framework for Associative Memory that converges to actual (not just generalized) fixed points.

**Experimental Designs Or Analyses:**

The paper offers no experimental validation of their claims in the main paper. However, Appendix G offers a study on memory retrieval and learning. Given my time and because this section has been relegated to the appendix, I will give only brief overview and comments:

I have a difficult time interpreting the results as the plots are noisy (many overlaid lines. Unclear what the takeaway message is for each figure). I have several questions on this section and I believe it can be improved substantially:

1. Are these experiments tested in the 1-step retrieval paradigm or is recurrence involved?
2. Fig 2 caption says that the masking did not work well, but I don't understand the justification of "violating the $\mu \in \mathcal{M}$ assumption. How is "masking" even applied? Via zero-ing out pixels? Replacing pixels with gaussian noise? Adding gaussian noise? Surely there must be a "masking" choice that keeps the correct pattern in the memory set.
3. How can I understand these results in light of the theoretical guarantees of the paper? Would be nice to see a reference line for what the theory would guarantee, or if the theory does not specify a bound for a particular experiment to explicitly mention that. Additionally, some of the plots are really noisy and its hard to tell what your method is.
4. Would be nice to see some qualitative pictures to understand what the experiments test.

I will continue my review as though this is a purely theoretical paper since these results do not add additional insights or clearly confirm the proposed theory.

**Methods And Evaluation Criteria:**

See "Experimental Designs & Analyses"

**Other Comments Or Suggestions:**

Please take my comments with a grain of salt since there were several aspects of this work that I did not understand.

1. The paper would benefit from a formal definition/justification of what "non-parametric" means in the context of NPH models. Isn't the matrix $\mathbf{W}$ the parameter matrix? For example, does the definition used to describe the SVR learning task work for understanding the NPH models?:
    > one that does not assume a specific functional form, and is flexible in the number of parameters. [L256]

2. The statement "competitive alternatives with memory-enhanced functionalities" [L415 right col] requires some experimental justification which should be discussed in the main paper.

3. For clarity of exposition, I would recommend the term "query" be used instead of "contaminated memory patterns" $\xi_\mu + \delta \xi_\mu$ (as currently described in e.g., Lemma 3.1 and [L200 right col]), as used in the beginning of sec 3.1 and again in 4.1. I believe this will help sell the message that this framework applies to both auto- and hetero- associative memories in the important introduction to the method.

**Other Strengths And Weaknesses:**

Strengths:
- **A general, novel approach**. The generalizability of this framework to existing methods of single-step associative memories is compelling.

Weaknesses:
- **Unclear empirical results**. It is not clear how the empirical results included in the appendix can be used to make any statements as to the efficacy and potential of this approach. Additionally, because the error in this method depends on how close together the memories are, it is unclear how strong the bounds are on large datasets.
- **Missing necessary background**. The introduction is difficult to read without strong background on papers Hu et al. and Wu et al. published in the past two years. The background explains this paper's relevance to attention (P3), describing the one-step update rule of Ramsauer et al. and the desiderata for learning an associative memory (i.e., must satisfy a monotonically decreasing energy function (T1) that converges to fixed points (T2)), but it does not elucidate the claims made in the introduction regarding sparsity (P2) and efficiency (P1). It would be nice if the related work section in the appendix (which I hope has a chance to make it into the main paper in some form when accepted) could serve as all the necessary background needed to understand this paper's contributions.

**Questions For Authors:**

1. I don't understand what's special about the Sparse-Structured formulation of NPH models. Definition 3.3 is simply the SVR optimization problem of Eq. (3.2) on a subset of the data points. If I understand correctly, you will never learn those data points which are not masked?

**Relation To Broader Scientific Literature:**

Specifically, NPH models store memories in a matrix that maps features back to the data space. This is a very general framework with which we can understand many recent optimizations of the MHN, esp. in the attention operation.

**Theoretical Claims:**

1. SVR can be used to train a matrix $\mathbf{W}$ that stores (query, memory) pairs
2. This framework can be extended to several modern Hopfield Networks (e.g., Ramsauer et al. Choromonski et al., Beltagy et al.) as an extension.

I did not check the proofs for these claims in the appendix. The definitions and methods are sound (see "questions" for my confusion about sparsity).

---

> ### Author Rebuttal · Authors · 2025-03-27
>
> Thanks for the constructive feedback. We have revised our paper to address each concern in detail.
>
> The revised draft is in this [anonymous Dropbox folder](https://www.dropbox.com/scl/fo/81sygn18f4ma3ridm1xlf/AHFYlvzMMlYZnNRhBN9U8mw?rlkey=e1tvpqs6v83kx2rspfvmfgswh&st=z9iuk4iu&dl=0).  All modifications are marked in BLUE color.
>
> ---
>
> ### 1. Empirical Clarity and Figure Explanations
>
> > **Comment:** The reviewer found our experimental setup unclear regarding retrieval steps, masking strategy, and theoretical connections. They also requested stronger links between figures and the underlying theory.
>
> **Response:**
>
> 1. **One-Step Retrieval:** We now explicitly state that *all experiments use a single feed-forward (one-step) retrieval* without iterative updates: https://imgur.com/a/lAJLKR6
> 2. **Masking Strategy:** The masking is done on similarity score level following random masked attention. It sometimes removes the target memory if done randomly (e.g., Random Masking). This is what the statement in Fig2 caption is about.  In contrast, top-$k$ masking retains the most relevant entries and preserves the correct memory in the set.
> 3. **Linking Theory to Experiments:** We have revised Appendix G.1 and G.2 to include direct references to the theorems and remarks in the main text:
>    - **Figures 2 & 3 (Thm 4.1, Prop 4.1, Coro 4.1.1):** These results highlight memory capacity limits and robustness under sparsity or noise.
>    - **Figure 2 (Remark 4.2):** Explains how different noise levels affect retrieval.
>    - **Figure 4:** Demonstrates quick convergence and stable performance under $\epsilon$-sparse retrieval. We note that random masking can remove the target memory, violating the $\mu \in \mathcal{M}$ assumption (Theorem 3.2).
>
> In the revised draft, we have updated the figure captions and added clarifications such that readers can immediately see the theoretical motivations for each experiment.
>
> Examples of these changes appear in the updated Appendix G.1 ("Results") and G.2 ("Captions" for Figures 2–4). E.g., https://imgur.com/a/OS2q77m and https://imgur.com/a/JjxZwxv and https://imgur.com/a/50mFsX5.
>
> ---
>
> ### 2. Focus on Hopfield Memory Models & Novelty
>
> > **Comment:** The reviewer questions the distinction between Hopfield networks and Hopfield memory models, and the novelty of our approach.
>
> **Response:**  We emphasize that **Hopfield memory models** and **Hopfield networks/layers** are conceptually distinct:
>
> - **Hopfield memory model**: A fixed content-addressable **memory model** with no training; retrieval is based on similarity to stored patterns. This is our focus.
> - **Hopfield network (e.g., Hopfield Layer)**: A **neural layer** integrated into deep learning, trained with backprop. This builds on the “Dense Associative Memory–Transformer Attention” correspondence (Ramsauer et al. 2021), later generalized by Hu et al. (2023) and Wu et al. (2024).
>
> Our work **extends the prior modern Hopfield memory model** by framing it as a nonparametric regression problem. This allows efficient variants (Sec 3–4) and connects it to attention mechanisms (e.g., Performer) under a rigorous, unified theory. To our knowledge, this unification has not been done before and constitutes a novel contribution.
>
> We have added a remark regarding above points https://imgur.com/a/E93HSM5. In the final version, we promise to move this remark to Sec 2 (background) if space allows.
>
> ---
>
> ### 3. Terminology & Background
>
> > **Comment:** The reviewer recommended refining terminology and providing enough background to ensure clarity.
>
> **Response:**
> - **Cite Hoover et al. 2024:** https://imgur.com/a/Ix7cp7n
> - **Question: Masked Target?** Yes, if a pattern is masked, it will not be memorized by the SVR. This allows us use different masking strategies to accompany different sparsity patterns in sec 3.2 & E.
> - **Nonparametric:** Yes, it does describe NPH. Despite having a matrix $W$ in the **primal** formulation of SVR, the model parameter is not fixed. This is more obvious in the **dual** form of SVR. In sec 3, we cast the retrieval dynamics $\mathcal{T}$ as SVR. The number of memories is proportional to to the number of support vectors, and hence proportional to model size. Thus the model size is not fixed. This is nonparametric (Remark 3.4). We had revised the Appendix C to strengthen this: https://imgur.com/a/E3Fnic5.
> - **Background in Main Text:** We agree that the related work section can provide more background to main text . When finalizing the paper, we will try our best to integrate this background if space allows.
> - **“Memory-Enhanced Functionalities” (L415, right):** We have added a remark to clarify this in Appendix F (https://imgur.com/a/E93HSM5).  In the final version, we promise to move this remark to Sec 2 (background) if space allows. This remark ties them to the benefits of content-addressable memory systems in real-world applications.
>
> ---
>
> Thank you for the review. Happy to discuss more :)

---

> > ### Comment · Reviewer_v1Bp · 2025-04-03
> >
> > Thank you for answering my questions and updating your draft for clarity. I am satisfied with the updated changes in the manuscript and believe the paper is now a stronger submission. I am updating my score to a 4.

---

> > > ### Author Response · Authors · 2025-04-03
> > >
> > > Thank you very much for your thorough review and for increasing the score. We are pleased that our revisions addressed your concerns and truly appreciate your feedback.

---

### Official Review · Reviewer_zGHJ · 2025-03-12

**Overall Recommendation:** 3

**Summary:**

This work leverages the concept of soft-margin Support Vector Regression (SVR) to reformulate modern Hopfield models as a non-parametric regression task, where a noisy target pattern is mapped to a reconstructed target pattern using Support Vector Machines (SVMs). By applying the Lagrange multiplier method, the problem is transformed by incorporating boundary conditions, ultimately deriving the mapping function $T_{SVR,\Phi}$, which serves as the update rule for the model.

Within this unified framework, the authors successfully reproduce the standard dense modern Hopfield model while also proposing a generalized Sparse-Structured Modern Hopfield Model. This new formulation significantly reduces computational complexity and introduces three specific efficient variants: Random Masked Modern Hopfield Model, Efficient Modern Hopfield Model, Top-K Modern Hopfield Model.

Furthermore, the authors conduct a rigorous theoretical analysis under this framework, leading to more precise sparsity-dependent retrieval error bounds and proving the fixed-point convergence of the sparse-structured Hopfield model.

**Claims And Evidence:**

The submission makes three main claims, each of which is examined below regarding its supporting evidence and potential concerns.

Claim #1: The paper provides a nonparametric framework and sub-quadratic complexity

* Supported:
The paper reformulates modern Hopfield models using soft-margin SVR, demonstrating a nonparametric regression-based approach.
Chapter 3 provides a formal derivation of the Sparse-Structured Modern Hopfield Model, proving its reduced computational complexity. Appendix G.5 provides empirical FLOPs and runtime comparisons.

* Concerns:
The computational efficiency improvements rely on asymptotic analysis, but practical benefits are not always consistent across implementations (e.g., PyTorch’s inefficiency with random masked Hopfield).

Claim #2: The paper provides a formal characterization of retrieval error bounds, noise robustness, and memory capacity.

* Supported:
Tighter retrieval error bound (Corollary 4.1.1, 4.1.2)
Stronger noise robustness (Remark 4.2)
Exponential capacity scaling (Lemma 4.1, Proposition 4.1)

* Concerns:
These results seem natural extensions of classical Hopfield models, raising the question of whether they provide fundamentally new insights.
Theoretical error bounds are not numerically validated—while the framework is well-formulated, empirical verification of retrieval error predictions is missing.

Claim #3: The framework bridges modern Hopfield models with attention mechanisms (e.g., BigBird, Longformer, Linear Attention).

* Supported:
The theoretical formulation indeed connects the proposed models to various attention mechanisms (such as BigBird, Longformer, Linear Attention, and Kernelized Attention).
Appendix G contains numerical experiments, showing performance metrics such as loss curves and accuracy trends.

* Concerns:
The experimental section lacks clear explanations of what conclusions should be drawn.
Weak connection between experiments and theoretical findings.

**Essential References Not Discussed:**

When discussing Hopfield networks, please also reference Amari’s earlier work (e.g., Amari, 1972), which laid a more detailed mathematical foundation for associative memory models and preceded Hopfield’s 1982 formulation.

**Experimental Designs Or Analyses:**

I reviewed the experimental design and analyses, particularly those presented in Appendix G. The experiments are well-structured, with clear descriptions of the models, parameters, and evaluation metrics such as ACC, Loss, FLOPs, and computational time.

The FLOPs and runtime comparisons in Appendix G.5 provide empirical validation of the efficiency of different sparse Hopfield variants. The results confirm that Random Feature Hopfield, Linear Hopfield, and Window Hopfield achieve reduced FLOPs with minimal performance loss, supporting the claim of computational efficiency.

However, some concerns remain:

* Empirical validation of theoretical retrieval error bounds is missing. While the theoretical framework is rigorous, numerical verification comparing predicted retrieval errors with actual retrieval performance would further strengthen the findings.
* Lack of clear interpretation of experimental results. In Appendix G.1–G.4, the figures and tables present results, but there is no explicit discussion of how these findings support the theoretical claims. A clearer explanation of the key takeaways would improve the clarity and impact of the experimental section.

**Methods And Evaluation Criteria:**

The main body of the paper primarily focuses on theoretical derivations, while experiments are mainly presented in the appendix. Each experiment includes a corresponding description of the models and parameters used, ensuring clarity in implementation details.
The concerns regarding the experimental validation have already been discussed in the previous question.

**Other Comments Or Suggestions:**

On line 100, right side, the phrase "We also verify their efficacy through thorough numerical" is repetitive.

**Other Strengths And Weaknesses:**

One of the major strengths of this paper is its clear and well-structured presentation. Despite being a theoretical work, the explanations are concise, logically organized, and easy to follow, making the complex mathematical formulations accessible. The clarity of writing ensures that the theoretical contributions are well-communicated without unnecessary complexity.

**Questions For Authors:**

1. The paper provides rigorous theoretical retrieval error bounds, but there is no numerical verification comparing predicted vs. actual retrieval performance. Could you provide empirical validation to confirm that the theoretical bounds accurately reflect practical performance?
2. The appendix suggests that Hu et al., 2023 achieves better capacity and robustness in many cases. What are the key advantages of the proposed model over Hu et al., 2023, beyond computational complexity?
3. The figures and tables in Appendix G.1–G.4 lack clear explanations regarding how they support the theoretical claims. Could you provide a more explicit discussion of the key takeaways from these results?
4. Some conclusions, such as tighter retrieval error bounds and stronger noise robustness, seem to be natural extensions of classical Hopfield models. What new theoretical insights does your framework provide beyond these expected properties?

**Relation To Broader Scientific Literature:**

This paper introduces a novel approach by leveraging soft-margin SVR to reformulate modern Hopfield models as a non-parametric regression task. To the best of my knowledge, this specific formulation has not been explored in prior work, making it a notable contribution to the field.

Building on this framework, the paper proposes a generalized Sparse-Structured Modern Hopfield Model, which aims to enhance computational efficiency and scalability. However, based on the experimental results in the appendix, the proposed model does not consistently outperform existing methods. Specifically, Hu et al., 2023 appears to achieve better capacity and robustness in many cases.

Furthermore, while the paper derives sparsity-dependent retrieval error bounds and proves fixed-point convergence, some of the additional conclusions—such as tighter retrieval error bounds and stronger noise robustness—are relatively intuitive extensions of classical Hopfield networks. These properties are already well understood in the standard Hopfield model, raising the question of whether the new framework offers fundamentally novel insights beyond a reformulation of known results.

**Theoretical Claims:**

I carefully reviewed the theoretical analysis presented in the main text and found the derivations to be well-structured and logically sound. Additionally, I briefly browsed the proofs in appendix C and D and did not notice any apparent issues.

---

> ### Author Rebuttal · Authors · 2025-03-26
>
> ## Reviewer’s Comment (Claims and Evidence & Experimental Designs or Analyses)
> > **Concerns**
> > - Claim2: The theoretical results (noise robustness, tighter retrieval error bounds) do not appear numerically verified in detail.
> > - Claim3: The experimental section doesn’t explicitly connect to the theory.
>
> > The paper lacks explicit discussion tying the figures/tables in Appendix G.1–G.4 to the theoretical claims.
> > Empirical validation of the theoretical retrieval error bounds is missing.
>
> ### **Authors’ Response**
>
> **In response to Concerns of Claim 2**, numerical verifications of the retrieval error bounds appear in **Sec. G.1** and are summarized in **Fig. 2** (pages 29–30).
>
>
> **In response to Concerns of Claim 3,** we believe there might be an oversight. In the “Results” paragraphs, as well as the captions of **Figs. 2 and 4**, we explicitly linked our numerical findings to their underlying theoretical claims.
>
> We feel that these “Results” paragraphs and captions are sufficiently informative (also check our concluding remarks, line 411-420, right).
>
> We apologize for any confusion caused. We have revised the draft with better clarity  (e.g., https://imgur.com/a/OS2q77m and https://imgur.com/a/JjxZwxv and https://imgur.com/a/50mFsX5.)
>
> ---
>
> ## Reviewer’s Comment (Relation to Broader Scientific Literature)
> > Certain properties, such as tighter error bounds and stronger noise robustness, might be seen as natural extensions of classical Hopfield networks. What’s truly novel here?
>
> ### **Authors’ Response**
> 1. **Nonparametric Framework**: Prior works do not formulate Hopfield retrieval as a **soft-margin SVR**, which is our key novelty. By bridging Hopfield networks with regression theory, we gain new insights and design flexibility (e.g., kernelized or masked retrieval).
> 2. **Sub-Quadratic Sparse Hopfield**: We extend the dense model to a *computationally efficient* variant while retaining classical benefits (exponential capacity, fixed-point convergence).
> 3. **Sparsity Analysis**: Our results detail *when* sparse retrieval can outperform dense retrieval, complementing prior sparse Hopfield analyses (Hu et al., 2023).
>
> ---
>
> ## Reviewer’s Comment (Essential References Not Discussed)
> > ... Amari (1972) for earlier foundational work...
>
> ### **Authors’ Response**
> Thanks for the suggestion. We had updated the draft accordingly (https://imgur.com/a/IYMKyu3).
>
> ---
>
> ## Reviewer’s Comment (Questions for Authors)
> > 1. Are the theoretical retrieval error bounds actually verified numerically?
> > 2. How does this approach compare to Hu et al. (2023), which reportedly achieves better capacity and robustness in some cases?
> > 3. Could you clarify the main takeaways of the figures/tables in Appendix G.1–G.4?
> > 4. Are these “tighter” bounds and “stronger noise robustness” truly novel beyond standard Hopfield extensions?
>
> ### **Authors’ Response**
>
> 1. **Numerical Verification (Q1)**
>    - Yes. **Sec. G.1 (Fig. 2)** (pp. 29–30) shows empirical vs. predicted retrieval error for varied sparsity levels, confirming Theorem 4.1. Please also see https://imgur.com/a/JjxZwxv. We can add further numeric comparisons if requested.
>
> 2. **Comparison with Hu et al. (2023) (Q2)**
>    - **Fundamentally Different Mechanisms**: Hu et al. rely on a data-**dependent** entropic regularizer (still $O(n^2)$). Our approach imposes data-**independent** masks for **sub-quadratic** retrieval.
>    - **Focus on Efficiency**: We do **not** claim consistent accuracy superiority; rather, we stress efficiency (Thm 3.2).
>    - **Extended Theory**: Our SVR-based approach and explicit sparsity scaling (Theorem 4.1, Remark 3.6) broaden the theory beyond Hu et al.’s entropic analysis.
>
> 3. **Main Takeaways in Appendix G (Q3)**
>    - **Sec. G.1–G.4**: We evaluate different sparse strategies (top-$k$, random masks, etc.) under noise and large-scale tasks. Each figure corresponds to Thm 4.1 or Coro 4.11 or Prop 4.1, showing that careful sparsity can match or surpass dense retrieval in sparse regimes.
>
> 4. **Novel Theoretical Insights (Q4)**
>     - We recast retrieval as a **soft-margin SVR** problem, yielding **margin-based error bounds** and **stronger noise-robustness** than purely energy-based or entropic methods (e.g., Hu et al., 2023).
>     - This framework supports a **diverse range of sparsity schemes**—top-$k$, random masking, or learned patterns—yet still guarantees **fixed-point convergence** and **exponential capacity**.
>     - **Explicit sparsity-dependent bounds** show how selectively ignoring irrelevant patterns can *improve* retrieval, providing concrete criteria for balancing accuracy and efficiency.
>     - Consequently, these “tighter” bounds and “stronger” robustness are not minor tweaks to classical Hopfield models; rather, they stem from the **SVR-based margin analysis** and support **sub-quadratic** implementations for large-scale memory retrieval.
>
> ---
>
> Thank you for the review. Happy to discuss more :)

---

> > ### Comment · Reviewer_zGHJ · 2025-04-05
> >
> > I have carefully read the authors’ response and the revised version of the paper. I sincerely appreciate the authors’ efforts and clarifications. The content in Appendix G is now significantly clearer than before, and the alignment between theory and experiments is much improved. While these changes have enhanced the presentation, they do not substantially change the basis of my original evaluation. As such, I do not intend to change my score at this point.

---

> > > ### Author Response · Authors · 2025-04-05
> > >
> > > We are glad our revisions have improved clarity. Thank you again for your detailed review and for your time and effort. Your constructive comments have certainly improved this work!

---

### Official Review · Reviewer_DjFS · 2025-03-14

**Overall Recommendation:** 3

**Summary:**

In this work, the authors replace the energy minimization step by support vector regression which trains on the pairs of a pattern and its perturbed version, regressing the true pattern, in terms of perturbed ones. They also provide a sparse version which uses a subset of patterns.  In addition, they perform synthetic and realistic memory retrieval and supervised learning tasks with this model.

## Update after rebuttal

I requested that the authors should clarify what they mean by the sparse model inheriting the appealing theoretical properties of the dense model so that the reader does not get a false impression, which they agreed to. I have upgraded the score to 3.

**Claims And Evidence:**

They claim they propose a nonparametric framework for deep learning compatible modern Hopfield network. Turning approximate retrieval into a regression problem is interesting, but seeing approximate retrieval as denoising makes sense. They say they introduce the first efficient sparse modern Hopfield model with sub-quadratic complexity, which is true. In the abstract, they claim the sparse model inherits the appealing theoretical properties of its dense analog. This is where I begin to get confused. Results like Corollary 4.11 showing the error of sparse recovery being lower than dense recovery only apply to patterns kept in the sparse model. Target patterns that are masked out may not get properly recovered. This seems to be borne out by the empirical results. From what I can understand the sparse Hopfield model in Hu et al 2023 is a very different thing where the regularizer is chosen so that attention is sparse. Sparsity by random masking is not the same thing. Thus their claims of sparsity-induced advantages (the topic of many of their theoretical results) ring a bit hollow to me.

After the discussion with the authors, I feel they should clarify this point in the main text.

**Essential References Not Discussed:**

I am not aware of any.

**Experimental Designs Or Analyses:**

They are sound.

**Methods And Evaluation Criteria:**

The benchmarks and methods are fine.

**Other Comments Or Suggestions:**

There are a few typos, like page 4: $\mathcal T_\Xi(x):\mathbb{R}^d\to\mathbb{R}^d$.

**Other Strengths And Weaknesses:**

No comments.

**Questions For Authors:**

I hope I am misunderstanding something. I keep feeling the general statements made about sparse Hopfield model defined here are too good to be true, as explained above.

Also why is $\Phi$ defined in Lemma 3.1 so that we weigh $\xi_mu$ with the attention between $x$ and $\xi_mu+\delta \xi_mu$, rather than with $\xi_mu$ (Eq. 3.5)?

**Relation To Broader Scientific Literature:**

Replacing the energy model by a trainable support vector regression is an interesting idea with the basis functions inspired by softmax attention. Perhaps we do not need the crutch of Hopfield models anymore?

**Theoretical Claims:**

The theoretical claims seem correct but they are of limited use, if sparse model only is good for subset of patterns. Am I missing something?

---

> ### Author Rebuttal · Authors · 2025-03-26
>
> The revised draft is in this [anonymous Dropbox folder](https://www.dropbox.com/scl/fo/81sygn18f4ma3ridm1xlf/AHFYlvzMMlYZnNRhBN9U8mw?rlkey=e1tvpqs6v83kx2rspfvmfgswh&st=z9iuk4iu&dl=0).  All modifications are marked in BLUE color. Thanks!
>
> ---
>
> ### **Reviewer’s Comment (Claims and Evidence 1)**
> > *In the abstract, they claim the sparse model **inherits the appealing theoretical properties** of its dense analogue...Results like Corollary 4.11 showing the error of sparse recovery being lower than dense recovery **only applies to patterns kept in the sparse model**. Target patterns which are masked out may not get properly recovered. This seems to be borne out by the empirical results.*
>
> ---
>
> ### **Authors’ Response**
>
> **When we say the sparse model “inherits” the theoretical properties of the dense version, we mean it retains the key Hopfield guarantees** (e.g., alignment with transformer attention, rapid convergence to fixed-point memories, exponential storage capacity). Our results show that introducing sparsity does not break these fundamental properties.
>
> Yes, performance can degrade when target patterns are masked out. **This reflects a general accuracy–efficiency tradeoff** (see lines 421–431). However, our numerical results (Fig. 4, Sec. G.5) show that, in practice, the degradation is mild: sparse masking achieves decent accuracy while significantly improving efficiency (similar to how linear/random feature approximations to softmax perform in other contexts).
>
> ---
>
> ### **Reviewer’s Comment (Theoretical Claims)**
> > *The theoretical claims seem correct but they are of limited use if the sparse model is only good for a subset of patterns. Am I missing something?*
>
> ---
>
> ### **Authors’ Response**
> **We do NOT claim the sparse model always outperforms the dense model in all situations.** Instead, we specify **conditions** where sparse retrieval can yield better accuracy (e.g., when many irrelevant memories introduce noise). Under those conditions, our empirical results (e.g., Fig. 2, p. 30) support the theoretical claim https://imgur.com/a/JjxZwxv.
>
> ---
>
> ### **Reviewer’s Comment (Claims And Evidence 2)**
> > *From what I can understand, the sparse Hopfield model in Hu et al. (2023) is a very different thing where the regularizer is chosen so that attention is sparse. Sparsity by random masking is not the same thing. Thus, their claims of sparsity-induced advantages ring a bit hollow to me.*
>
> ---
>
> ### **Authors’ Response: Sparse Hopfield Model – Sparsity Masking vs. Regularization**
> We appreciate the comparison. To clarify, our model uses **sparse masking** (Definition 3.3), with random masking as one special case (Example 1 in line 275). In **Remark 4.1 (line 368)**, we compared our approach to [Hu et al. 2023]. Here is a brief overview:
>
> - **Different Sparsity Mechanisms**
>   - **Ours:** Data-***independent***, predefined masking for efficiency.
>   - **Hu et al. 2023:** Data-***dependent***, regularizer-based sparsity (but still with $O(n^2)$ complexity like [Ramsauer et al. 2020]).
>
> - **Focus on Analysis**
>   We study how controlling the sparsity dimension $k$ affects retrieval accuracy (Theorem 4.1). Hu et al. (2023) do not provide that level of interpretability.
>
> - **No Claim of Superiority**
>   We do not assert that random/sparse masking categorically outperforms regularizer-based methods; we emphasize their **analytical tractability** and **efficiency**.
>
> ---
>
> ### **Reviewer’s Comment (Questions For Authors)**
> > *Why is $\Phi$ defined in Lemma 3.1 to weigh $\xi_{\mu}$ by the attention between $x$ and $\xi_{\mu} + \delta \xi_{\mu}$, rather than $\xi_{\mu}$ itself (Eq. 3.5)?*
>
> ---
>
> ### **Authors’ Response**
> 1. **SVR-Based Nonparametric Interpretation for MHM** (lines 172–202; 159–169; Appendix C.1)
>    - We cast Hopfield retrieval as a soft-margin SVR.
>    - Each stored pattern $\xi_\mu$ is paired with $\xi_\mu + \delta\xi_\mu$.
>    - This enforces the SVR model to map any noisy query near $\xi_\mu$ back to $\xi_\mu$.
>    -  Equation (3.2) enforces $\epsilon$-retrieval (Definition 3.2).
>
>
> 2. **Aligning with Basin of Attraction (lines 258–274) and $\epsilon$-Retrieval**
>    - Hopfield models attract queries near $\xi_{\mu}$ back to $\xi_{\mu}$.
>    - $\xi_{\mu} + \delta\xi_{\mu}$ defines a local basin around each pattern.
>    - The SVR constraints ensure $\lvert T_{\mathrm{SVR},\Phi}(x) - \xi_{\mu}\rvert \le \epsilon$ within that basin, enabling **noise-robust retrieval**  (from partial or corrupted queries i.e., Definition 3.2.)
>
> 3. **Avoiding Trivial SVR Solutions**
>    - Using $(\xi_{\mu}, \xi_{\mu})$ only would let the system learn the **identity** (no actual denoising).
>    - Including $\delta\xi_{\mu}$ forces **true associative memory** behavior.
>
> Overall, using $\xi_{\mu} + \delta\xi_{\mu}$ ensures the nonparametric retrieval aligns with Hopfield’s attractor dynamics, supports noise-robust memory retrieval, and prevents trivial SVR solutions.
>
> ---
>
> Thank you for the review. Happy to discuss more :)

---

> > ### Comment · Reviewer_DjFS · 2025-04-07
> >
> > I feel the authors should clarify what they mean by the sparse model inheriting the appealing theoretical properties of the dense model so that the reader does not get a false impression. I am upgrading the score to 3.

---

> > > ### Author Response · Authors · 2025-04-08
> > >
> > > We are very happy that our rebuttal and revision have met your expectations. Yes, we will clarify the "inherit" issue thoroughly in the final version. Thank you again for your time, effort, and attention to detail! Your constructive comments have certainly improved this work!

---

### Decision · Program_Chairs · 2025-05-01

**Decision:**

Accept (poster)

**Comment:**

This paper reformulate modern Hopfield models as a non-parametric regression task tackled with soft-margin Support Vector Regression,  where noisy patterns are mapped to reconstructed target patterns.

The reviewers point out as strengths the originality of the approach on the context of Hopfield networls, as well as its generalizability to existing methods of single-step associative memories. The new formulation significantly reduces computational complexity. The main weaknesses are the limited and unclear empirical results (included only in the appendix) which are not sufficient to prove the effectiveness of the proposed approach. There were also some concerns about what "sparse" means in the context of this work in comparison to prior work on sparse Hopfield networks. The paper would benefit also from improving the background and discussing the limitations of the proposed method. The authors' rebuttal addresses several of these concerns, which makes me lean towards acceptance.